# Biosynthetic potential of the global ocean microbiome

Lucas Paoli[1], Hans-Joachim Ruscheweyh[1,18], Clarissa C. Forneris[2,18], Florian Hubrich[2,18], Satria Kautsar[3], Agneya Bhushan[2], Alessandro Lotti[2], Quentin Clayssen[1], Guillem Salazar[1], Alessio Milanese[1], Charlotte I. Carlström[1], Chrysa Papadopoulou[1], Daniel Gehrig[1], Mikhail Karasikov[4,5,6], Harun Mustafa[4,5,6], Martin Larralde[7], Laura M. Carroll[7], Pablo Sánchez[8], Ahmed A. Zayed[9], Dylan R. Cronin[9], Silvia G. Acinas[8], Peer Bork[7,10,11], Chris Bowler[12,13], Tom O. Delmont[13,14], Josep M. Gasol[8], Alvar D. Gossert[15], André Kahles[4,5,6], Matthew B. Sullivan[8,16], Patrick Wincker[13,14], Georg Zeller[7], Serina L. Robinson[2,17✉], Jörn Piel[2✉] & Shinichi Sunagawa[1✉]

Natural microbial communities are phylogenetically and metabolically diverse. In addition to underexplored organismal groups[1], this diversity encompasses a rich discovery potential for ecologically and biotechnologically relevant enzymes and biochemical compounds[2,3]. However, studying this diversity to identify genomic pathways for the synthesis of such compounds[4] and assigning them to their respective hosts remains challenging. The biosynthetic potential of microorganisms in the open ocean remains largely uncharted owing to limitations in the analysis of genome-resolved data at the global scale. Here we investigated the diversity and novelty of biosynthetic gene clusters in the ocean by integrating around 10,000 microbial genomes from cultivated and single cells with more than 25,000 newly reconstructed draft genomes from more than 1,000 seawater samples. These efforts revealed approximately 40,000 putative mostly new biosynthetic gene clusters, several of which were found in previously unsuspected phylogenetic groups. Among these groups, we identified a lineage rich in biosynthetic gene clusters ('*Candidatus* Eudoremicrobiaceae') that belongs to an uncultivated bacterial phylum and includes some of the most biosynthetically diverse microorganisms in this environment. From these, we characterized the phospeptin and pythonamide pathways, revealing cases of unusual bioactive compound structure and enzymology, respectively. Together, this research demonstrates how microbiomics-driven strategies can enable the investigation of previously undescribed enzymes and natural products in underexplored microbial groups and environments.

Microorganisms drive global biogeochemical cycles, support food webs, and underpin the health of animals and plants[5]. Their immense phylogenetic, metabolic and functional diversity represents a rich discovery potential for new taxa[1], enzymes and biochemical compounds, including natural products[6]. In environmental communities, such molecules confer microorganisms with diverse physiological and ecological functions ranging from communication to competition[2,7]. In addition to their original functions, these natural products and their genetically encoded production pathways include examples used for biotechnological and therapeutic applications[2,3]. The identification of such pathways and compounds has largely been facilitated by studying cultivable microorganisms. However, taxonomic surveys of natural environments have revealed that the vast majority of microbial life has not yet been cultivated[8]. This cultivation bias has limited our ability to tap into much of the microbially encoded functional diversity[4,9].

[1]Department of Biology, Institute of Microbiology and Swiss Institute of Bioinformatics, ETH Zurich, Zurich, Switzerland. [2]Department of Biology, Institute of Microbiology, ETH Zurich, Zurich, Switzerland. [3]Bioinformatics Group, Wageningen University, Wageningen, The Netherlands. [4]Department of Computer Science, ETH Zurich, Zurich, Switzerland. [5]Biomedical Informatics Research, University Hospital Zurich, Zurich, Switzerland. [6]Swiss Institute of Bioinformatics, Lausanne, Switzerland. [7]Structural and Computational Biology Unit, European Molecular Biology Laboratory, Heidelberg, Germany. [8]Department of Marine Biology and Oceanography, Institute of Marine Sciences ICM-CSIC, Barcelona, Spain. [9]Center of Microbiome Science, EMERGE Biology Integration Institute, Department of Microbiology, The Ohio State University, Columbus, OH, USA. [10]Max Delbrück Centre for Molecular Medicine, Berlin, Germany. [11]Department of Bioinformatics, Biocenter, University of Würzburg, Würzburg, Germany. [12]Institut de Biologie de l'ENS (IBENS), Département de biologie, École normale supérieure, CNRS, INSERM, Université PSL, Paris, France. [13]Research Federation for the Study of Global Ocean Systems Ecology and Evolution, FR2022/Tara Oceans GOSEE, Paris, France. [14]Metabolic Genomics, Genoscope, Institut de Biologie François Jacob, CEA, CNRS, Univ Evry, Université Paris Saclay, Evry, France. [15]Department of Biology, Biomolecular NMR Spectroscopy Platform, ETH Zurich, Zurich, Switzerland. [16]Department of Civil, Environmental and Geodetic Engineering, The Ohio State University, Columbus, OH, USA. [17]Department of Environmental Microbiology, Swiss Federal Institute of Aquatic Science and Technology (Eawag), Dübendorf, Switzerland. [18]These authors contributed equally: Hans-Joachim Ruscheweyh, Clarissa C. Forneris, Florian Hubrich. ✉e-mail: Serina.Robinson@eawag.ch; jpiel@ethz.ch; ssunagawa@ethz.ch

To overcome these limitations, technological advances over the past decade have enabled researchers to directly (that is, without previous cultivation) sequence pieces of microbial DNA from whole communities (metagenomics) or single cells. The possibility to assemble such pieces into larger genomic fragments and to reconstruct several metagenome assembled genomes (MAGs) or single amplified genomes (SAGs), respectively, has opened new paths to the previously taxon-centric investigation of microbiomes (that is, microbial communities and their genetic material in a given environment)[10–12]. Indeed, recent surveys have vastly extended the phylogenomic representation of microbial diversity on Earth[1,13] and revealed that most of the functional diversity in different microbiomes had previously not been captured by reference genome sequences (REFs) from cultivated microorganisms[14]. The ability to place uncovered functional diversity into the host genomic (that is, genome-resolved) context has been critical to predict yet uncharacterized microbial lineages that putatively encode new natural products[15,16] or to trace such compounds to their original producers[17]. A combinatorial approach of metagenomic and single-cell genomic analyses, for example, led to the recognition of 'Candidatus Entotheonella', a group of metabolically rich, sponge-associated bacteria, as producers of multiple classes of candidate drugs[18]. However, despite recent attempts to establish genome-resolved explorations of various microbiomes[16,19], for the ocean—the largest ecosystem on Earth—over two-thirds of global metagenomic data still remain unaccounted for[16,20]. Thus, the biosynthetic potential of the ocean microbiome in general and its potential as a reservoir of new enzymology and natural products specifically remain largely underexplored.

To examine the biosynthetic potential of the ocean microbiome at the global scale, we first integrated ocean microbial genomes obtained using cultivation-dependent and cultivation-independent methods to establish an extensive phylogenomic and gene functional database. By mining this database, we uncovered a diverse array of biosynthetic gene clusters (BGCs), the majority of them from yet un-characterized gene cluster families (GCFs). We further identified an uncharted family of bacteria that display the highest known diversity of BGCs in the open oceans to date. We selected two ribosomally synthesized and post-translationally modified peptide (RiPP) pathways on the basis of their genetic dissimilarity to currently known ones for experimental validation. Functional characterization of these pathways revealed examples of unexpected enzymology as well as a structurally unusual compound with protease inhibitory activity.

## Phylogenomic representation of the ocean microbiome

We first sought to establish a global genome-resolved data resource focusing on its bacterial and archaeal constituents. To this end, we aggregated metagenomic data along with contextual information from 1,038 ocean water samples from 215 globally distributed sampling sites (latitudinal range = 141.6°) and several depth layers (from 1 to 5,600 m deep, covering epipelagic, mesopelagic and bathypelagic zones)[21–23] (Fig. 1a, Extended Data Fig. 1a and Supplementary Table 1). In addition to providing broad geographical coverage, these size-selectively filtered samples enabled us to compare different components of the ocean microbiome, including virus-enriched (<0.2 μm), prokaryote-enriched (0.2–3 μm), particle-enriched (0.8–20 μm) and virus-depleted (>0.2 μm) communities.

Using this dataset, we reconstructed a total of 26,293 predominantly bacterial and archaeal MAGs (Fig. 1b and Extended Data Fig. 1b). We generated these MAGs on the basis of assemblies from individual, rather than pooled, metagenomic samples to prevent the collapsing of natural sequence variations across samples from different locations or time points (Methods). Furthermore, we grouped genomic fragments on the basis of their abundance correlation across large numbers of samples (between 58 and 610 samples, depending on the survey; Methods). We

found this to be a computationally intensive, yet important step[24], that was omitted in several large-scale MAG reconstruction efforts[16,19,25], and substantially improved both the number (mean, 2.7 times) and quality score (mean, +20%) of genomes reconstructed from the ocean metagenomes studied here (Extended Data Fig. 2a and Supplementary Information). Overall, these efforts have increased the number of ocean water microbial MAGs by a factor of 4.5 (6 when counting high-quality MAGs only) compared with the most comprehensive MAG resource available to date[16] (Methods). This set of newly created MAGs was then combined with 830 manually curated MAGs[26], 5,969 SAGs[27] and 1,707 REFs.[27] of marine bacteria and archaea into a combined collection of 34,799 genomes (Fig. 1b).

We next evaluated the newly established resource for its improved ability to represent ocean microbial communities and to assess the impact of integrating different genome types. On average, we found that it captured about 40–60% of ocean metagenomic data (Fig. 1c), corresponding to a two- to threefold increase in coverage with a more consistent representation across depths and latitudes compared with previous reports based solely on MAGs[16] or SAGs[20]. Furthermore, to obtain a systematic measure of the taxonomic diversity within the established collection, we annotated all genomes using the Genome Taxonomy Database (GTDB) Toolkit (Methods) and clustered them using a 95% whole-genome average nucleotide identity cut-off[28] to define 8,304 species-level clusters (species). Two thirds of these species (including new clades) were previously not represented in the GTDB and 2,790 of them were uncovered by MAGs reconstructed in this study (Fig. 1d). Moreover, we found that the different genome types were highly complementary, with 55%, 26% and 11% of the species being exclusively composed of MAGs, SAGs and REFs, respectively (Fig. 1e). Furthermore, MAGs covered all 49 phyla detected in the water column, whereas SAGs and REFs represented only 18 and 11 of them, respectively. However, SAGs better represented the diversity of the most abundant clades (Extended Data Fig. 3a), such as the order Pelagibacterales (SAR11), with nearly 1,300 species covered by SAGs as opposed to only 390 by MAGs. Notably, REFs rarely overlapped with either MAGs or SAGs at the species level and represented >95% of the approximately 1,000 genomes that were not detected in the set of open ocean metagenomes studied here (Methods), mostly owing to representatives that were isolated from other types of marine samples (such as sediments or host-associated). To enable its broad use by the scientific community, this ocean genomic resource—which also includes unbinned fragments (for example, from predicted phages, genomic islands and fragments of genomes with insufficient data for MAG reconstruction)—can be accessed alongside taxonomic and gene functional annotations as well as contextual environmental parameters at the Ocean Microbiomics Database (OMD; https://microbiomics.io/ocean/).

## Biosynthetic potential of the global ocean microbiome

Next, we set out to investigate the richness and the degree of novelty of the biosynthetic potential in the open ocean microbiome. To this end, we first used antiSMASH on all of the MAGs, SAGs and REFs detected in the set of 1,038 ocean metagenomes (Methods) to predict a total of 39,055 BGCs. We then clustered them into 6,907 non-redundant GCFs and 151 gene cluster clans (GCCs; Supplementary Table 2 and Methods) to account for inherent redundancy (that is, the same BGC can be encoded in several genomes) and fragmentation of BGCs in metagenomic datasets. Incomplete BGCs did not significantly inflate, if at all (Supplementary Information), the number of GCFs and GCCs, which contained at least one complete member BGC in 44% and 86% of the cases, respectively.

At the GCC level, we found a high diversity of predicted RiPPs and other natural products (Fig. 2a). Among these, aryl polyenes, carotenoids, ectoines and siderophores, for example, belonged to GCCs with wide phylogenomic distributions and high prevalence across ocean

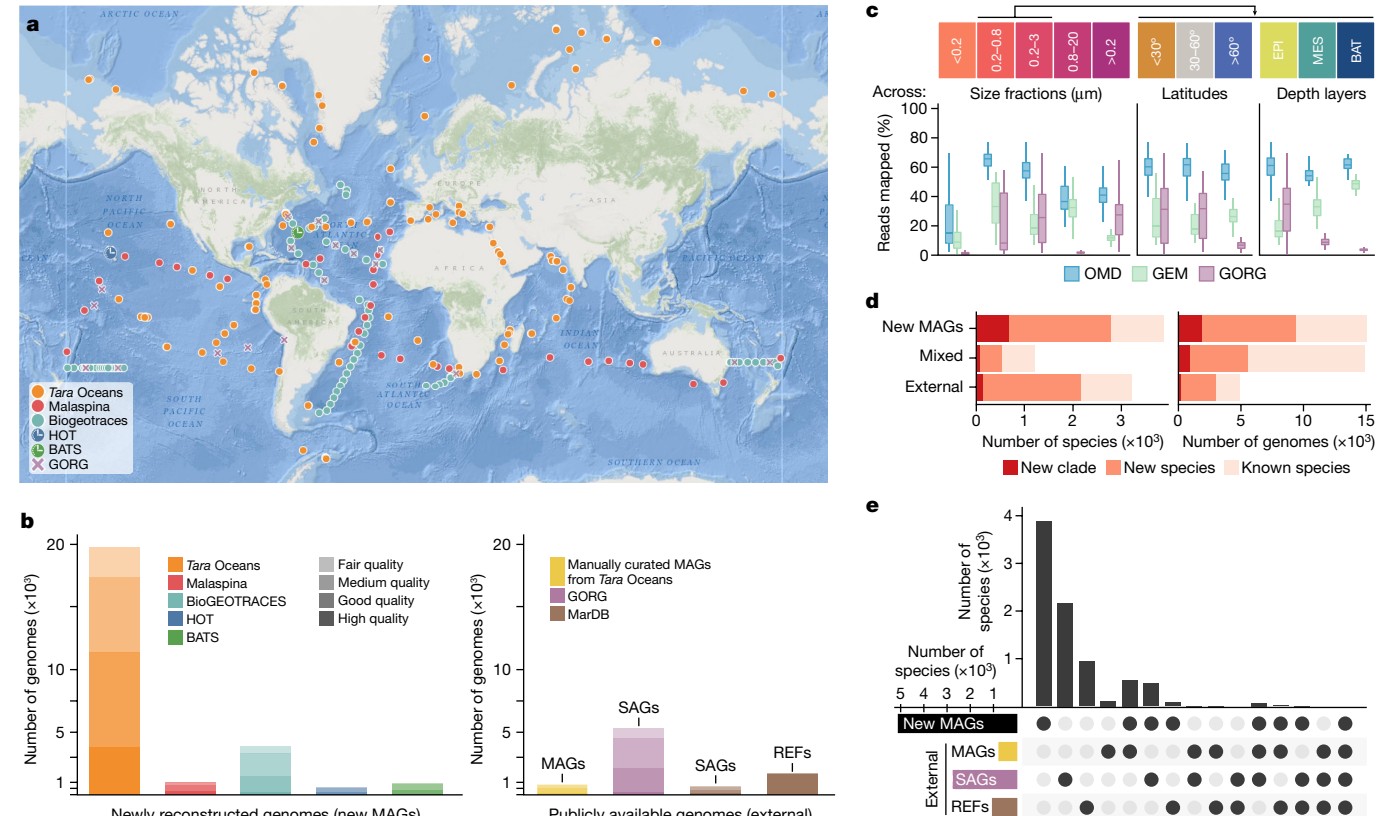

**Fig. 1 | Reconstruction of MAGs at the global scale fills gaps in ocean phylogenomic diversity. a**, A total of 1,038 publicly available ocean microbial community genomes (metagenomes) were collected at 215 globally distributed sites (between 62° S to 79° N and 179° W to 179° E). Map tiles © Esri. Sources: GEBCO, NOAA, CHS, OSU, UNH, CSUMB, National Geographic, DeLorme, NAVTEQ and Esri. **b**, These metagenomes were used to reconstruct MAGs (Methods and Supplementary Information), which varied in numbers and quality (Methods) across different datasets (colour coded). Reconstructed MAGs were complemented with publicly available (external) genomes, including manually curated MAGs[26], SAGs[27] and REFs.[27] to compile the OMD. **c**, The OMD improves the genomic representation (mapping rates of metagenomic reads; Methods) of ocean microbial communities by a factor of two to three compared with previous reports based solely on SAGs (GORG)[20] or MAGs (GEM)[16], with a more consistent representation across depth and latitudes. <0.2, n = 151; 0.2–0.8, n = 67; 0.2–3, n = 180; 0.8–20, n = 30; >0.2, n = 610; <30°, n = 132; 30–60°, n = 73; >60°, n = 42; EPI, n = 174; MES, n = 45; BAT, n = 28. **d**, Grouping the OMD into species-level (95% average nucleotide identity) clusters identified a total of around 8,300 species, over half of which were previously uncharacterized based on taxonomic annotations using the GTDB (release 89)[13]. **e**, A breakdown of the species by genome type reveals a high complementarity of MAGs, SAGs and REFs in capturing the phylogenomic diversity of the ocean microbiome. Specifically, 55%, 26% and 11% of the species were specific to MAGs, SAGs and REFs, respectively. BATS, Bermuda Atlantic Time-series; GEM, Genomes from Earth's Microbiomes; GORG, Global Ocean Reference Genomes; HOT, Hawaiian Ocean Time-series.

metagenomes, possibly indicative of widespread microbial adaptations to the ocean environment, including resistance to reactive oxygen species, oxidative and osmotic stress or uptake of iron (Supplementary Information). This functional diversity contrasted with recent analyses of ~1.2 million BGCs from any of the ~190,000 genomes deposited in the NCBI RefSeq database (BiG-FAM/RefSeq, hereafter RefSeq)[29] that showed a dominance of non-ribosomal peptide synthetase (NRPS) and polyketide synthase (PKS) BGCs (Supplementary Information). We also found that 44 (29%) GCCs were only remotely related to any RefSeq BGCs ($\bar{d}_{RefSeq} > 0.4$; Fig. 2a and Methods), and 53 (35%) GCCs were encoded only in MAGs, highlighting the potential for discovery of previously undescribed chemistry within the OMD. Given that each of these GCCs is likely to represent highly diverse biosynthetic functions, we further analysed the data at the level of GCFs, which aim to provide a more fine-grained grouping of BGCs that are predicted to encode similar natural products[29]. A total of 3,861 (56%) of the identified GCFs did not overlap with RefSeq and >97% of the GCFs were not represented in MIBiG, one of the most extensive databases of experimentally validated BGCs[30] (Fig. 2b). Although finding many potentially new pathways in an environment that is not well represented by reference genomes is not unexpected, our approach of dereplicating BGCs into GCFs before comparative analyses, which differs from previous reports[16], enabled us to provide unbiased novelty estimates. The majority of that novel diversity (3,012 GCFs, that is, 78%) corresponded to predicted terpenes, RiPPs or other natural products, and a large fraction (1,815 GCFs, that is, 47%) was encoded in phyla that are not generally known for their biosynthetic potential. As opposed to PKS and NRPS clusters, these compact BGCs are less likely to be fragmented during metagenomic assembly[31] and can make easier targets for time- and resource-consuming functional characterization of their products.

Beyond richness and novelty, we also investigated the biogeographical structuring of the ocean microbiome's biosynthetic potential. Grouping samples by the mean metagenomic copy number distribution of GCFs (Methods) revealed that low-latitude, epipelagic, prokaryote-enriched and virus-depleted communities, mostly from surface or deeper sunlit waters were enriched in RiPP and terpene BGCs. By contrast, polar, deep-ocean, virus-enriched and particle-enriched communities were associated with higher abundances of NRPS and PKS BGCs (Extended Data Fig. 4 and Supplementary Information). Finally, we found that well-studied tropical and epipelagic communities were the most promising sources of new terpenes (Extended Data Fig. 5a,b), and that the least explored communities (polar, deep, virus- and particle-enriched) had the highest potential for the discovery of NRPS, PKS, RiPPs and other natural products (Extended Data Fig. 5a).

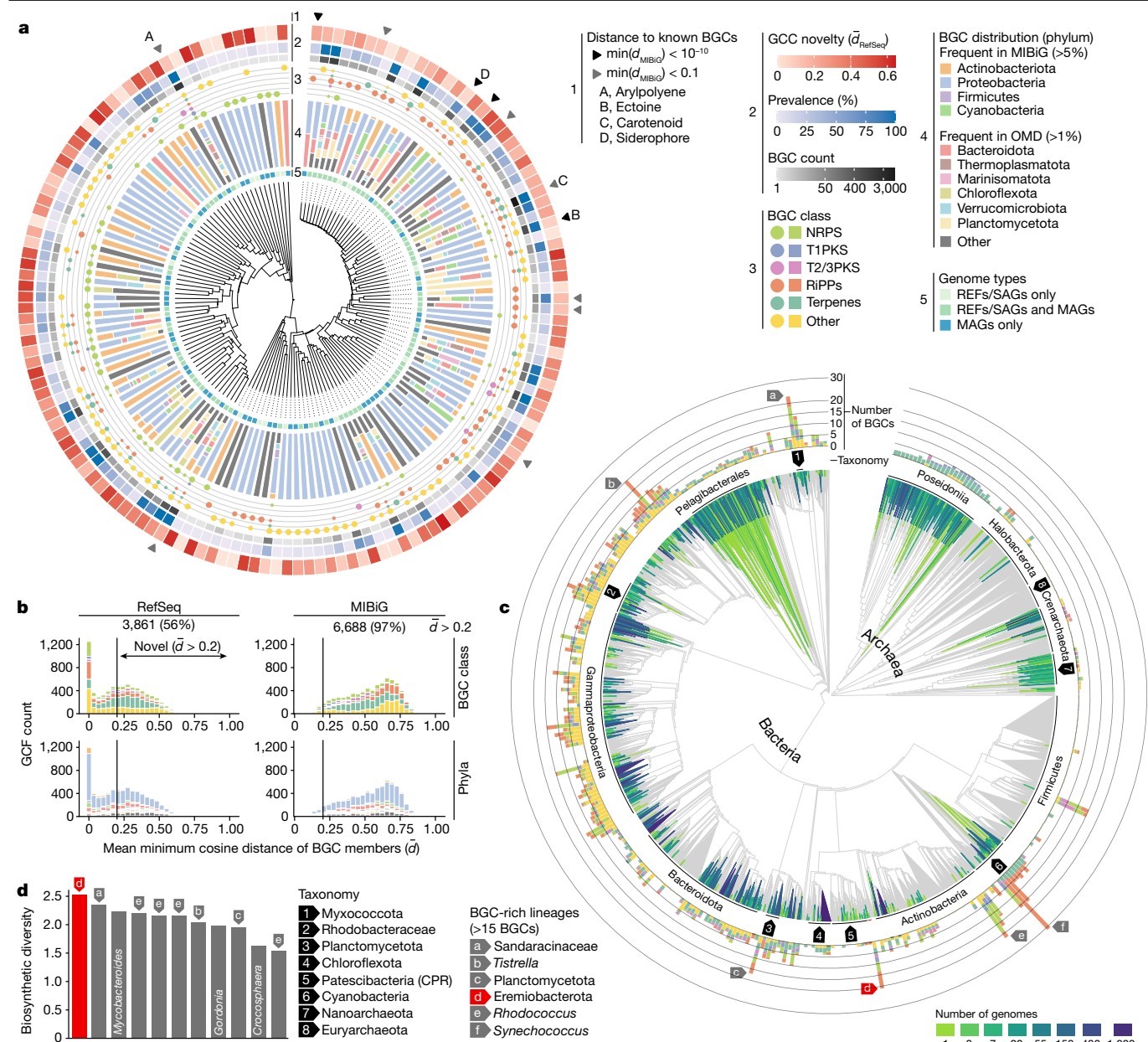

**Fig. 2 | Novelty and phylogenomic distribution of the ocean microbiome biosynthetic potential.** A total of 39,055 BGCs were clustered into 6,907 GCFs and 151 GCCs. **a**, Representation of the data (inner to outer layers). Hierarchical clustering based on BGC distances of the GCCs, 53 of which were captured only by MAGs. GCCs comprise BGCs from different taxa (ln-transformed phylum frequencies) and different BGC classes (circle sizes correspond to their frequencies). The outer layers indicate, for each GCC, the number of BGCs, the prevalence (percentage of samples) and the distance (minimum cosine distance of BGCs ($\min(d_{MIBiG})$)) to BGCs from BiG-FAM. GCCs with BGCs closely related to experimentally validated BGCs (MIBiG) are highlighted by arrows. **b**, Comparing GCFs to computationally predicted (BiG-FAM) and experimentally validated (MIBiG) BGCs uncovered 3,861 new ($\bar{d} > 0.2$) GCFs. Most of them (78%) encode

RiPPs, terpenes and other putative natural products. **c**, All genomes in the OMD detected across 1,038 ocean metagenomes were placed onto the GTDB backbone trees to reveal the extent of the phylogenomic coverage of the OMD. Clades without any genome in the OMD are coloured grey. The number of BGCs corresponds to the highest number of predicted BGCs per genome in a given clade. For visualization, the last 15% of the nodes were collapsed. The arrows denote BGC-rich clades (>15 BGCs) with the exception of *Mycobacteroides*, *Gordonia* (next to *Rhodococcus*) and *Crocosphaera* (next to *Synechococcus*). **d**, An unknown species of '*Ca*. Eremiobacterota' displayed the highest biosynthetic diversity (Shannon index based on natural product types). Each bar represents the genome with the highest number of BGCs within a species. T1PKS, type I PKS; T2/3PKS, type II and III PKS.

## Identification of undescribed BGC-rich lineages

To complement the survey of the biosynthetic potential of the ocean microbiome, we sought to map its phylogenomic distribution and identify new BGC-rich clades. To this end, we placed the ocean microbial genomes in the standardized bacterial and archaeal phylogenomic trees of the GTDB[13], and overlayed the putative biosynthetic

pathways that they encode (Fig. 2c). We readily detected in ocean water samples (Methods) several BGC-rich clades (representatives with >15 BGCs) that are either well known for their biosynthetic potential, such as Cyanobacteria (*Synechococcus*) and Proteobacteria (such as *Tistrella*)[32,33], or that have recently garnered attention for their natural products, such as Myxococcota (Sandaracinaceae), *Rhodococcus* and Planctomycetota[34–36]. Interestingly, we found within

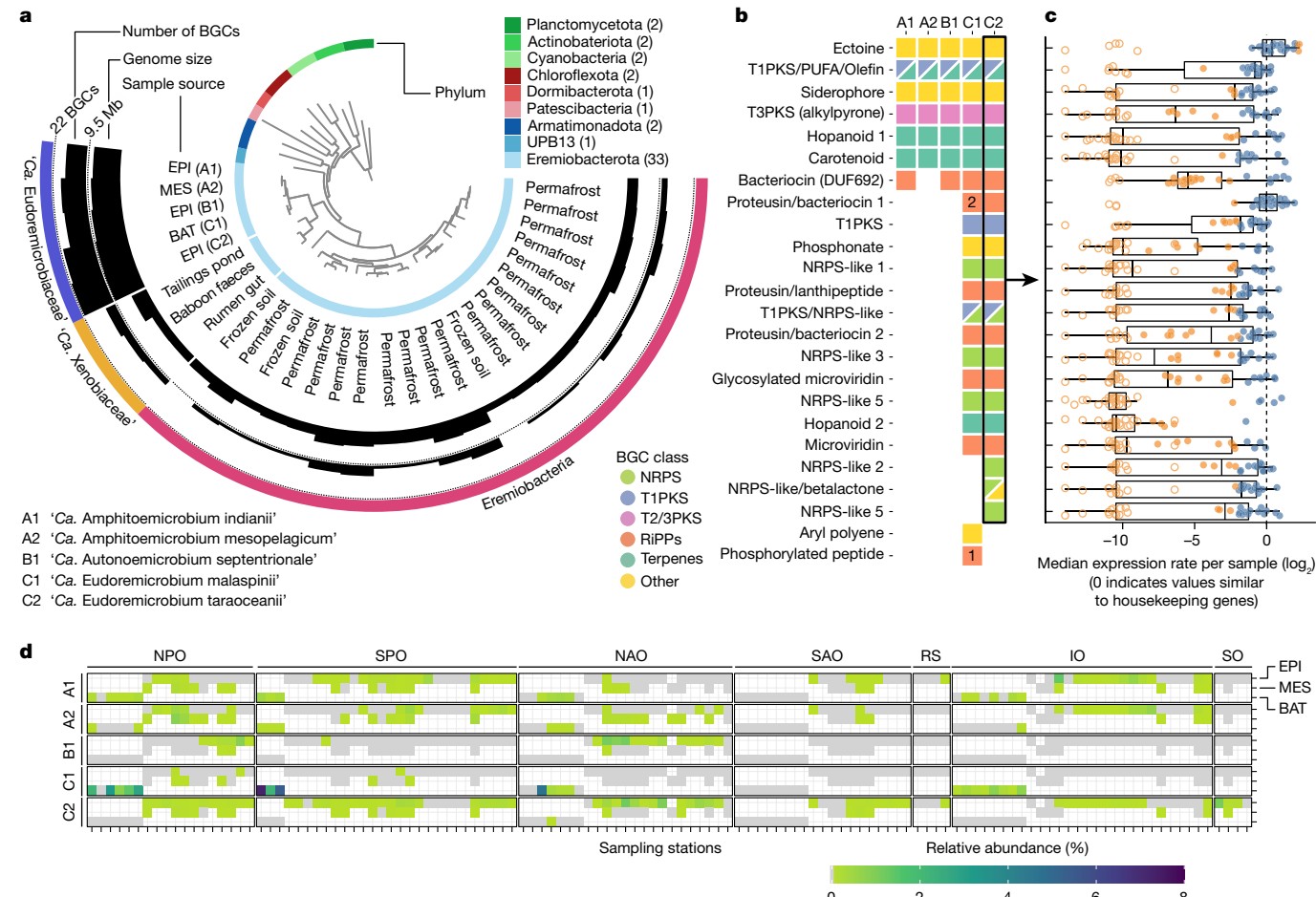

**Fig. 3 | Phylogeny, biosynthetic potential and distribution of the BGC-rich family 'Ca. Eudoremicrobiaceae'. a**, Phylogenomic placement of five 'Ca. Eudoremicrobiaceae' spp. revealed a BGC richness specific to the ocean lineage discovered in this study. The phylogenomic tree includes all 'Ca. Eremiobacterota' MAGs available in the GTDB (release 89) and representatives from additional phyla (the number of genomes is indicated in parentheses) for evolutionary context (Methods). The outermost layer indicates family-level ('Ca. Eudoremicrobiaceae' and 'Ca. Xenobiaceae') and class-level ('Ca. Eremiobacteria') taxonomy. The five species described in this study are denoted by an alphanumeric code and a proposed binomial name (Supplementary Information). **b**, 'Ca. Eudoremicrobiaceae' spp. share a core of seven BGCs. The missing BGC from clade A2 was attributed to incompleteness of the representative MAG (Supplementary Table 3). BGCs specific to 'Ca. Amphithomicrobium' and 'Ca. Amphithomicrobium' (clades A and B) are not

displayed. **c**, All BGCs encoded by 'Ca. Eudoremicrobium taraoceanii' were found to be expressed across the set of 623 metatranscriptomes sampled by *Tara* Oceans. The filled circles indicate active transcription. The orange circles indicate values below or above a $log_2$-transformed fold change from the expression rate of housekeeping genes (Methods). **d**, Relative abundance profiles (Methods) showed that 'Ca. Eudoremicrobiaceae' spp. are abundant and prevalent in most ocean basins and throughout the water column (from the surface to a depth of at least 4,000 m). On the basis of these estimations, we found that 'Ca. E. malaspinii' comprises up to 6% of the prokaryotic cells in bathypelagic particle-associated communities. We considered a species to be present at a station if it was detected in any of the size fractions of a given depth layer. IO, Indian Ocean; NAO, North Atlantic Ocean; NPO, North Pacific Ocean; RS, Red Sea; SAO, South Atlantic Ocean; SO, Southern Ocean; SPO, South Pacific Ocean.

these clades several previously unexplored lineages. For example, those species with the richest biosynthetic potential within the phyla Planctomycetota and Myxococcota belonged to an uncharacterized candidate order and genus, respectively (Supplementary Table 3). Overall, this shows that the OMD provides access to previously uncharted phylogenomic information, including for microorganisms that may represent new targets for the discovery of enzymes and natural products.

We further characterized BGC-rich clades not only by counting the maximum number of BGCs encoded by their members, but also by assessing the diversity of these BGCs, which accounts for the frequency of different candidate natural product types (Fig. 2c and Methods). We found that the most biosynthetically diverse species were represented by bacterial MAGs exclusively reconstructed in this study. These bacteria belong to the uncultivated phylum '*Candidatus*

Eremiobacterota', which has remained largely unexplored except in a few genomic studies[37,38]. Notably, 'Ca. Eremiobacterota' spp. have been analysed only from terrestrial environments[39] and have not been known to include any BGC-rich representatives. Here we initially reconstructed eight MAGs from the same species (with a nucleotide identity of >99%) from deep (between 2,000 m and 4,000 m) and particle-enriched (0.8–20 μm) ocean metagenomes collected by the Malaspina expedition[23]. Accordingly, we propose that this species is named '*Candidatus* Eudoremicrobium malaspinii', after the nereid (sea nymph) of fine gifts in Greek mythology and the expedition. 'Ca. E. malaspinii' had no previously known relatives below the order level based on phylogenomic annotation[13] and therefore belongs to a new bacterial family for which we propose 'Ca. E. malaspinii' as the type species and 'Ca. Eudoremicrobiaceae' as its official name (Supplementary Information). The short-read metagenomic reconstruction

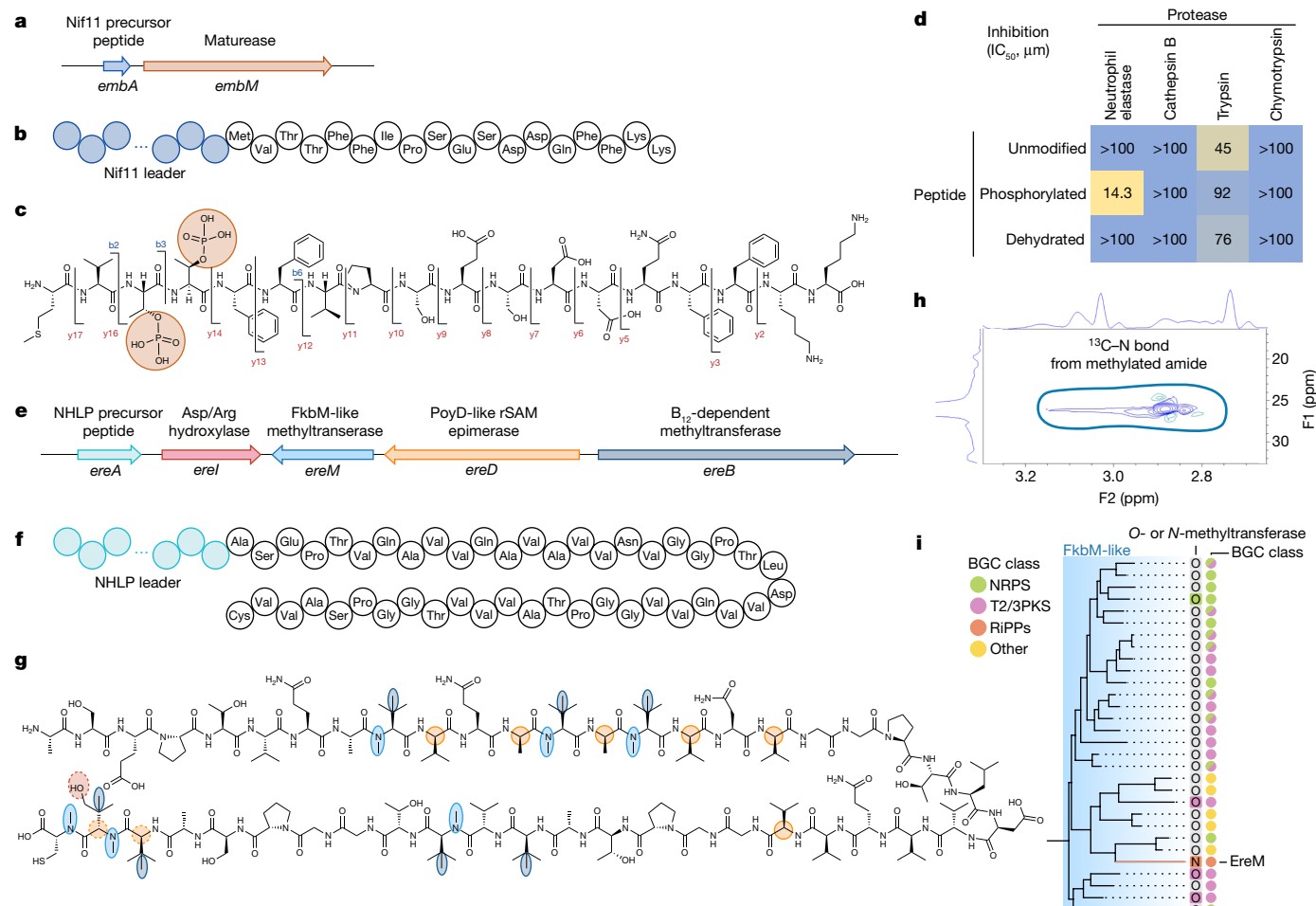

**Fig. 4 | 'Ca. Eudoremicrobiaceae' spp. are a source of unusual enzymology and natural product structure. a–c,** In vitro heterologous expression and in vitro enzyme assays of a novel ($\bar{d}_{RefSeq}$ = 0.29) RiPP biosynthetic cluster specific to the deep ocean species '*Ca*. E. malaspinii' led to the production of a di-phosphorylated product. **c,** Modifications were identified using high-resolution (HR) MS/MS (fragmentation is indicated by the b and y ions on the chemical structure) and NMR (Extended Data Fig. 9). **d,** This phosphorylated peptide displayed low-micromolar mammalian neutrophil elastase inhibition, which was not found for the control and dehydrated peptides (dehydration induced by chemical elimination). The experiment was repeated three times, leading to similar outcomes. **e–g,** Heterologous expression of a second novel $\bar{d}_{RefSeq}$ = 0.33) proteusin biosynthetic cluster sheds light on the functionality of four maturases modifying a 46-amino-acid core peptide. Residues are coloured on the basis of predicted modification sites from HR-MS/MS, isotope labelling and NMR analyses (Supplementary Information). Dashed colouring indicates that the modification occurs on either of the two residues. The figure represents a compilation of numerous heterologous constructs to display the activity of all maturases on the same core. **h,** Inset of the NMR data of the backbone amide *N*-methylation. The complete results are shown in Extended Data Fig. 10. **i,** Phylogenetic placement of the FkbM maturase of the proteusin cluster among all FkbM domains found in the MIBiG 2.0 database revealed an enzyme of this family with *N*-methyltransferase activity (Supplementary Information). Schematic representations of the BGCs (**a**,**e**), the structure of the precursor peptides (**b**,**f**) and the proposed chemical structures of the natural products (**c**,**g**) are shown.

of '*Ca*. E. malaspinii' draft genomes was corroborated by ultra-low input, long-read metagenomic sequencing of one sample and targeted assembly (Methods) into a single 9.63 Mb linear chromosome with a 75 kb repeat as the only remaining ambiguity.

To establish a phylogenomic context for this species, we searched for closely related species through targeted genome reconstructions in additional eukaryote-enriched metagenomic samples from the *Tara* Oceans expedition[40]. In brief, we aligned metagenomic reads to '*Ca*. E. malaspinii'-related genomic fragments and assumed increased recruitment rates in a given sample to be indicative of the presence of additional relatives (Methods). As a result, we recovered 10 MAGs, and the combined set of 19 MAGs represents five species across three genera within the newly defined family (that is, '*Ca*. Eudoremicrobiaceae'). After manual inspection and quality control (Extended Data Fig. 6 and Supplementary Information), we found that '*Ca*. Eudoremicrobiaceae' spp. representatives have larger genomes (8 Mb) and a

richer biosynthetic potential (ranging from 14 to 22 BGCs per species) compared with members of other '*Ca*. Eremiobacterota' clades (up to 7 BGCs) (Fig. 3a–c).

Exploring the abundance and distribution of '*Ca*. Eudoremicrobiaceae', we found that its members are prevalent in most oceanic basins as well as throughout the water column (Fig. 3d). Locally, they account for up to 6% of ocean microbial communities, making them a numerically substantial component of the global ocean microbiome. Furthermore, we found the relative abundances of '*Ca*. Eudoremicrobiaceae' spp. and their BGCs expression levels to be the highest in eukaryote-enriched fractions (Fig. 3c and Extended Data Fig. 7), suggesting possible interactions with particulate matter, including planktonic organisms. This observation, and the homology of some '*Ca*. Eudoremicrobium' BGCs to known pathways producing cytotoxic natural products could suggest a predatory behaviour (Supplementary Information and Extended Data

Fig. 8), akin to other specialized metabolite-producing predators, such as *Myxococcus*[41]. The detection of '*Ca*. Eudoremicrobiaceae' in less accessible (deep ocean) or eukaryote-enriched, rather than prokaryote-enriched samples, probably explains why these bacteria and their unsuspected BGC diversity had remained unclear in the context of natural-products research.

## New enzymes and natural products

We finally sought to experimentally validate the promising prospects of our microbiomics-driven work for the discovery of new pathways, enzymes and natural products. Among the different BGC classes, RiPP pathways are known to encode a wealth of chemical and functional diversity owing to the various modifications installed post-translationally on a core peptide by maturase enzymes[42]. We therefore selected two '*Ca*. Eudoremicrobium' RiPP BGCs (Fig. 3b and 4a–e) that were predicted to produce novel metabolites on the basis of their dissimilarity to any known BGC ($\bar{d}_{\mathrm{MIBiG}}$ and $\bar{d}_{\mathrm{RefSeq}}$ above 0.2).

The first RiPP pathway ($\bar{d}_{\mathrm{MIBiG}} = 0.41$, $\bar{d}_{\mathrm{RefSeq}} = 0.29$) was found only in the deep-ocean species '*Ca*. E. malaspinii' and encodes a precursor peptide modified by a sole maturase (Fig. 4a,b). In this maturase, we found a single functional domain homologous to the dehydration domain of lanthipeptide synthetases, which normally catalyses phosphorylation and subsequent elimination[43] (Supplementary Information). We therefore predicted the modifications of the precursor peptide to include such a two-step dehydration. However, using tandem mass spectrometry (MS/MS) and nuclear magnetic resonance spectroscopy (NMR), we identified a poly-phosphorylated linear peptide (Fig. 4c). Although this was unexpected, we found several lines of evidence supporting that it is the final product: the absence of dehydration in two different heterologous hosts as well as in vitro assays, the identification of mutated key residues in the dehydration catalytic site of the maturase, which were consistently found in all reconstructed '*Ca*. E. malaspinii' genomes (Extended Data Fig. 9 and Supplementary Information), and finally the bioactivity of the phosphorylated product rather than the chemically synthesized, dehydrated form (Fig. 4d). Indeed, we found that it displayed low-micromolar protease inhibitory activity against neutrophil elastase within a concentration range ($IC_{50} = 14.3\ \mu M$) comparable to other relevant natural products[44], although the ecological role of this unusual natural product remains to be elucidated. On the basis of these results, we propose naming this pathway 'phospeptin'.

The second case represents a complex RiPP pathway specific to '*Ca*. Eudoremicrobium' spp. ($\bar{d}_{\mathrm{MIBiG}} = 0.46$, $\bar{d}_{\mathrm{RefSeq}} = 0.33$) that is predicted to encode a proteusin natural product (Fig. 4e). These pathways are of particular biotechnological interest owing to the expected density and diversity of unusual chemical modifications installed by enzymes encoded in relatively short BGCs[45]. We found that this proteusin differs from previously characterized ones as it lacks both the $NX_5N$ core motif of polytheonamides and the lanthionine rings of landornamides[46]. To overcome the limitations of common heterologous expression models, we used them along a non-standard *Microvirgula aerodenitrificans* system to characterize the four maturase enzymes of the pathway (Methods). Using a combination of MS/MS, isotope labelling and NMR, we found that these maturases install up to 21 modifications, including L- to D-amino acid epimerization, hydroxylation as well as *C*- and backbone amide *N*-methylation, on a 46 amino-acid core peptide (Fig. 4f,g, Extended Data Figs. 10–12 and Supplementary Information). Among the maturases, we characterized the first occurrence of a FkbM *O*-methyltransferase family member[47] in a RiPP pathway and, unexpectedly, found that this maturase introduces backbone *N*-methylations (Fig. 4h,i and Supplementary Information). Although this modification is known in NRP natural products[48], enzymatic *N*-methylation of the amide bond is a challenging yet biotechnologically relevant reaction[49] that has to date been specific to the borosin RiPP family[50,51]. The identification of this

activity in other enzyme and RiPP families may provide opportunities for new applications and expand the functional diversity[52] of proteusins along with their chemical diversity. On the basis of the identified modifications and the unusual length of the proposed structure for the product, we suggest naming the pathway 'pythonamide'.

The finding of unexpected enzymology within a functionally characterized enzyme family[47] illustrates both the promise of environmental genomics for new discoveries and, at the same time, the limited power of functional inferences made solely by sequence homology. Thus, together with the report of a non-canonical bioactive poly-phosphorylated RiPP, our results demonstrate the resource-intensive, yet critical value of synthetic biology efforts to fully uncover the functional richness, diversity and unusual architectures of biochemical compounds.

## Conclusions

Here we have demonstrated the extent of microbially encoded biosynthetic potential and its genomic context in the global ocean microbiome, facilitating future research by making the generated resources available to the scientific community (https://microbiomics.io/ocean/). We found that the majority of both its phylogenomic and functional novelty is accessible only through the reconstruction of MAGs and SAGs, particularly in underexplored microbial communities, which could direct future bioprospecting efforts. Although we focused here on '*Ca*. Eudoremicrobiaceae' as a particularly biosynthetically 'talented' lineage, many of the predicted BGCs within other uncovered microbial groups are likely to encode previously undescribed enzymology that produces compounds with ecologically and/or biotechnologically relevant activities.

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

## Methods

### Metagenomic data selection, assembly and binning
Metagenomic datasets from major oceanographical surveys and time-series studies with sufficient sequencing depth were included to maximize the coverage of global ocean microbial communities across ocean basins, depth layers and time. These datasets (Supplementary Table 1 and Fig. 1) included metagenomes from samples collected by *Tara* Oceans (virus-enriched, $n = 190$; and prokaryote-enriched, $n = 180$)[12,22], BioGEOTRACES expeditions ($n = 480$), the Hawaiian Ocean Time-series (HOT, $n = 68$), the Bermuda-Atlantic Time-series Study (BATS, $n = 62$)[21] and the Malaspina expedition ($n = 58$)[23]. Sequencing reads from all metagenomes were quality filtered using BBMap (v.38.71) by removing sequencing adapters from the reads, removing reads that mapped to quality control sequences (PhiX genome) and discarding low quality reads using the parameters trimq = 14, maq = 20, maxns = 0 and minlength = 45. Downstream analyses were performed on quality-controlled reads or if specified, merged quality-controlled reads (bbmerge.sh minoverlap = 16). Quality-controlled reads were normalized (bbnorm.sh target = 40, mindepth = 0) before they were assembled with metaSPAdes (v.3.11.1 or v.3.12 if required)[53]. The resulting scaffolded contigs (hereafter scaffolds) were finally filtered by length (≥1 kb).

The 1,038 metagenomic samples were grouped into several sets and, for each sample set, the quality-controlled metagenomic reads from all samples were individually mapped against the scaffolds of each sample, resulting in the following numbers of pairwise readset to scaffold mappings: *Tara* Oceans virus-enriched (190 × 190), prokaryote-enriched (180 × 180), BioGEOTRACES, HOT and BATS (610 × 610) and Malaspina (58 × 58). Mapping was performed using the Burrows–Wheeler-Aligner (BWA) (v.0.7.17-r1188)[54], allowing the reads to map at secondary sites (with the -a flag). Alignments were filtered to be at least 45 bases in length, with an identity of ≥97% and covering ≥80% of the read sequence. The resulting BAM files were processed using the jgi_summarize_bam_contig_depths script of MetaBAT2 (v.2.12.1)[55] to provide within- and between-sample coverages for each scaffold. The scaffolds were finally binned by running MetaBAT2 on all samples individually with parameters --minContig 2000 and --maxEdges 500 for increased sensitivity. We used MetaBAT2 in lieu of ensemble binning approaches as it was shown to be the best performing single binner in independent benchmarks[56] and was found to be 10 to 50 times faster than other usual binners[57]. To test for the effect of abundance correlation, a randomly selected subsample of the metagenomes (10 from each of the two *Tara* Ocean datasets, 10 from BioGEOTRACES, five from each time-series and five from Malaspina) was additionally binned using only within-sample coverage information (Supplementary Information).

### Selection of additional genomes
Additional (external) genomes were included in downstream analyses, namely 830 manually curated MAGs from a subset of the *Tara* Oceans dataset[26], 5,287 SAGs from the GORG dataset[20], as well as 1,707 isolate REFs and 682 SAGs from the MAR databases (MarDB v.4)[27]. For the MarDB dataset, genomes were selected on the basis of the available metadata if the sample type matched the following regular expression: '[S|s]ingle.?[C|c]ell|[C|c]ulture|[I|i]solate'.

### Quality evaluation of metagenomic bins and external genomes
The quality of each metagenomic bin and external genome was evaluated using both the 'lineage workflow' of CheckM (v.1.0.13) and Anvi'o (v.5.5.0)[58,59]. Metagenomic bins and external genomes were retained for downstream analyses if either CheckM or Anvi'o reported a completeness/completion of ≥50% and a contamination/redundancy of ≤10%. These metrics were then aggregated into a mean completeness (mcpl) and a mean contamination (mctn) value to classify genome quality according to community standards[60] as follows: high quality: mcpl ≥ 90% and mctn ≤ 5%; good quality: mcpl ≥ 70% and mctn ≤ 10%; medium quality: mcpl ≥ 50% and mctn ≤ 10%; fair quality: mcpl ≤ 90% or mctn ≥ 10%. Filtered genomes were further attributed quality scores ($Q$ and $Q'$) as follows: $Q = \text{mcpl} - 5 \times \text{mctn}$; and $Q' = \text{mcpl} - 5 \times \text{mctn} + \text{mctn} \times (\text{strain heterogeneity})/100 + 0.5 \times \log[N_{50}]$ (as implemented in dRep[61]).

### Species-level clustering of the genome collection and comparison to other resources
To allow for comparative analyses between different data resources and genome types (MAGs, SAGs and REFs), the full set of 34,799 genomes was dereplicated on the basis of both whole-genome average nucleotide identity (ANI) using dRep (v.2.5.4)[61] with a 95% ANI threshold[28,62] (-comp 0 -con 1000 -sa 0.95 -nc 0.2) and single-copy marker genes using SpecI[63], providing species-level clustering of the genomes. A representative genome was selected based on the maximum quality score defined above ($Q'$) for each of the dRep clusters, which were considered to be a proxy for species membership.

To estimate mapping rates, all 1,038 metagenomic readsets were mapped against the 34,799 genomes included in the OMD using BWA (v.0.7.17-r1188, -a). Quality-controlled reads were mapped in single-end mode and the resulting alignments were filtered to keep only alignments of ≥45 bp in length and with an of identity ≥ 95%. The per-sample mapping rate is the percentage of reads that remained after filtering divided by the total number of quality-controlled reads. Using the same method, each of the 1,038 metagenomes was downsampled to 5 million inserts (Extended Data Fig. 1c) and mapped to the GORG SAGs within the OMD and all of the GEM[16]. The number of MAGs in the GEM catalogue[16] that were recovered from ocean waters was determined on the basis of a keyword query on the source of metagenomes, selecting for ocean water samples (as opposed to marine sediments, for example). Specifically, we selected 'aquatic' as the 'ecosystem_category', 'marine' as the 'ecosystem_type' and filtered 'habitat' for 'deep ocean', 'marine', 'marine oceanic', 'pelagic marine', 'seawater', 'marine', 'seawater', 'surface seawater', 'surface seawater'. This resulted in 5,903 MAGs (734 high quality), distributed across 1,823 OTUs (here, species).

### Taxonomic and functional genome annotation
Prokaryotic genomes were taxonomically annotated using GTDB-Tk (v.1.0.2)[64] with the default parameters against the GTDB r89 release[13]. Anvi'o was used to identify eukaryotic genomes on the basis of domain prediction and completion of ≥50% and redundancy of ≤10%. The taxonomic annotation of a species is defined as the one of its representative genome. Excluding eukaryotes (148 MAGs), each genome was functionally annotated by first calling complete genes using prokka (v.1.14.5)[65] with the 'Archaea' or 'Bacteria' parameter specified as appropriate, which also reported non-coding genes and CRISPR regions, among other genomic features. The predicted genes were annotated by identifying universal single-copy marker genes (uscMGs) with fetchMGs (v.1.2)[66], assigning orthologous groups with emapper (v.2.0.1)[67] based on eggNOG (v.5.0)[68] and performing queries against the KEGG database (release 2020-02-10)[69]. This last step was performed by aligning the proteins to the KEGG database using DIAMOND (v.0.9.30)[70] with a query and subject coverage of ≥70%. The results were further filtered on the basis of the bitscore being ≥50% of the maximum expected bitscore (reference against itself) according to the NCBI Prokaryotic Genome Annotation Pipeline[71]. The gene sequences were additionally used as input to identify BGCs in the genomes using antiSMASH (v.5.1.0)[72] with the default parameters and the different cluster blasts turned on. All genomes and annotations have been compiled along with contextual metadata into the OMD, which is available online (https://microbiomics.io/ocean/).

### Gene-level profiling
Similar to the methods described previously[12,22], we clustered the >56.6 million protein-coding genes from the bacterial and archaeal

genomes of the OMD at 95% identity and 90% coverage of the shorter gene using CD-HIT (v.4.8.1)[73] into >17.7 million gene clusters. The longest sequence was selected as the representative gene of each gene cluster. The 1,038 metagenomes were then mapped to the >17.7 million cluster representatives with BWA (-a) and the resulting BAM files were filtered to retain only alignments with a percentage identity of ≥95% and ≥45 bases aligned. Length-normalized gene abundance was calculated by first counting inserts from best unique alignments and then, for ambiguously mapped inserts, adding fractional counts to the respective target genes in proportion to their unique insert abundances.

## Species-level profiling with mOTUs

The genomes in the extended OMD (augmented with additional MAGs from '*Ca*. Eudoremicrobiaceae', see below) were added to the database (v.2.5.1) of the metagenomic profiling tool mOTUs[74] to generate an extended mOTUs reference database. Only genomes with at least six out of the ten uscMGs in single copy were kept (23,528 genomes). The extension of the database resulted in 4,494 additional species-level clusters. Profiling of the 1,038 metagenomes was performed using the default parameters of mOTUs (v.2). On the basis of the mOTU profiles, a total of 989 genomes (95% REFs, 5% SAGs and 99.9% belonging to MarDB) contained within 644 mOTU clusters were not detected. This reflects the various additional marine isolation sources (most of the genomes not detected are associated with organisms isolated from, for example, sediments, marine hosts) of the MarDB genomes. To remain focused on the open ocean environment in this study, we excluded them from downstream analyses if they were not detected or not included in the extended mOTU database established in this study.

## Clustering and selection of BGCs

All BGCs from MAGs, SAGs and REFs in the OMD (see above) were combined with the ones identified across all the metagenomic scaffolds (antiSMASH v.5.0, default parameters) and processed with BiG-SLICE (v.1.1) for feature (PFAM domains) extraction[75]. On the basis of these features, we computed all-versus-all cosine distances between BGCs and clustered them (average linkage) into GCFs and GCCs, using a 0.2 and a 0.8 distance threshold, respectively. These thresholds are an adaptation of those previously used with Euclidean distances[75] to cosine distances, which alleviate some of the biases of the original BiG-SLICE clustering strategy (Supplementary Information).

BGCs were subsequently filtered, retaining only the ones encoded on scaffolds ≥5 kb to reduce the risk of fragmentation, as done previously[16], and excluding MarDB REFs and SAGs that were not detected in the 1,038 metagenomes (see above). This resulted in a total of 39,055 BGCs encoded by OMD genomes and an additional 14,106 identified on metagenomic fragments (that is, that were not binned into MAGs). These 'metagenomic' BGCs were used to estimate the proportion of the ocean microbiome biosynthetic potential not captured by the database (Supplementary Information). Each BGC was functionally characterized on the basis of predicted product types as defined by antiSMASH or coarser product classes, as defined in BiG-SCAPE[76]. To prevent sampling biases in quantitative analyses (taxonomic and functional compositions of GCCs/GCFs, GCF and GCC distances to reference databases as well as GCF metagenomic abundances), the 39,055 BGCs were further dereplicated by retaining only the longest BGC per GCF per species, leading to a total of 17,689 BGCs.

## Novelty of GCFs and GCCs

The novelty of GCCs and GCFs was estimated on the basis of distances to databases of computationally predicted (the RefSeq database within BiG-FAM)[29] and experimentally validated (MIBIG 2.0)[30] BGCs. For each of the 17,689 representative BGCs, we selected the minimum cosine distance to the respective database. These minimum distances were then averaged (mean) per GCF or GCC as appropriate. A GCF was considered to be novel if the distance to the database was above 0.2, which

corresponds to (on average) the complete separation between the GCF and the reference. For GCCs, we selected 0.4, twice the GCF-defining threshold, to capture remote relationships with the reference.

## Abundance and prevalence of GCFs and GCCs

The metagenomic abundance of a BGC was estimated as the median abundance of its biosynthetic genes (as defined by antiSMASH), which were available from the gene-level profiles. The metagenomic abundance of each GCF or GCC was subsequently computed as the sum of its representative BGCs (out of the 17,689). These abundance profiles were subsequently cell-normalized using the mOTU count per sample, which also accounts for the sequencing effort[22,74] (Extended Data Fig. 1d). The prevalence of a GCF or GCC was computed as the percentage of samples with an abundance of >0.

## Structure of the ocean microbiome biosynthetic potential

Euclidean distances between samples were computed on the basis of the normalized GCF profiles. These distances were dimensionally reduced using UMAP[77] and the resulting embedding was used for unsupervised density-based clustering with HDBSCAN[78]. The optimal minimum number of points of a cluster (and therefore the number of clusters) used by HDBSCAN was determined by maximizing the cumulative cluster membership probability. The identified clusters (as well as random balanced subsamples of these clusters, to account for biases in permutational multivariate analysis of variance (PERMANOVA)) were tested for significance using PERMANOVA against the non-reduced Euclidean distances. The average genome size of a sample was computed on the basis of relative abundances of mOTUs and the estimated genome sizes of the member genomes. Specifically, the mean genome size for each mOTU was estimated as the mean of the completeness-corrected sizes (for example, the corrected size of a 75% complete genome with a length of 3 Mb is 4 Mb) of its member genomes (after filtering for genomes with a mean completeness of ≥70%). Then, per sample, the average genome size was computed as the sum of the relative abundance-weighted mOTU genome sizes.

## Phylogenomic distribution of BGCs

The filtered set of BGCs encoded by genomes in the OMD (in scaffolds ≥5 kb and excluding MarDB REFs and SAGs that were not detected in the 1,038 metagenomes, see above) along with their predicted product classes were displayed on the GTDB bacterial and archaeal trees on the basis of the GTDBTk phylogenomic placement of the genomes (see above). We first reduced the data on a per-species basis, using the genome with most BGCs in that species as a representative. For visualization, the representatives were further binned along the tree and, similarly, for each binned clade, the genome containing the most BGCs was selected as representative. BGC-rich species (at least one genome with >15 BGCs) were further analysed by computing the Shannon diversity index of the product types encoded in these BGCs. Chemical hybrids and other complex BGCs (as predicted by antiSMAH) were considered to be from the same product type if all of the predicted product types were identical, irrespective of their order within the cluster (for example, a proteusin–bacteriocin hybrid is identical to bacteriocin–proteusin hybrid).

## Long-read sequencing of '*Ca*. Eudoremicrobium'

Leftover DNA (an estimated 6 ng) from the sample Malaspina MP1648, corresponding to the biosample SAMN05421555 and matching the short-read Illumina metagenomic readset SRR3962772, was processed for an ultralow input PacBio sequencing protocol to produce a >20 Gb Hifi Pacbio metagenome using the PacBio kits SMRTbell gDNA Sample amplification kit (100-980-000) and the SMRTbell Express Template Prep kit 2.0 (100-938-900). In brief, the remaining DNA was sheared using a Covaris (g-TUBE, 52104), repaired and purified (ProNex beads). The purified DNA was then library prepped, amplified, purified (ProNex

beads) and size-selected (>6 kb, Blue Pippin) before a final purification step (ProNex beads) and sequencing on the Sequel II platform.

## Targeted binning of '*Ca*. Eremiobacterota'

After the reconstruction of the first two '*Ca*. Eremiobacterota' MAGs, we identified six additional ones with ANI > 99% (these are included in Fig. 3) that were initially filtered out on the basis of contamination estimates (later identified as gene duplications, see below). We additionally recovered bins identified as '*Ca*. Eremiobacterota' from a different study[23] and used them along with the eight MAGs from our study as a reference for subsampled mapping (5 million reads) of metagenomic reads from 633 eukaryote-enriched (>0.8 µm) samples using BWA (v.0.7.17-r1188, -a flag). On the basis of enriched specific mappings (after 95% alignment identity and 80% read coverage filtering), 10 metagenomes (expected coverage, ≥5×) were selected for assembly and 49 additional metagenomes (expected coverage, ≥1×) for abundance correlation. Using the same parameters as described above, these samples were binned and 10 additional '*Ca*. Eremiobacterota' MAGs were recovered. These 16 MAGs (which excludes the two that were already in the database) bring the total number of genomes in the extended OMD to 34,815. The MAGs were assigned to taxonomic ranks on the basis of their genomic similarity and GTDB placement. The 18 MAGs were dereplicated using dRep into 5 species (within-species ANIs were >99%) and 3 genera (within-genus ANIs ranged between 85% and 94%)[79] within the same family. Species representatives were manually selected on the basis of completeness, contamination and $N_{50}$. Proposed naming is available in Supplementary Information.

## Manual evaluation of '*Ca*. Eremiobacterota' MAGs

To evaluate the completeness and contamination of '*Ca*. Eremiobacterota' MAGs, we assessed the presence of uscMGs, in addition to lineage- and domain-specific single-copy marker gene sets used by CheckM and Anvi'o. The identification of duplications among 2 out of the 40 uscMGs was confirmed by phylogenetic reconstruction (see below) to rule out any potential contamination (which would have corresponded to 5% on the basis of these 40 marker genes). Additional inspection of the representative MAGs of the five '*Ca*. Eremiobacterota' species confirmed low rates of contaminants in these reconstructed genomes on the basis of abundance correlation and sequence composition (Supplementary Information) using the Anvi'o interactive interface[59].

## Phylogenomics of '*Ca*. Eudoremicrobiaceae'

For phylogenomic analyses, we selected the representative MAGs of the five '*Ca*. Eudoremicrobiaceae' species, all '*Ca*. Eremiobacterota' genomes available in GTDB (r89)[13] and representatives of additional phyla (including UBP13, Armatimonadota, Patescibacteria, Dormibacterota, Chloroflexota, Cyanobacteria, Actinobacteria and Planctomycetota). All of these genomes were annotated as previously described to extract single-copy marker genes and to annotate BGCs. GTDB genomes were retained on the basis of the completeness and contamination criteria mentioned above. The phylogenomic analysis was performed using the Anvi'o phylogenomics workflow[59]. The tree was constructed with IQTREE (v.2.0.3) (default parameters and -bb 1000)[80] on an alignment (MUSCLE, v.3.8.1551)[81] of 39 concatenated ribosomal proteins identified by Anvi'o, with positions trimmed for coverage in at least 50% of the genomes[82] and using Planctomycecota as the outgroup based on the GTDB tree topology. Individual trees for the 40 uscMGs were built using the same tools and parameters.

## Trait and lifestyle prediction of '*Ca*. Eudoremicrobiaceae'

We used Traitar (v.1.1.2) with the default parameters (phenotype, from nucleotides)[83] to predict general microbial traits. We investigated the potential predatory lifestyle on the basis of a previously developed predatory index[84], which relies on the protein-coding gene content of a genome. Specifically, we used DIAMOND to compare the proteins

from a genome to the OrthoMCL database (v.4)[85] using the parameters --more-sensitive --id 25 --query-cover 70 --subject-cover 70 --top 20 and counted genes that matched predatory and non-predatory marker genes. The index is the difference between the number of predatory and non-predatory markers. As an additional control, we also analysed the genome of '*Ca*. Entotheonella' factor TSY1[18] on the basis of its similar characteristics to '*Ca*. Eudoremicrobium' (large genome size and biosynthetic potential). We further tested a potential link between predatory and non-predatory marker genes with the biosynthetic potential of '*Ca*. Eudoremicrobiaceae' and found at most one gene (from either type, that is, predatory/non-predatory, of marker genes) overlapping with BGCs, suggesting that BGCs do not confound the predatory signal. Additional annotations of the genomes to specifically investigate secretion systems, pili and flagella were performed using TXSSCAN (v.1.0.2) for unordered replicons[86].

## Transcriptomic profiling of '*Ca*. E. taraoceanii'

Transcriptomic profiling was performed by mapping 623 metatranscriptomes from *Tara* Oceans prokaryote- and eukaryote-enriched fractions[22,40,87] (using BWA, v.0.7.17-r1188, -a flag) to the five representative '*Ca*. Eudoremicrobiaceae' genomes. After 80% read coverage and 95% identity filtering, the BAM files were processed using FeatureCounts (v.2.0.1)[88] (using the parameters featureCounts --primary -O --fraction -t CDS,tRNA -F GTF -g ID -p) to compute the number of inserts per gene. The resulting profiles were normalized to gene length and mOTU marker gene abundances (median length-normalized insert count of genes with insert count of >0) and log$_2$-transformed[22,74] to obtain relative per-cell expression levels of each gene, which also accounts for between-samples differences in sequencing effort. Such ratios allow for comparative analyses by mitigating the issues of compositionality when working with relative abundance data. Only samples with >5 out of the 10 mOTU marker genes were considered for further analyses, ensuring that a large enough fraction of the genome was detected.

The normalized transcriptomic profiles of '*Ca*. E. taraoceanii' were dimension-reduced using UMAP and the resulting representation was used for unsupervised clustering using HDBSCAN (see above) to identify expression states. The significance of differences between the identified clusters was tested by PERMANOVA in the original (non-reduced) distance space. Differential expression between these states was tested across 201 KEGG pathways identified in the genome (see above) and 6 functional groups, namely, BGCs, secretion systems and flagellar genes from TXSSCAN, degradative enzymes (proteases and peptidases) from prokka and predatory and non-predatory markers from the predatory index. For each sample, we computed the median normalized expression for each category (note that BGC expression was itself computed as the median expression of the biosynthetic genes of that BGC) and tested for significance (FDR-corrected Kruskal–Wallis test) across the different states.

## Experimental validation of a novel phosphorylated RiPP pathway ('*Ca*. E. malaspinii', HLLJDLBE BGC 75.1)

**Materials for heterologous expression.** Synthetic genes were purchased from GenScript and PCR primers were ordered from Microsynth. Phusion polymerase from Thermo Fisher Scientific was used for DNA amplification. NucleoSpin plasmid and NucleoSpin Gel and PCR Clean-up kits from Macherey–Nagel were used to purify DNA. Restriction enzymes and T4 DNA ligase were purchased from New England Biolabs. Chemicals were purchased from Sigma-Aldrich, with the exception of isopropyl-β-D-1-thiogalactopyranoside (IPTG) (Biosynth) and 1,4-dithiothreitol (DTT, AppliChem) and used without further purification. The antibiotics chloramphenicol (Cm), spectinomycin dihydrochloride (Sm), ampicillin (Amp), gentamicin (Gt) and carbenicillin (Cbn) were purchased from AppliChem. The medium components Bacto Tryptone and Bacto Yeast Extract were purchased from BD Biosciences. Sequencing-grade trypsin was purchased from Promega.

**Cloning of *embA*, *embM* and *orf3* (*embI*) for protein expression.** Gene sequences were extracted from BGC 75.1 predicted by antiSMASH on the type material of '*Ca*. E. malaspinii' (Supplementary Information).

The genes *embA* (locus, MALA_SAMN05422137_METAG-scaffold_127-gene_5), *embM* (locus, MALA_SAMN05422137_METAG-scaffold_127-gene_4) and *embAM* (including intergenic region) were ordered as synthetic constructs in pUC57(Amp$^R$), with and without codon-optimization for expression in *Escherichia coli*. The gene *embA* was subcloned into the first multiple cloning site (MCS1) of pACYCDuet-1(Cm$^R$) and pCDFDuet-1(Sm$^R$) with BamHI and HindIII cut sites. The genes *embM* and *embMopt* (codon optimized) were subcloned into MCS1 of pCDFDuet-1(Sm$^R$) with BamHI and HindIII and in the second multiple cloning site (MCS2) of pCDFDuet-1(Sm$^R$) and of pRSFDuet-1(Kan$^R$) with NdeI/XhoI. The *embAM* cassette was subcloned into pCDFDuet1(Sm$^R$) with the BamHI and HindIII cut sites. The gene *orf3/embI* (locus, MALA_SAMN05422137_METAG-scaffold_127-gene_3) was constructed by overlap extension PCR with primers EmbI_OE_F_NdeI and EmbI_OE_R_XhoI, digested with NdeI/XhoI and ligated to pCDFDuet-1-EmbM(MCS1), which was digested using the same restriction enzymes (Supplementary Table 6). Restriction enzyme digestions and ligations were performed according to the manufacturer's (New England Biolabs) procedures.

All constructs generated above were introduced into chemically competent *E. coli* DH5α and plated onto LB agar with appropriate antibiotic selection. Plasmids were purified from single colonies and sequenced using sequencing primers to verify proper insertion of genes (Supplementary Table 6). The genes *embA* and *embAM* were additionally subcloned in a modified pLMB509m(Gt$^R$) vector for *M. aerodenitrificans* expression through Gibson assembly, with the inclusion of an N-terminal His$_6$ purification tag in the final EmbA protein product[89]. A list of the Gibson assembly primers EmbA_F_plmb, EmbA_R_plmb, Plmb_F_EmbA and Plmb_R_EmbA is provided in Supplementary Table 6. Transformation of *E. coli* DH5α, isolation of plasmids and validation of the correct clones by sequencing was followed by introduction of NHis$_6$-EmbA-pLMB509m and NHis$_6$-EmbAM-pLMB509m into *E. coli* SM10 for conjugation.

**Heterologous expression and purification for protein isolation for in vitro assays.** Chemically competent *E. coli* BL21(DE3) was introduced into pCDFDuet-1-EmbA(MCS1) and pCDFDuet-1-EmbM(MCS1). The same conditions were used for expression of both N-terminally His$_6$-tagged proteins. Overnight cultures were prepared from single colonies and used to inoculate (1%, v/v) TB medium (2 × 200 ml) in 1 l baffled Erlenmeyer flasks supplemented with spectinomycin (50 mg ml$^{-1}$). Cells were grown at 37 °C, 200 rpm, until an optical density at 600 nm (OD$_{600}$) of around 1.0, cooled on an ice bath for 20 min and induced with a final concentration of 0.5 mM IPTG. The cultures were further incubated at 16 °C, 180 rpm for 18–20 h, and subsequently collected by centrifugation (8,000g for 20 min) and frozen.

Purifications of NHis$_6$-EmbA and NHis$_6$-EmbM were carried out at 4 °C, using the same procedure for both proteins. Cell pellets were resuspended in 5 ml g$^{-1}$ of lysis buffer (50 mM Tris, 300 mM NaCl, 5 mM imidazole, 10% glycerol, pH 7.8). The suspension was supplemented with lysozyme (1 mg ml$^{-1}$), DNase I (10 U ml$^{-1}$) and protease inhibitor cocktail (0.2%, v/v) and stirred at 37 °C for 30 min. After cooling the suspension for 15 min on an ice bath, cells were lysed by sonication (30% amplitude, 10s on/off cycles, for a total of 3 min), and the clarified lysate was obtained by centrifugation (27,000g for 30 min). The supernatant was loaded onto 4 ml of Ni-NTA agarose resin that had been equilibrated with a lysis buffer in a fritted purification column. The resin was washed with 10 column volumes (CV) of lysis buffer, 3 CV of wash buffer (50 mM Tris, 300 mM NaCl, 40 mM imidazole, 10% glycerol, pH 7.8) and finally eluted with 3 CV of elution buffer (50 mM Tris, 300 mM NaCl, 250 mM imidazole, 10% glycerol, pH 7.8) in 1.5 ml fractions. Elution fractions were analysed by SDS–PAGE, pooled and concentrated in spin concentrators with the appropriate molecular weight cut-off. NHis$_6$-EmbA and NHis$_6$-EmbM were buffer-exchanged using a PD MiniTrap G25 column pre-equilibrated with G25 buffer (50 mM Tris, 300 mM NaCl, 10% glycerol, pH 8.0). The concentration of buffer-exchanged proteins was determined by measuring the absorbance of purified proteins at 280 nm and using the calculated values for molecular mass and extinction coefficient for each protein.

**In vitro enzymatic activity assays with EmbA and EmbM.** Extensive screening of enzymatic reaction parameters, including temperature, time, enzyme and substrate concentration, buffer pH and salinity resulted in the following best condition set for EmbM turnover: EmbA was added to a glass vial to a final concentration of 200 mM. Final concentrations of 2 mM of MgCl$_2$ and 2 mM of adenosine 5′-triphosphate (ATP) were added to the reactions. EmbM was not added to control reactions and added to a final concentration of 10 mM in turnover experiments. The enzymatic reaction was stirred at 37 °C for 72 h, and the reaction mixture was supplemented with 2 mM of ATP every 24 h. Reaction scales ranged from 100 μl, for analytical purposes, to 3 ml for product isolation.

Modified EmbA was proteolysed with trypsin for MS analysis and with LahT150 for MS analysis and product isolation[90]. Trypsin cleavage was performed by diluting the reaction mixture with 2× trypsin buffer (50 mM Tris, 5 mM CaCl$_2$, pH 8.0), adding 1:20 trypsin:EmbA and incubating overnight at 37 °C. LahT150 cleavage was performed by adding LahT150 at a 1:10 ratio to EmbA, and incubating at room temperature overnight for small-scale reactions, or for 24 h, for reactions larger than 1 ml.

Large-scale enzymatic reactions were purified using solid-phase extraction (SPE) with a Phenomenex Strata C18-E reverse phase column (2 g sorbent). The sorbent was first washed with 24 ml MeOH and equilibrated with 24 ml H$_2$O (+0.1% formic acid). The proteolysis reactions were loaded onto the sorbent, which was then washed with 24 ml H$_2$O (+0.1% formic acid). The peptide products were eluted with 24 ml of 1:1 MeCN:H$_2$O (+0.1% formic acid) and 12 ml MeCN (+0.1% formic acid). Elution fractions were pooled, dried on a Genevac concentrator and stored at −20 °C.

**Co-expression of EmbA, EmbM and Orf3 (EmbI) in heterologous hosts and purification.** A wide variety of *E. coli* co-expression conditions of EmbA, EmbM and Orf3 were screened (Supplementary Table 6). In general, chemically competent BL21(DE3) or Tuner(DE3) were transformed with different combinations of the constructs described above, and selected with appropriate antibiotics on LB plates. Overnight cultures were prepared from single colonies and 1% (v/v) of culture was used to inoculate 200 ml of medium (TB, LB, XPPM)[91] supplemented with appropriate antibiotic selection in 1 l baffled Erlenmeyer flasks. Cells were grown at 37 °C, 200 rpm until an OD$_{600}$ of around 1.2. For low-temperature growths, cultures were subsequently cooled in ice baths for 20 min, induced with 0.5 mM IPTG and incubated at 16 °C at 180 rpm. Incubation times varied from 24 h to 7 days. High temperature growths were induced with a final concentration of 0.5 mM IPTG without cooling and incubated at 37 °C, 200 rpm from 6 h to 16 h. Cultures were collected by centrifugation (8,000g for 20 min), frozen and purified as described above. Proteolysis reactions with trypsin or LahT150 for MS analysis were performed as described in the previous section.

Transformation of *M. aerodenitrificans* DSMZ 15089 with NHis$_6$-EmbA-pLMB509m and NHis$_6$-EmbAM-pLMB509m was achieved according to a published procedure[89]. Culturing conditions followed an adaptation of a previously described method[89]. Cultures were collected, stored and purified as described above. Trypsin digestion was used for MS analysis.

**β-Elimination of phosphorylated peptides.** Phosphorylated peptide intermediates obtained by co-expression of EmbA and EmbM

were submitted to β-elimination conditions at the 100 μl scale. EmbA (200 μM) in G25 buffer was used either as an intact protein or as a trypsin-digest product. The pH of the solution was adjusted to pH 13 with 1 M NaOH, and the elimination reaction proceeded at 37 °C for 4 h to afford dehydrobutyrine (Dhb)-containing products. Derivatization of Dhb was performed by adding a final concentration of 50 mM DTT to the reaction. The pH of the solution was adjusted to 7 with HCl (aq.) before MS analysis.

β-Elimination of phosphorylated peptide intermediates obtained through in vitro enzymatic reactions was performed using EmbA that had been proteolysed with LahT150 and purified using SPE as described above. EmbA (0.6 mmol) was resuspended in 3 ml of $H_2O$, and the pH of the solution was adjusted to 14 with 1 M NaOH. The reaction stirred at 37 °C for 48 h, and was subsequently neutralized to pH 7 with HCl (aq.). SPE purification was performed as described above.

**HPLC–HR-MS and MS/MS analysis.** HPLC–HR-MS and MS/MS analyses were performed on a Thermo Scientific Dionex UltiMate 3000 UHPLC coupled to a Thermo Scientific Q Exactive mass spectrometer using heated electrospray ionization in positive ion mode with a Phenomenex Kinetex 2.6 μm XB-C18 100 Å (150 × 4.6 mm) column. The column temperature was set to 50 °C and the flow rate to 0.5 ml min$^{-1}$. Samples were centrifuged before injections and target peptides were eluted with a gradient of 15–55% MeCN (+0.1% formic acid) over 15 min. Full MS was performed at a resolution of 70,000 (AGC target, $1 \times 10^6$; maximum IT, 100 ms) and parallel reaction monitoring was performed at a resolution of 17,500 (AGC target, $2 \times 10^5$; maximum IT, 100 ms; isolation windows in the range of 2.0 $m/z$).

**NMR analysis.** 2D [$^{13}$C,$^1$H] HSQC spectra with multiplicity editing were recorded at natural $^{13}$C abundance on ~4 mM solutions of full length EmbA in unmodified and modified form. Spectra were recorded at 25 °C on a Bruker AVNEO 600 MHz spectrometer equipped with a TCI CryoProbeTM. The following spectral parameters were used: 2,048 complex points at a spectral width of 16 ppm, centred at 4.7 ppm in the direct $^1$H dimension and 512 complex points at a spectral width of 80 ppm centred at 42 ppm in the indirect $^{13}$C dimension. A number of scans of 40 was used, which resulted in a measurement time of 24 h per spectrum.

**Antibiotic activity assays.** *E. coli* DSM 1103, *Staphylococcus aureus* ssp. *aureus* ATCC 29213, *Pseudomonas aeruginosa* DSM 1117, *Acinetobacter baumanii* DSM 30007, *Enterococcus faecalis* DSM 2570, *Rhodococcus* sp. L233, *Aquimarina* sp. Aq135, *Rheinheimera aquimaris* B26, *Vibrio spartinae* (salt marsh isolate), *Pseudoalteromonas rubra* DSM 6842, *Saccharomyces cerevisiae* W301-1A and *Pichia pastoris* (*Komagataella phaffii*) NRR Y-11430 were tested for antimicrobial activity with the dehydrated peptide from BGC 75.1. Bioactivity assays were carried out in accordance with the 2003 guidelines of the Clinical and Laboratory Standards Institute (CLSI) using the microtitre method.

*E. coli, S. aureus, A. baumanii*, and *P. aeruginosa* overnight cultures were grown in LB at 37 °C. *P. rubra, R. aquimaris* (37 °C), *V. spartinae* (30 °C), *P. rubra* and *Aquimarina* sp. Aq135 (24 °C) were cultured in marine broth at their respective growth temperature optima indicated in parentheses. *Rhodococcus* sp. L233 was cultured in R2A medium at 24 °C. *S. cerevisiae* and *P. pastoris* were cultured in YPD medium at 28 °C. An overview of strains, growth conditions and taxonomy is provided in Supplementary Table 6.

Microbial seed cultures were initiated by inoculating 5 ml of medium for each strain and by incubating overnight shaking at 200 rpm. Each culture was then diluted with their respective growth medium to an initial OD$_{600}$ of 0.02 in 80 μl volume per well in sterile 96-well plates (one per strain tested). Assays were set up in duplicates, with appropriate controls including solvent (water) controls and positive controls consisting of two different broad-spectrum antibiotics (chloramphenicol

and ampicillin) with final concentrations of 50 μM. Cycloheximide and benomyl were used as positive controls for *S. cerevisiae* and *P. pastoris*.

Two final concentrations of the peptide, using water as a solvent, were tested: 50 μM and 25 μM. Then, 96-well plates were parafilmed and incubated at room temperature without shaking. OD$_{600}$ was determined after 10 s of plate agitation at the following time points: 2 h, 4 h, 6 h, 8 h, 18 h, 21 h, 24 h and 48 h.

**Protease inhibition assays.** Inhibition assays against neutrophil elastase and cathepsin B were performed using the Neutrophil Elastase Inhibitor Screening Kit (MAK213, Sigma-Aldrich) and the Cathepsin B Inhibitor Screening Kit (MAK200, Sigma-Aldrich). Assays were performed according to the manufacturer's protocol. A microplate reader spectrofluorometer (Tecan Infinite M200 Pro) was used to measure fluorescence and data were processed in Prism 9 to calculate IC$_{50}$ values.

Inhibition assays with trypsin and chymotrypsin were set up in 96-well microtitre plates (black, clear bottom). To each well, 25 ml of stock solutions of different concentrations of peptides in 50 mM Tris, pH 8 buffer (assay buffer); 2 ml of enzyme stock solution (chymotrypsin, V1062, Promega, 1 nM final concentration; trypsin, V5111, Promega, 3 nM final concentration); and 48 ml of protease buffer (40 mM Tris, 10 mM CaCl$_2$, pH 8) were added. Phenylmethylsulfonyl fluoride was used as a positive control. The plates were incubated at room temperature for 10 min. Subsequently, 23 ml of assay buffer and 2 ml of a 500 mM solution of substrate (chymotrypsin: Suc-Ala-Ala-Pro-Phe-AMC, 3114-v, Peptanova; trypsin: Boc-Ile-Glu-Gly-Arg-AMC, 3094-v, Peptanova) in DMSO were added to the wells. Enzyme activity was measured at 37 °C for 1 h, by measuring the fluorescence emission of the hydrolysed product ($l_{ex}$ = 342 nm, $l_{em}$ = 440) in a microplate reader spectrofluorimeter (Tecan Infinite M200 Pro). Data were processed in Prism 9 to calculate IC$_{50}$ values.

**MTT assays.** Inhibition against HeLa cells was tested for the phosphorylated, chemically dehydrated and control (unmodified) forms of peptide 75.1. Stock HeLa cells were resuspended in 10 ml HEPES-buffered high-glucose Dulbecco's modified Eagle medium (DMEM) supplemented with GlutaMAX (Gibco). The medium also contained 10% fetal calf serum and 50 mg ml$^{-1}$ gentamicin. The cells were centrifuged for 5 min at 1,000$g$ and room temperature. The medium was discarded, and the cells were resuspended in 10 ml fresh medium. The cells were put in a culture dish and incubated for 3–4 days at 37 °C. The cells were checked under the microscope and treated further only when 60–80% of the surface was covered with cells. The medium was removed from the culture flask and the cells were washed with 10 ml phosphate-buffered saline (PBS). The PBS was discarded and the cells treated with 2 ml trypsin-EDTA solution. When the cells were detached, 10 ml of medium was added and centrifuged for 5 min at 1,000$g$ and room temperature. The supernatant was discarded and 10 ml fresh medium was added. Then, 2 ml of the cell suspension was put into a fresh culture flask containing 10 ml medium. Cells healthy enough for cytotoxicity assays were counted and diluted to a 10,000 cells per ml solution. Then, 96-well plates were filled with 200 μl cell suspension per well. All of the plates were incubated overnight at 37 °C. The outer wells were not used for the assay. A starting concentration of 100 μM of phosphorylated, chemically dehydrated and control (unmodified) forms of peptide 75.1 (2 μl of 1.25 mM stock solutions in DMSO) were added to the B lane wells. Doxorubicin was used as a positive control at 1 mg ml$^{-1}$, and DMSO was used as a negative control. A total of 50 μl of lane B was transferred into lane C and mixed, and this transfer to the adjacent lane was repeated until lane G. The plates were then incubated for 3 days. Then, 50 μl of 3-(4,5-dimethylthiazol-2-yl)-2,5-diphenyltetrazolium bromide (MTT) (1 mg ml$^{-1}$ in water) was added to all of the wells and incubated for 3 h at 37 °C. The supernatant was discarded and 150 μl of dimethyl sulfoxide (DMSO) was added to all of the wells. Absorbance was measured at 570 nm and the IC$_{50}$ was calculated using the 'drc' package in R.

## Experimental validation of a novel proteusin pathway ('*Ca.* E. malaspinii', HLLJDLBE BGC 3.1)

**Cloning (*E. coli* and *M. aerodenitrificans*).** Genes of the type material of '*Ca.* E. malaspinii' (Supplementary Information) in the antiSMASH-predicted BGC 3.1 (region 1) on MALA_SAMN05422137_METAG-scaffold_4, including *Nhis-ereA* (gene_139), *ereI* (gene_140), *ereM* (gene_141), *ereD* (gene_142) and *ereB* (gene_143) were codon-optimized and ordered as synthetic constructs as described in Supplementary Table 5 for expression in *E. coli*. Genes were further subcloned by Gibson assembly using the primers listed in Supplementary Table 5. All generated constructs were introduced into chemically competent *E. coli* DH5α and plated on LB agar with appropriate antibiotic selection. Plasmids were purified from single colonies and Sanger sequencing was used to verify proper insertion of genes (Supplementary Table 5).

*Nhis-ereAD*, *Nhis-ereAIMD*, *Nhis-ereADB* and *Nhis-ereAIMDB* with intergenic regions substituted with the *M. aerodenitrificans* ribosome-binding site (TTAGGAGAGTGCGGG) were subcloned by Gibson assembly in a modified pLMB509m(Gt[R]) vector[89]. Introduction into *E. coli* DH5α, isolation of plasmids and validation of correct clones by sequencing was followed by introduction of all constructs into *E. coli* SM10 for conjugation into *M. aerodenitrificans*. Conjugation was performed as described previously[89].

**Heterologous expression and protein purification.** Chemically competent *E. coli* BL21(DE3) cells were transformed with all of the constructs listed in Supplementary Table 5. Overnight cultures were prepared from single colonies and used to inoculate (1%, v/v) TB medium (1 × 50 ml) in 250 ml baffled Erlenmeyer flasks supplemented with appropriate antibiotics. Cells were grown at 37 °C, 200 rpm, until an $OD_{600}$ of around 0.8–1.2 and induced with a final concentration of 1 mM IPTG. The cultures were further incubated at 16 °C, 180 rpm for 16 h, 24 h or 48 h as specified per experiment and subsequently collected by centrifugation (6,000$g$ for 15 min) and frozen.

For purification from *M. aerodenitrificans*, 20 ml precultures in nutrient broth with the appropriate antibiotics were inoculated from freezer stocks and grown overnight at 30 °C. Then, 1% (v/v) cultures were used to inoculate 20 ml TB starter cultures with the appropriate antibiotics. Then, 1% (v/v) of starter cultures were then used to inoculate 200 ml in 1 l non-baffled Erlenmeyer flasks. Cells were grown at 30 °C, 180 rpm, until an $OD_{600}$ of around 0.8–1.2 and induced with a final concentration of 0.2% L-arabinose. The cultures were further incubated at 30 °C, 180 rpm for 16 h, 24 h, 48 h or 72 h as specified for each experiment and subsequently collected by centrifugation (6,000$g$ for 15 min) and frozen.

For protein purification, cell pellets were resuspended in 5 ml $g^{-1}$ of lysis buffer (50 mM Tris, 300 mM NaCl, 5 mM imidazole, 10% glycerol, pH 8.0). Cells were lysed by sonication (30% amplitude, 10 s on/off cycles, for a total of 3 min), and the clarified lysate was obtained by centrifugation (27,000$g$ for 45 min). The supernatant was loaded onto 1 ml of Ni-NTA agarose resin that had been equilibrated with a lysis buffer in a fritted purification column. The resin was washed with 10 column volumes (CV) of lysis buffer, 3 CV of wash buffer (50 mM Tris, 300 mM NaCl, 40 mM imidazole, 10% glycerol, pH 7.8) and finally eluted with 3 CV of elution buffer (50 mM Tris, 300 mM NaCl, 250 mM imidazole, 10% glycerol, pH 7.8). Elution fractions were analysed by SDS–PAGE, pooled and buffer-exchanged into SGT buffer (50 mM Tris, 300 mM NaCl, 10% glycerol, pH 8.0) in Amicon Ultra-4 10K spin filter devices (Millipore Sigma). The concentration of spun, buffer-exchanged proteins was determined using the Roti-Quant Bradford reagent (Carl Roth).

**Proteolytic cleavage for analysis of core peptides and generation of the core region.** For the LahT150 digest, 8 μl of purified LahT150 (ref. [90]) at a concentration of 20 μM was added at a 1:10 ratio to 72 μl of approximately 200 μM of spun-concentrated protein in glass MS vial inlets.

Digests were incubated overnight at room temperature and analysed by HPLC–HR-MS and MS/MS after approximately 16 h.

For the proteinase K digest, 16 μl of the elution was mixed with 20 μl of proteinase K buffer (100 mM Tris, 4 mM $CaCl_2$, pH 8.0) and 4 μl of proteinase K (2 mg ml$^{-1}$). Proteolytic cleavage was carried out in PCR tubes (12 h at 50 °C) and analysed by HPLC–HR-MS and MS/MS after approximately 16 h.

**Marfey's analysis of amino acids contributing to the matured precursor peptides.** Advanced Marfey's analysis was performed as described previously[92,93]. Before Marfey's analysis, 500 μg of matured Nhis-EreA precursor peptide was hydrolysed in 6 M HCl at 110 °C for 16 h. The mixture was dried in a speedvac concentrator and washed twice using 400 μl water. The dried samples were solubilized in 75 μl water, 25 μl 1 N NaHCO$_3$ and 125 μl $N\alpha$-(2,4-Dinitro-5-fluorophenyl)-L-valinamide (4 mg ml$^{-1}$ in acetone) and the mixture was heated for 1 h at 45 °C. The reaction was neutralized with 25 μl 1 N HCl and diluted with 250 μl acetonitrile/water (1:1). The samples were transferred to a HPLC vial and subsequently analysed using HPLC–HR-MS. Amino acids were verified by mass and retention time when compared to authentic amino acid standards treated in the same way as described above.

**HPLC–HR-MS method for Marfey's analysis.** Analytical HPLC–HR-MS samples (5–20 μl injections) were separated on a Dionex Ultimate 3000 *RS* UHPLC equipped with the Phenomenex Kinetex C18 (2.6 m, 100 Å, 150 × 4.6 mm) column heated to 50 °C. HPLC separation was performed according to the following method (solvent A, $H_2O$ + 0.1% FA; solvent B, ACN + 0.1% FA; flow rate: 1 ml min$^{-1}$): 2 min at 30% B; 2–9 min from 30% to 100% B; 9–10 min at 100% B; 10–10.1 min from 100% to 30% B, 10.1–14 min at 30%.

The HPLC was coupled to a Thermo Scientific Q Exactive Hybrid Quadrupole-Orbitrap Mass Spectrometer using heated electrospray ionization in positive-ion mode (spray voltage, 3,500 V; capillary temperature, 268.75 °C; probe heater temperature, 350 °C; S-lens level, 70). Full MS was detected at a resolution of 70,000 (AGC target, 1 × 10$^6$; maximum IT, 100 ms).

**Purification of SAM for in vitro EreM assays.** Before the enzymatic assays, 25 mg of commercially available SAM (Sigma-Aldrich) was dissolved in 1 ml and injected in five portions (200 μl each) to conduct additional purification on an Agilent 1260 HPLC equipped with a SemiPrep Hydro RP (4 m, 80 Å, 250 × 10 mm) column at ambient temperature. HPLC separation was performed using the following method (solvent A, 2.5 mM ammonium acetate; solvent B, 75% MeOH; flow rate, 2.5 ml min$^{-1}$): 2 min at 15% B; 2–7 min from 15% to 30% B; 7–10 min from 30% to 100% B; 10–16 min at 100% B; 16–17 min from 100% to 15% B; 17–22 min at 15%.

The purification process was monitored at 260 nm. SAM eluted in a broad peak at 6–8 min and was collected in a Falcon tube on ice, immediately flash-frozen after collection in liquid nitrogen and subsequently lyophilized. The related impurities eluted later from the column and were discarded. The lyophilized pure SAM was dissolved in 20 mM HCl and aliquoted as 100 mM stock solutions that were stored at −80 °C until further use.

**In vitro EreM assays.** SAM (0.5 mM), 1 μM 5′-methylthioadenosine nucleosidase (MTAN), 50 μM epimerized Nhis-EreA precursor, 30 mM $MgCl_2$ and 5 μM EreM were dissolved in SGT buffer (50 mM Tris, 300 mM NaCl, 10% glycerol, pH 8.0) to a total volume of 100 μl and incubated in glass MS vial inserts overnight at 37 °C. Then, 5 μM LahT was added to the in vitro assays and incubated for 2 h at room temperature before analysis by HPLC–HR-MS/MS.

**In vitro EreM assays with $^{13}$C-labelled SAM.** $^{13}$C-SAM (0.5 mM), 1 μM MTAN, 50 μM epimerized Nhis-EreA precursor, 30 mM $MgCl_2$ and 5 μM

EreM were dissolved in SGT buffer (50 mM Tris, 300 mM NaCl, 10% glycerol, pH 8.0) to a total volume of 100 µl and incubated in glass MS vial inserts overnight at 37 °C. Then, 5 µM LahT was added to the in vitro assays the next day and incubated for 2 h at room temperature before analysis by HPLC−HR-MS/MS.

**In vitro EreM assays with $^{13}C$-labelled $^{13}CH_3$-SAM using an enzyme cascade starting with $^{13}C$-labelled $^{13}CH_3$-L-methionine for NMR spectroscopy.** $^{13}CH_3$-L-methionine (5 mM), 10 mM adenosine-triphosphate, 100 mM KCl, 30 mM $MgCl_2$, 1 µM MTAN, 11 µM SAM synthase, 40 µM epimerized Nhis-EreA precursor and 40 µM EreM were dissolved in 50 mM Tris at pH 8.0 to a total volume of 500 µl and incubated in an Eppendorf tube overnight at 37 °C. Then, 50 µl of $D_2O$ was added to the assay mixture and transferred into an NMR tube. Proton and HSQC spectroscopy was performed on the Bruker 600 MHz NMR spectrometer equipped with a cryoprobe. To determine the location of protons attached to $^{13}C$-labelled carbons, an additional $^1H$ NMR was recorded with parameters that enable decoupling of carbons and protons. These parameters cause a splitting of the respective protons attached to $^{13}C$-labelled carbons.

**In vitro EreI assays.** $(NH_4)_2FeSO_4$ (800 µM) was incubated with 50 µM EreI in SGT buffer (50 mM Tris, 300 mM NaCl, 10% glycerol, pH 8.0) for 20 min on ice. Then, 1 mM 2-oxoglutarate and 1 mM dithiothreitol were added and incubated for another 20 min on ice. Epimerized Nhis-EreA precursor (5 µM) was added to yield a total volume of 100 µl and the reaction mixture was incubated in Eppendorf tubes for 20 min at 30 °C. The reaction was quenched by boiling the assay mixture at 95 °C for 10 min. LahT (5 µM) was added to the in vitro assays and incubated for 2 h at room temperature before analysis using HPLC−HR-MS/MS.

**Site-directed mutagenesis to generate core variants.** Primers for site-directed mutagenesis (Supplementary Table 5) were synthesized and used to amplify template DNA from EreAD-pET Duet (Amp$^R$). Mutagenesis was accomplished using PCR amplification, KLD treatment and enrichment, and transformation into *E. coli* DH5α for isolation of plasmids and validation of the correct clones by sequencing. Owing to the highly repetitive nature of the core sequence, truncated core variants were also generated during site-directed mutagenesis and also tested for EreM modification (Supplementary Table 5).

**Orthogonal $D_2O$-based induction system for labelling epimerized residues.** *E. coli* BL21 (DE3) cells were co-transformed with *Nhis-ereA* in pACYCDuet-1 and *ereD* in pCDFBAD/*Myc*-His A and plated on LB agar containing chloramphenicol (25 µg ml$^{−1}$) and ampicillin (100 µg ml$^{−1}$) and grown for 20 h at 37 °C or until colonies appeared. These colonies were used to inoculate 20 ml LB with chloramphenicol (25 µg ml$^{−1}$) and ampicillin (100 µg ml$^{−1}$), and the cultures were grown overnight. The next day, four separate 50 ml Falcon tubes containing TB medium, (15 ml), chloramphenicol (25 µg ml$^{−1}$) and ampicillin (100 µg ml$^{−1}$) were inoculated with 150 µl and shaken at 37 °C, 250 rpm to an $OD_{600}$ of 1.6−2. Cultures were cooled on ice for 30 min, induced with IPTG (0.1 mM final concentration) and shaken (180 rpm at 16 °C) for 18 h. The cultures were centrifuged (10 min at 10,000*g*) and the supernatant was removed. The cell pellets were then washed with TB medium (1 × 15 ml) to remove any residual IPTG. This was followed by two washes with 1 ml TB in $D_2O$. The washed cell pellets were resuspended in 15 ml TB medium in $D_2O$ containing ampicillin (100 µg ml$^{−1}$ in $D_2O$) and L-arabinose (0.2% w/v in $D_2O$) and shaken (180 rpm at 16 °C) for 18 h. The cultures were combined and centrifuged (30 min at 15,000*g*) and the pellets were resuspended in 4 ml lysis buffer per gram of cell pellet and purified as described above. For the control, the same procedure was followed with normal TB medium.

**Generation of the *M. aerodenitrificans* Δaer mutant.** The suicide vector pSW8197 (ref. [94]) was used as a basis to create a stable and markerless deletion of the aeroneamide (*aer*) cluster in *M. aerodenitrificans*. The primer pairs Aer1f/r and Aer2f/r were used to amplify 500 bp homologous regions up- and downstream of the *aer* cluster. The resulting DNA products were assembled into PCR-amplified pSW8197 (pSWAerKO-f/r) using Gibson assembly and transformed into *E. coli* SY327 electrocompetent cells and sequence-verified after plasmid extraction from the resulting colonies.

The resulting plasmid (pSW8197_aerKO) was then transformed into chemically competent *E. coli* ST18 donor cells[95] and selected with 50 µg ml$^{−1}$ kanamycin and 5-aminolevulinic acid (required by *E. coli* sT18). Conjugation with the *M. aerodenitrificans* wild type was performed as previously described, plated on LB agar plates with 50 µg ml$^{−1}$ 5-aminolevulinic acid and incubated at 37 °C for 24 h. Integrants were selected by plating on nutrient agar plates containing 25 µg ml$^{−1}$ kanamycin and 400 µg ml$^{−1}$ ampicillin at 30 °C and confirmed by PCR (aerKO seqF/pswAraC-R). Positive integrants were grown non-selectively in 5 ml nutrient broth overnight at 30 °C and plated on nutrient agar with 0.5% (w/v) L-arabinose (to induce counter-selectable ccdB toxin) and 100 µg ml$^{−1}$ ampicillin, resulting in colonies with successful deletions or wild-type revertants. Successful deletion mutants of the *aer* cluster were identified using PCR with the primers aerKO seqF/R, which anneal to regions on the genome outside the flanking regions used to construct the vector. The resulting PCR product was verified by sequencing.

**Phylogenetic analysis of FkbM-family proteins.** PfamScan classified EreM from '*Ca.* E. malaspinii' as belonging solely to the FkbM methyltransferase (PF05050) family. To identify other FkbM-family proteins involved in natural product biosynthesis, the FkbM-family methyltransferase HMM (Methyltransf_21.HMM in Pfam_A) was used to query all protein-coding sequences in MIBiG (v.2.0)[30] using hmmsearch in HMMER v.3.1b2 (http://hmmer.org/) with the default parameters and the --cut_nc PFAM noise cut-off. Hits within 37 characterized BGCs in MIBiG were identified (Supplementary Table 5) and associated literature was manually assessed for experimental evidence of FkbM-family methyltransferase activity. Eight proteins were excluded on the basis of the FkbM hit falling outside the defined BGC cluster boundaries and having no apparent role in biosynthesis based on the final natural product structure (Supplementary Table 5). Four FkbM family members had experimental evidence for *O*-methyltransferase activity in the form of heterologous expression or genetic studies. A total of 25 FkbM-family proteins were documented in publications by authors to have likely *O*-methyltransferase activity on the basis of the final natural product structure, biosynthetic logic and bioinformatic evidence. The summed 29 FkbM-family proteins were aligned using MUSCLE (v.3.8.1551)[81] with two outgroups involved in proteusin biosynthesis, PoyE (AFS60641.1) and AerE (AFS60641.1) from a different methyltransferase protein family (PF05175). The protein alignment was assessed and all columns containing 50% or more gaps were removed using Trimal v.1.2. The trimmed alignment was used for phylogenetic model selection using IQ-TREE (v.2.0.3)[96] and the V5+F+R5 model was selected based on best-fit using the Bayesian and Akaike information criteria. IQ-TREE was then used to estimate a maximum-likelihood phylogeny with 5,000 resamplings using the ultrafast bootstrap approximation[97]. The scripts used for bioinformatic analysis are available at GitHub (https://github.com/serina-robinson/fkbm-bioinformatics).

**HPLC−HR-MS and MS/MS analysis.** HPLC−HR-MS and MS/MS analyses were performed on the Thermo Scientific Dionex UltiMate 3000 UHPLC coupled to a Thermo Scientific Q Exactive mass spectrometer using heated electrospray ionization in positive ion mode with the Phenomenex Kinetex 2.6 µm XB-C18 100 Å (150 × 4.6 mm) column. Analytical HPLC−HR-MS samples (5−20 µl injections) were separated on the Dionex Ultimate 3000 *RS* UHPLC equipped with a Phenomenex Kinetex C18 (2.6 m, 100 Å, 150 × 4.6 mm) column heated to 50 °C. HPLC separation was performed using the following standard methods:

Method A: solvent A: $H_2O$ + 0.5% FA; solvent B: 1-propanol + 0.5% FA; flow rate: 0.5 ml min⁻¹; 2 min at 20% B; 2–14 min from 20% to 80% B; 14–17 min at 80% B; 17–17.1 min from 80% to 20% B, 17.1–20 min at 20%.

Method B: solvent A: $H_2O$ + 0.1% FA; solvent B: acetonitrile + 0.1% FA; flow rate: 1.0 ml min⁻¹; 2 min at 2% B; 2–12 min from 2% to 100% B; 12–16 min at 100% B; 16–17 min from 100% to 2% B, 17.1–20 min at 2%.

The HPLC was coupled to a Thermo Scientific Q Exactive Hybrid Quadrupole-Orbitrap Mass Spectrometer using heated electrospray ionization in positive-ion mode (spray voltage, 3,500 V; capillary temperature, 268.75 °C; probe heater temperature, 350 °C; S-lens level, 70). Full MS was detected at a resolution of 70,000 (AGC target: $1 \times 10^6$; maximum IT, 100 ms). MS² fragmentation was performed at a resolution of 35,000 (AGC target, $2 \times 10^5$; maximum IT, 100 ms, isolation window, 4.0 $m/z$). Normalized collision energy was 20, 25 and 30 for +3 charge states. Parallel reaction monitoring was performed at a resolution of 17,500 (AGC target $2 \times 10^5$; maximum IT, 200 ms; isolation windows, 1.4 $m/z$) and a normalized collision energy of 18, 20 and 24 for +2 and +3 charge states.

## Statistics and reproducibility

Wherever appropriate, correction for multiple testing was performed using false-discovery rate correction. Wherever appropriate and if not specified otherwise, statistical tests performed were two-sided.

The box plots were plotted in R (v.4.0.0–v.4.1.2) using ggplot2 (v.3.3.0–v.3.3.5) and defined as follows: the bottom and top hinges correspond to the first and third quartiles (the 25th and 75th percentiles), the top whisker extends from the hinge to the largest value no further than 1.5 × IQR from the hinge (where the IQR is the interquartile range, or distance between the first and third quartiles). The bottom whisker extends from the bottom hinge to the smallest value at most 1.5 × IQR. Data points beyond the end of the whiskers are outliers are plotted individually, except for in Fig. 1c, owing to the large number of points and space constraints.

Fig. 1e was plotted using the R package UpSetR (v.1.4.0)[98]. The trees in Fig. 2 were plotted using the R package ggtree (v.3.3.0.901)[99].

## Reporting summary

Further information on research design is available in the Nature Research Reporting Summary linked to this paper.

## Data availability

The metagenomic and metatranscriptomic data used in this study were downloaded from the European Nucleotide Archive (ENA) and a summary of their accessions is provided in Supplementary Table 1. Publicly available genomes were downloaded from Figshare (https://doi.org/10.6084/m9.figshare.4902923) for manually curated MAGs from *Tara* Oceans, from ENA using the project accession PRJEB33281 for GORG and from https://mmp2.sfb.uit.no/databases/ for MarDB. The GEM MAGs were downloaded from https://portal.nersc.gov/GEM/. MAGs contained in the GTDB r89 were downloaded from https://data.gtdb.ecogenomic.org/releases/release89/. The MIBiG and BiG-FAM databases can be accessed at https://mibig.secondarymetabolites.org/ and https://bigfam.bioinformatics.nl/, respectively. The data produced in this study, including metagenomic assemblies, bins and MAGs have been deposited at the ENA under the accession PRJEB45951 and a list of individual accession identifiers is provided in Supplementary Table 1. Other supporting data have been deposited at Zenodo (https://doi.org/10.5281/zenodo.4474310), and the OMD can be interactively accessed online (https://microbiomics.io/ocean/). Additional material generated in this study is available on request.

## Code availability

The code used for the analyses performed in this study is accessible at GitHub (https://github.com/SushiLab/magpipe/) and archived at Zenodo (https://doi.org/10.5281/zenodo.6393817).

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

**Acknowledgements** We thank the members of *Tara* Oceans, which has been driven by a collaborative engagement of the *Tara* Ocean Foundation, countless scientific project members and associated institutional partners. We are grateful for the availability of the data generated in the context of the BioGEOTRACES, Malaspina, HOT, BATS and GORG projects as well as the data compiled as part of the Mar Databases project, which formed an essential basis of this work. We thank the staff at the ETH Zurich IT Services, HPC facilities and Biomolecular NMR Spectroscopy Platform as well as the Functional Genomics Center Zurich, in particular A. Patrignani, for their support; S. Booker for the gift of the pBAD42-BtuCEDFB plasmid; J. Vorholt, E. Rocha and R. Neher for their feedback throughout this project; P. Hugenholtz and T. J. Williams for their comments on the preprint version of this Article; and A. Jermy for his advice during the writing of this manuscript. This Article is contribution number 130 of *Tara* Oceans. This work was supported by funding from the ETH and the Helmut Horten Foundation; the Swiss National Science Foundation (SNSF) through project grants 205321_184955 to S.S., 205320_185077 to J.P. and the NCCR Microbiomes (51NF40_180575) to S.S.; by the Gordon and Betty Moore Foundation (https://doi.org/10.37807/GBMF9204) and the European Union's Horizon 2020 research and innovation programme under grant agreement no. 101000392 (MARBLES) to J.P.; by an ETH research grant ETH-21 18-2 to J.P.; and by the Peter and Traudl Engelhorn Foundation and by the European Union's Horizon 2020 research and innovation programme under the Marie Skłodowska-Curie grant agreement no. 897571 to C.C.F. S.L.R. was supported by an ETH Zurich postdoctoral fellowship 20-1 FEL-07. M.L., L.M.C. and G.Z. were supported by EMBL Core Funding and the German Research Foundation (DFG, Deutsche Forschungsgemeinschaft, project no. 395357507, SFB 1371 to G.Z.). M.B.S. was supported by the NSF grant OCE#1829831. C.B. was supported by the European Research Council (ERC) under the European Union's Horizon 2020 research and innovation programme (grant agreement Diatomic, no. 835067). S.G.A. was supported by the Spanish Ministry of Economy and Competitiveness (PID2020-116489RB-I00). M.K. and H.M. were funded by the SNSF grant 407540_167331 as part of the Swiss National Research Programme 75 'Big Data'. M.K., H.M. and A.K. are also partially funded by ETH core funding (to G. Rätsch).

**Author contributions** S.S., S.L.R. and J.P. conceived and co-supervised the study. L.P., S.S., J.P. and S.L.R. designed the research. L.P., H.-J.R., Q.C., S.K., G.S. and A.M. performed bioinformatic and statistical analyses. C.C.F., F.H., S.L.R., A.B., A.L., A.D.G. and C.I.C. performed experimental work. L.P., C.P., H.-J.R., D.G., M.K., H.M., M.L., L.M.C., G.Z. and A.K. designed and implemented web and data resources. A.A.Z., D.R.C., M.B.S., T.O.D., P.S., S.G.A., J.M.G., P.W., P.B. and C.B. contributed samples, data and scientific input. L.P. and S.S. drafted the manuscript with S.L.R., C.C.F., F.H. and J.P. All of the authors reviewed the submitted manuscript and approved the final version.

**Competing interests** The authors declare no competing interests.

**Additional information**
**Correspondence and requests for materials** should be addressed to Serina L. Robinson, Jörn Piel or Shinichi Sunagawa.

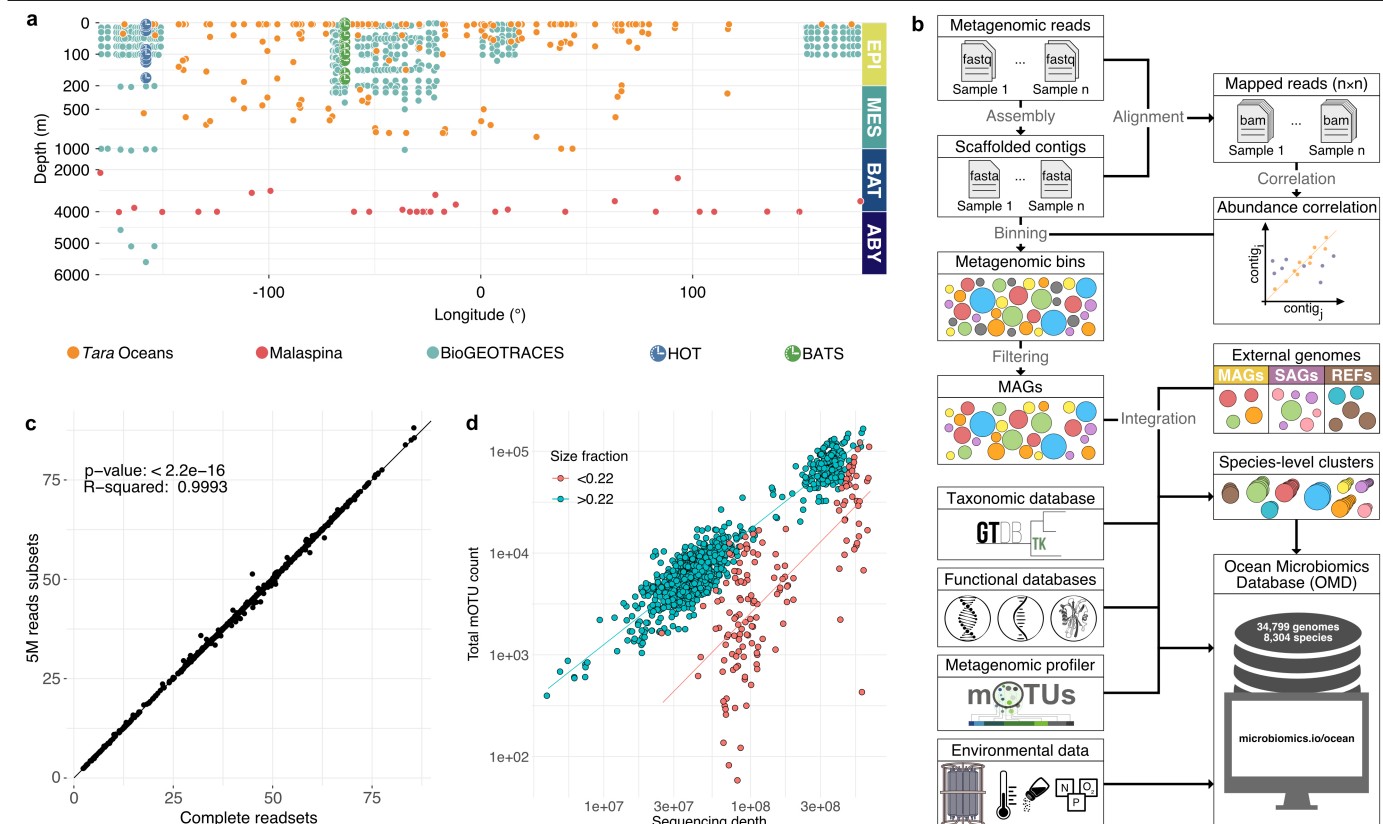

**Extended Data Fig. 1 | Depth distribution of the metagenomes used in this study; overview of the bioinformatic pipeline and proxies for sequencing depth. (a)** 1,038 publicly-available ocean microbial community genomes (metagenomes) were collected across all major depth layers (1 - 5,601 m) in the context of different ocean expeditions and time series programmes; EPI - epipelagic layer; MES - mesopelagic layer; BAT - bathypelagic layer; ABY - abyssopelagic layer. **(b)** Quality-controlled, high-throughput DNA sequencing reads from ocean microbial community samples were individually assembled into metagenomic scaffolded contigs (scaffolds). Sequencing reads from large subsets (n ranging from 58 to 610) of all samples were aligned to scaffolds of each individual sample to compute relative copy-number abundances for each scaffold in each sample. Based on a combination of tetranucleotide frequency, within-sample co-abundance and between-sample abundance correlations, scaffolds were grouped into a total of 62,874 metagenomic bins, each with total nucleotide sequence lengths of > 200 kb. These metagenomic bins were filtered for genome completeness and contamination, resulting in 26,293

metagenome assembled genomes (MAGs). These MAGs were complemented with external sets of MAGs, single amplified genomes (SAGs) and genomes from cultured isolates (REFs). The combined set of 34,799 genomes was clustered at the species level using a 95% average nucleotide identity (ANI) and, along with taxonomic and functional annotations, abundance profiles and contextual information, compiled into the Ocean Microbiomics Database (OMD); see methods for details (Methods). **(c)** Comparing mapping rates obtained from mapping subsampled readsets compared to those obtained from mapping the total number of reads shows that this procedure yields almost identical results at considerably less computational costs. **(d)** mOTUs counts as a good proxy for sequencing depth. We find a strong correlation in prokaryote-, particle-enriched and virus-depleted communities, while this correlation is more variable in virus-enriched communities. This observation is actually in support of using the mOTUs count rather than sequencing depth when focusing on the bacteria and archaeal component of microbial communities, as we do here.

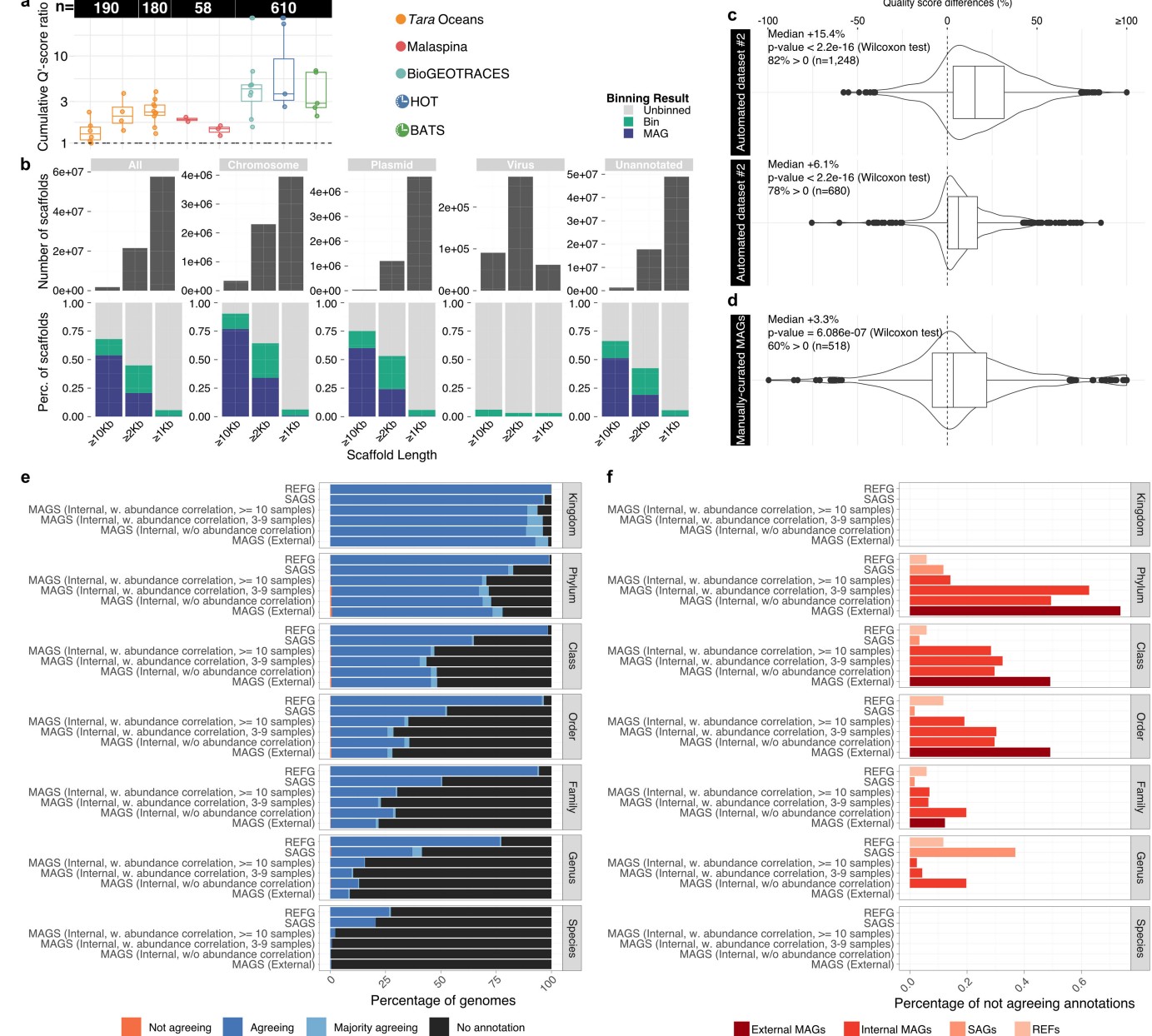

**Extended Data Fig. 2 | Impact of abundance correlation on MAGs recovery and quality, quality improvement over other ocean MAGs datasets, recovery of mobile genetic elements and evaluation of genome chimerism.**
**(a)** In this study, MAGs were reconstructed using abundance correlation information (Extended Data Fig. 1b) (Methods), which resulted in both higher cumulative quality scores per sample and individual quality scores per MAG. The ratio of cumulative quality scores (Supplementary Information) of MAGs binned with and without differential coverage information was on average (median) 2.3 across the different datasets. Per individual MAGs, a mean quality score increase of 20% was achieved. The number of samples used for differential coverage profiling are indicated above the boxplots. The colours of the boxplots reflect the different datasets as indicated in Fig. 1b. **(b)** We investigated the bin membership of > 80 M scaffolds across size and fragment type. These scaffolds were annotated to identify chromosomes, plasmids and phages (Supplementary Information). The difference between chromosomes and plasmids binning rates provides an evaluation of the bias of the MAG reconstruction against hypervariable regions within the genomes. Annotations were integrated to classify scaffolds as follows, **chromosomes** (*'eukrep = Prokarya & plasflow prediction = chromosome & cbar prediction = Chromosome & plasmidfinder plasmid = NaN & deepvirfinder p-value > 0.05 & virsorter score = NaN'*), **plasmids** (*'(plasmidfinder plasmid != NaN | (plasflow prediction = plasmid & cbar prediction = Plasmid)) & eukrep = Prokarya &*

*virsorter score not in [1, 2] & deepvirfinder p-value > 0.05'*), **viruses** (*'virsorter score > = 1 & deepvirfinder p-value < 0.01 & eukrep = Prokarya & plasflow prediction != plasmid & cbar prediction != Plasmid'*) or **unannotated**. By benchmarking the quality of the MAGs reconstructed in this study (Supplementary Information), we found that combining single-sample assemblies with large-scale abundance correlations achieved on average significantly higher community-defined quality scores[60] than and **(c)** two datasets of automatically generated MAGs, dataset #1[100] and dataset #2[25], and **(d)** even manually curated MAGs[26]. 'n' denotes the number of possible comparisons (i.e. number of shared species) with the different MAGs sets. All genomes in the extended OMD were evaluated for chimerism using the taxonomic annotation of 10 universal single copy marker genes (Supplementary Information). **(f)** For each taxonomic level, the genomes were classified as: "No annotation" if a maximum of one gene out of 10 was annotated; "Agreeing" if all genes had the same annotation; "Majority agreeing" if more than half agreed and "Not agreeing" otherwise. The evaluation was split for the genomes origin (y-axis). **(g)** Percentage of "Not agreeing" annotations over all the annotated clades (i.e. the sum of "Agreeing", Majority agreeing" and "Not agreeing"). Notably, across all MAGs the rate of disagreement was < 1% with that rate being ~0.1% for MAGs with differential coverage index ≥ 10 (i.e. 75% of the MAGs), suggesting the added value of abundance correlation in reducing the rates of chimera.

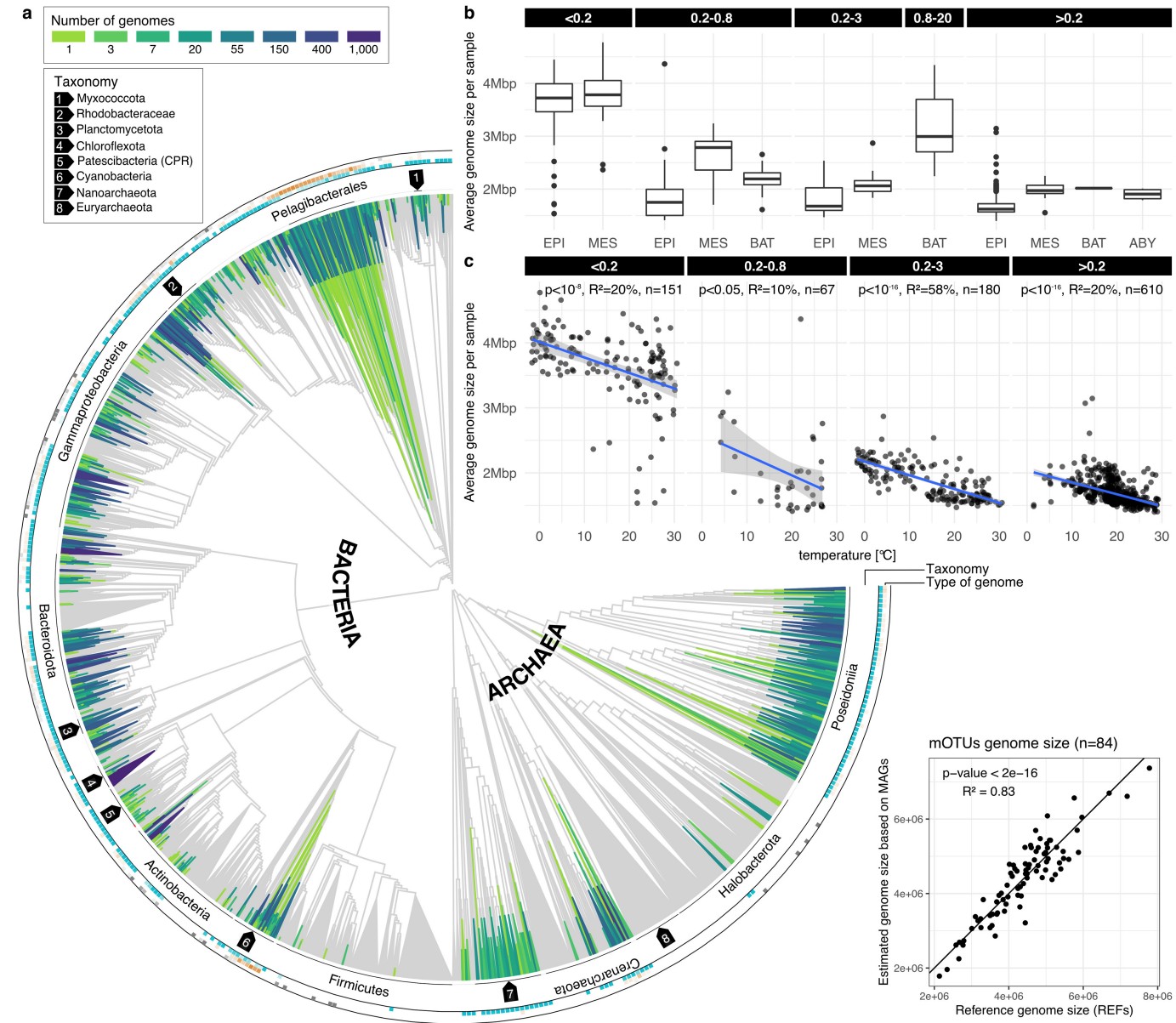

**Extended Data Fig. 3 | Different genome reconstruction strategies capture complementary phylogenomic diversity; trends in community genome sizes across the global ocean microbiome. (a)** Reconstructed MAGs, external MAGs, SAGs as well as REFs detected across the set of 1,038 ocean metagenomes were placed on the GTDB backbone trees[13] revealing that the different genome types (MAGs, SAGs and REFs) capture complementary phylogenomic diversity. Similar to Fig. 3, the green-to-blue colours of the branches indicate the number of genomes in that part of the tree. The inner layer denotes the taxonomy of specific clades (some indicated by arrows due to limited space). The outer layer represents the percentage of genomes across

the binned tree for each genome type. Clades without any genome from the OMD were left in grey. For visualization purposes, the last 15% of the nodes are collapsed. **(b, c)** The average genome size per sample was significantly larger in deeper waters (Kruskal Wallis test, p-value < 2*10$^{-16}$, n = 1,038) and was inversely correlated with temperature (linear model). **(d)** Comparing genome sizes from MAG-based predictions and reference genomes for 85 mOTUs (species-level) clusters with at least one reference genome. Genome sizes are estimated using MAGs of good quality and above only (completeness above 70%), a criteria that is met for > 80% of the mOTUs clusters.

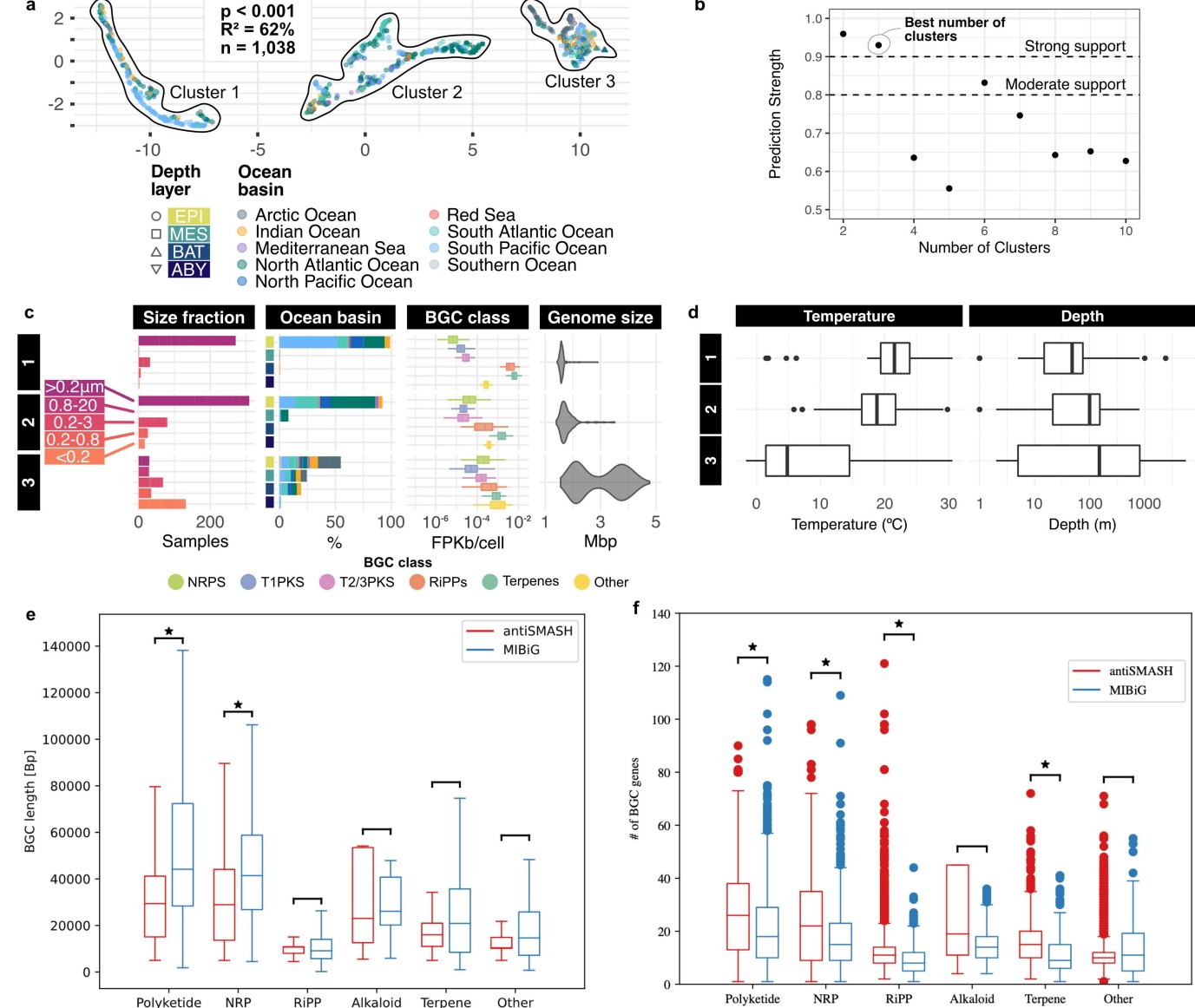

**Extended Data Fig. 4 | Structure and drivers of the ocean microbiome biosynthetic potential; evaluation of BGC completeness using length and number of genes between predicted and characterized BGCs. (a)** The abundances of GCFs (Methods) were used to compute distances between the 1,038 metagenomic samples. Using dimension reduction and density based clustering (Methods), we identified three sample clusters. **(b)** A prediction strength analysis strongly supports clustering the data into 3 groups (largest number of clusters above the 0.9 threshold). This is also confirmed by the Silhouette Index (data not shown). **(c)** These clusters were broken down by community origin, including size fractions, depth layers and ocean basins. We found significant differences in BGC class abundances (FDR-corrected pairwise Wilcoxon tests, p-value < $10^{-7}$, n = 1,038) and average genome sizes (FDR-corrected pairwise Wilcoxon tests, p-value < $2*10^{-16}$, n = 1,038) (Methods) between the clusters (Supplementary Table 2). **(d)** We found temperature and depth to be significantly different between the sample clusters identified based on biosynthetic potential composition (Kruskal Wallis test, p-value < $2*10^{-16}$, n = 1,038). RiPP - Ribosomally synthesized and Post-translationally modified Peptide; NRPS - Non-Ribosomal Peptide Synthetase; T1PKS - Type I Polyketide Synthase; T2/3PKS - Type II and III Polyketide Synthases. BGC length distributions across BGC classes are not significantly different (Wilcoxon test, significance denoted by '*' with p-value < $10^{-5}$, n >> 30) between the set of BGCs studied in this work (antiSMASH) and the characterized BGCs in MIBiG with the exception of the polyketides and non-ribosomal peptide synthetases, which may be expected based on the particularly large clusters they can encompass **(e)** and the BGCs studied in this work (antiSMASH) to have a similar or higher number of genes than the characterized BGCs in MIBiG **(f)**.

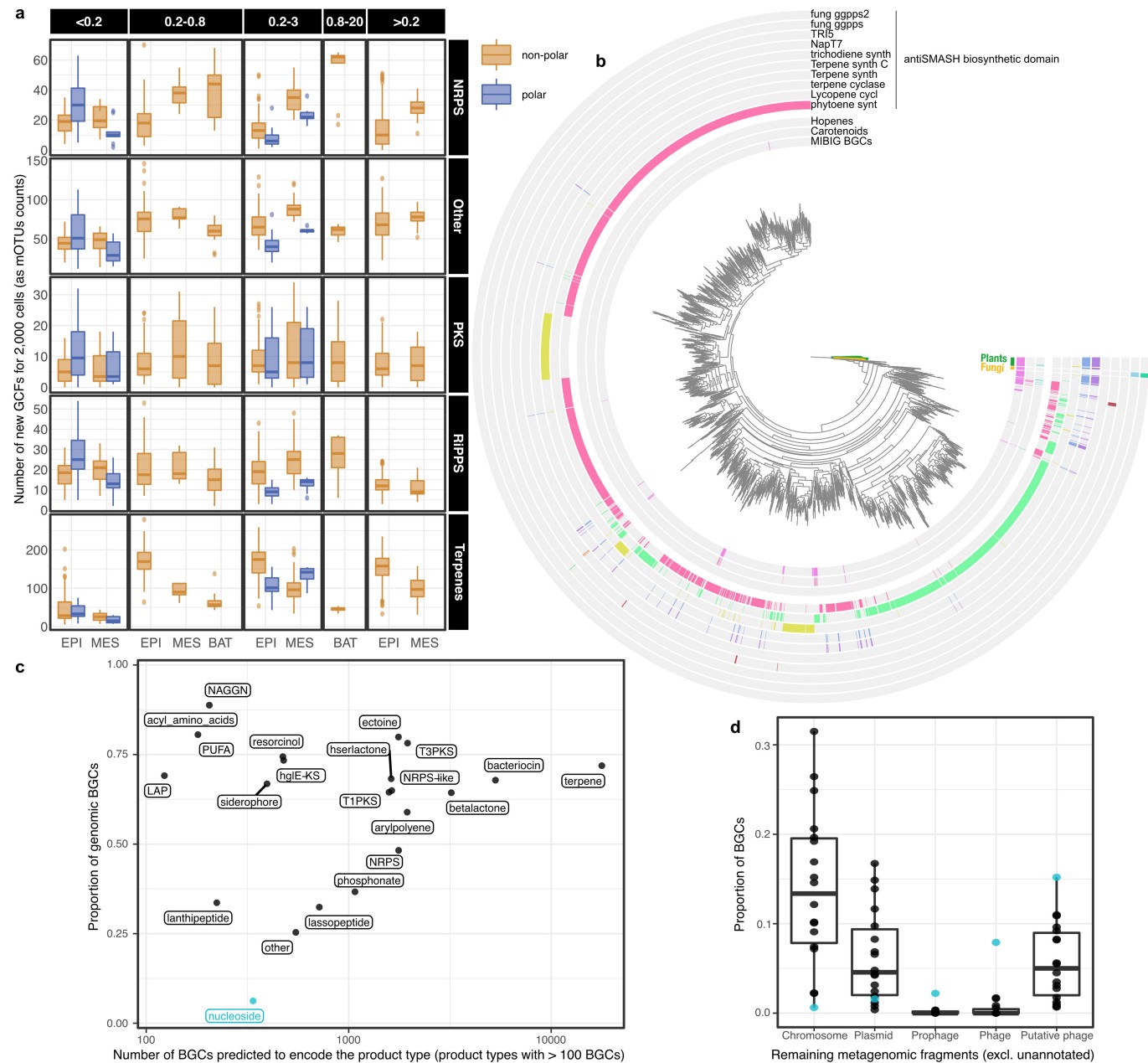

**Extended Data Fig. 5 | GCF novelty across latitude, depth layers and size fractions for each BGC class and distribution of nucleoside BGCs across genomic and metagenomic fragments. (a)** We estimated the discovery potential of different microbial communities by counting the number of new GCFs (Methods) detected in a sample after rarefaction of per-cell GCFs abundance profile to 2,000 cells. Although well studied communities (non-polar epipelagic prokaryote-enriched (0.2–3 μm) and virus-depleted (>0.2 μm)) displayed the highest discovery potential for terpenes, least explored communities (polar, deep, virus- and particle-enriched) were found to have the highest potential for NRPS, PKS, RiPPs or other natural products discovery. Polar is defined as absolute latitude > 60°. NRPS: Non-Ribosomal Peptide Synthetases; PKS: Polyketide Synthase; RiPP: Ribosomally Synthesized and Post-translationally modified Peptide. **(b)** An overview of the putative terpenoid diversity. A phylogenetic tree of all terpene biosynthetic core genes (as defined by antiSMASH) identified in the OMD, in the context of the 195 MIBiG terpene biosynthetic core genes, provides an overview of the terpenoid diversity and novelty. Briefly, the 31,398 terpene biosynthetic core genes identified across all predicted BGCs were filtered (length > = 120aa, removing < 2% of the sequences), dereplicated (using MMSEQS2 13.45111[101] clustering, 60% identity) into 2,904 protein sequences and aligned with the

195 MIBiG proteins using MAFFT v7.310[102]. The resulting alignment was trimmed with trimal to remove positions with more than 50% gaps and used to build the tree using FastTree v2.1.10[103]. The inner annotation layers indicate whether a gene is coming from a MIBIG cluster and if this one was annotated as a carotenoid or hopene cluster. The outer layers correspond to the biosynthetic core gene domain according to antiSMASH categories. Plants were used to root the tree. **(c)** Investigation of the proportion of BGCs binned within a MAG by product type showed that nucleosides were most rarely encoded in MAGs. **(d)** Breakdown by fragment type of the BGCs in the remaining metagenomic fragments. Strikingly, nucleoside BGCs were rarely encoded on predicted chromosome fragments and most often in predicted phage fragments (Supplementary Information). For this analysis, we refined the prediction described in Extended Data Fig. 2b with **prophages** (*'virsorter category = prophage & virsorter score > = 1 & eukrep = Prokarya & plasflow prediction != plasmid & cbar prediction != Plasmid'*), **phages** (*'virsorter category = phage & virsorter score > = 1 & deepvirfinder p-value < 0.01 & eukrep = Prokarya & plasflow prediction != plasmid & cbar prediction != Plasmid'*) and **putative phages** (*not in phages & (('virsorter category = phage & virsorter score > = 1) | deepvirfinder p-value < 0.05) & eukrep != Eukarya & plasflow prediction != plasmid & cbar prediction != Plasmid'*).

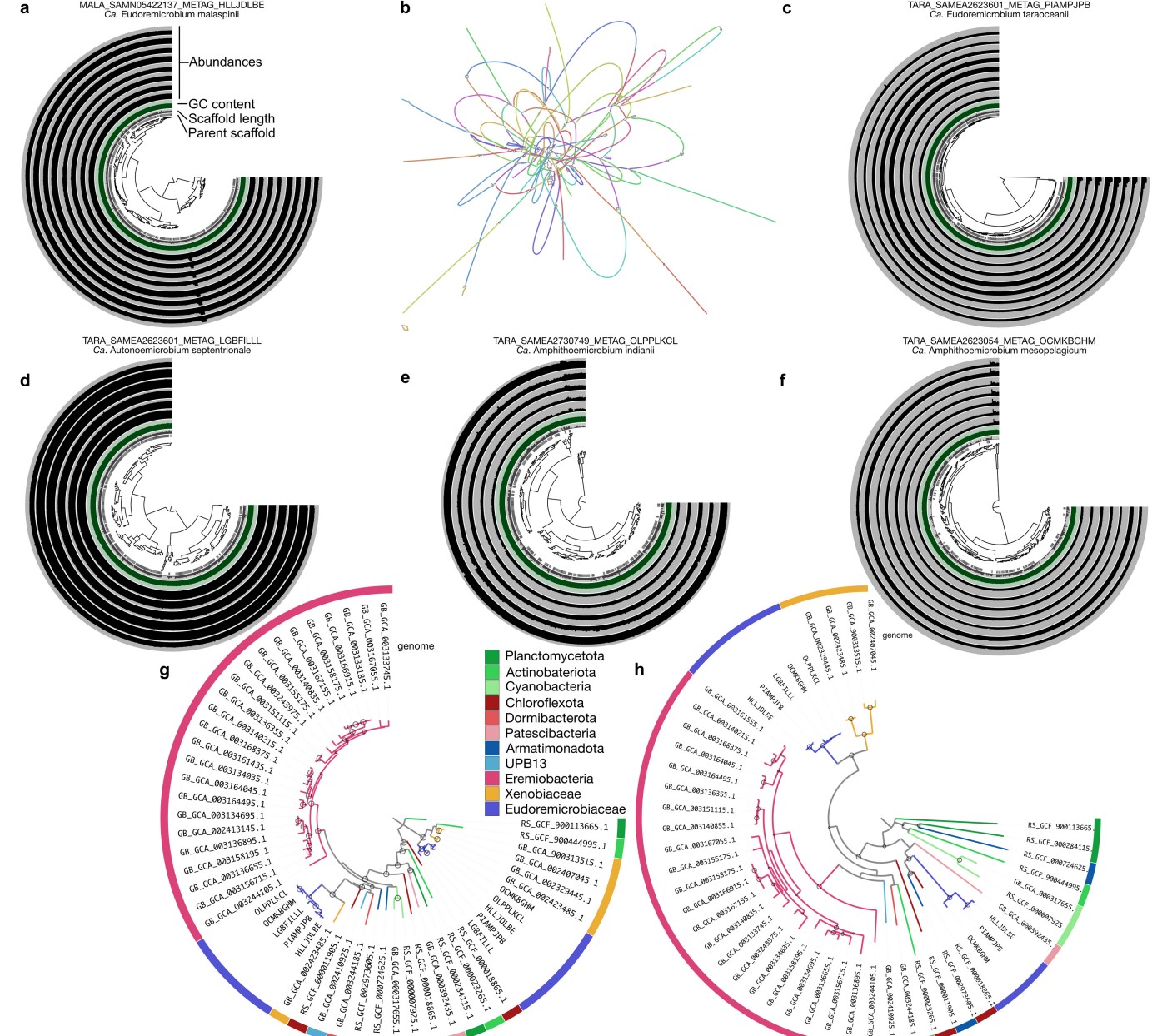

**Extended Data Fig. 6 | Manual inspection of *Ca*. Eudoremicrobiaceae MAGs and phylogeny of the duplicated marker gene COG0124. (a, c–f)** Anvi'o interface of representatives of the five *Ca*. Eudoremicrobiaceae species reveals stable abundance correlation patterns across the vast majority of the genomes, indicative of low contamination rates (Supplementary Information). **(b)** Inspection of the assembly graph for *Ca*. E. malaspinii (Supplementary Information) showed that all scaffolds from the representative genomes were connected with the exception of a single 20 kb one. **(g)** Investigating the evolutionary history of duplicated single-copy marker genes (here COG0124), we found consistent duplication across *Ca*. Eudoremicrobeaceae and the

parent order UBP9, thus ruling out the duplication as a signal of contamination in the binning process. The different evolutionary history of the second copy of COG0124 (right-hand side of the tree), with closer relationship to Actinobacteria suggests that introgression events (including before the UBP9 and *Ca*. Eudoremicrobiaceae split) could be the origin of the increased genome size and biosynthetic potential observed in *Ca*. Eudoremicrobiaceae. **(h)** Similar patterns can be found in the second duplicated marker gene (COG0522), although duplication was not detected across all *Ca*. Eudoremicrobeaceae spp. representatives.

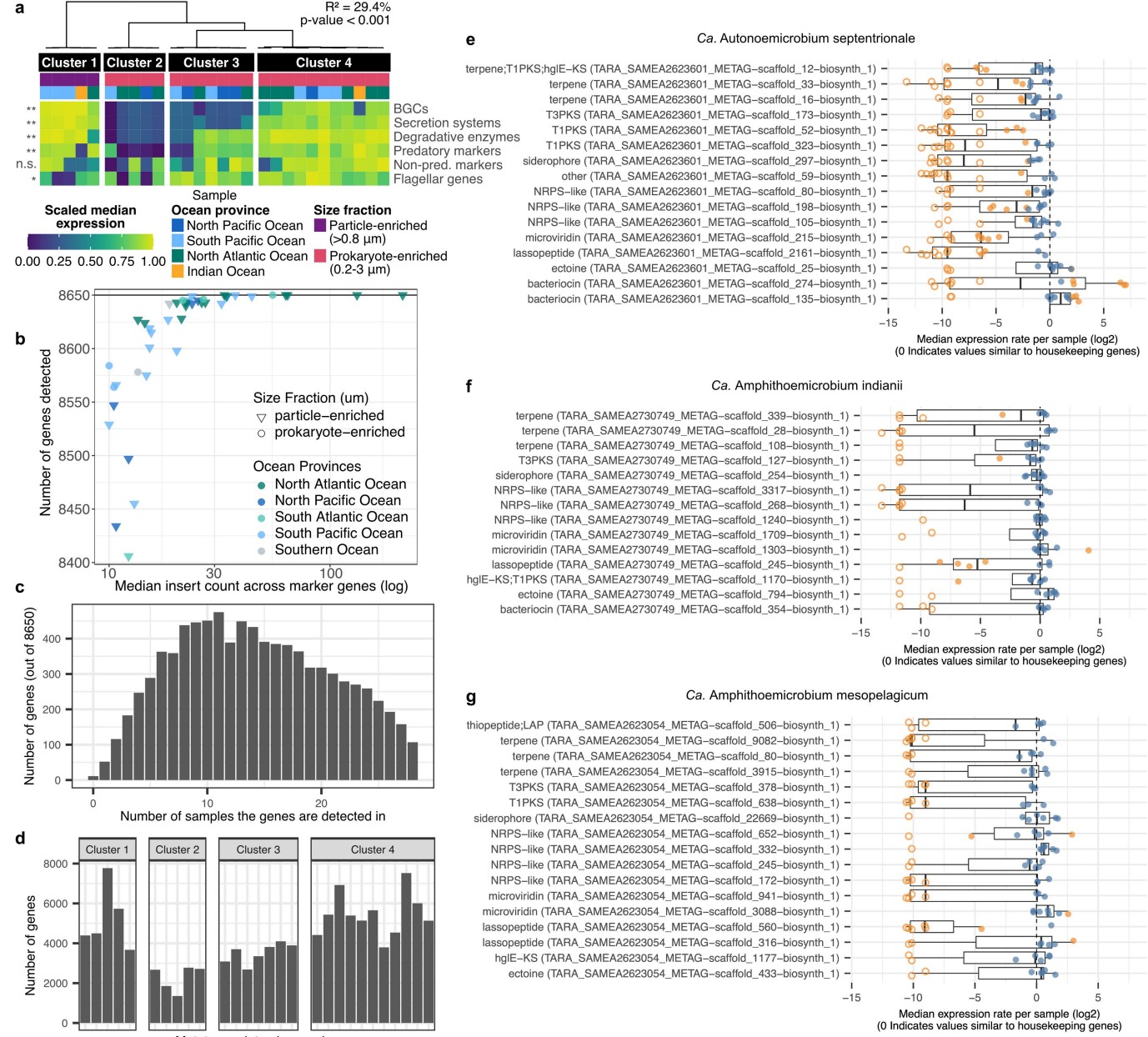

**Extended Data Fig. 7 | BGCs are the most differentially expressed genes in** *Ca*. **E. taraoceanii natural populations are expressed** *in natura* **across the** **Ca. Eudoremicrobiaceae family.** The 28 metatranscriptomic samples used for the *Ca*. E. taraoceanii expression analyses were selected based on the detection of at least 6 out of 10 universal single-copy marker genes. **(a)** Four discrete expression states explained 29.4% of the overall transcriptomic variance (PERMANOVA, p-value < 0.001, n = 28) across *Ca*. E. taraoceanii populations. One state (cluster 1) was exclusive to larger organismal size fractions. Leafs represent transcriptomic profiles and the dendrogram represents dimensionality-reduced distances (Methods). Genes associated with BGCs, secretion systems, degradative enzymes and predatory markers were differentially expressed across the states and represented the most discriminatory categories compared to 200 KEGG pathways (Supplementary Table 4). **(b)** We investigated the metagenomic detection of the 8,500 genes encoded by the *Ca*. E. taraoceanii representative, using methodology identical to the transcriptomic analyses (Methods). In samples where the 10 marker genes were detected, we counted the number of genes with one or more insert(s). We found that the 8,500 genes were detected in several ocean basins and different size fractions, with variation in detection rates likely due to

variable sequencing depths across samples and datasets. This indicates, at least for the gene set covered by the reconstructed genome, that niche partitioning may be driven by gene expression changes rather than gene content variation. **(c)** Distribution of the number of genes depending on the number of samples they were detected in. **(d)** Number of genes detected across the different metatranscriptomic samples. All BGCs encoded by **(e)** *Ca*. Autonomicrobium septentrionale, **(f)** *Ca*. Amphithomicrobium indianii and **(g)** *Ca*. Amphithomicrobium mesopelagicum representatives were found to be expressed in the natural environment (in the 623 *Tara* Oceans metatranscriptomic samples[22,87]. Some displayed near constitutive expression while others appear to be tightly regulated across the metatranscriptomes studied here. Filled circles indicate samples where active transcription was detected. Orange data points indicate values below or above a $\log_2$ fold change from the constitutive expression rate of housekeeping genes. All the BGCs encoded by *Ca*. E. taraoceanii were also found to be expressed (Fig. 3c). The expression of *Ca*. E. malaspinii BGCs could not be investigated since that species was not sufficiently abundant in the epipelagic and mesopelagic ocean, the only layers for which metatranscriptomes were available.

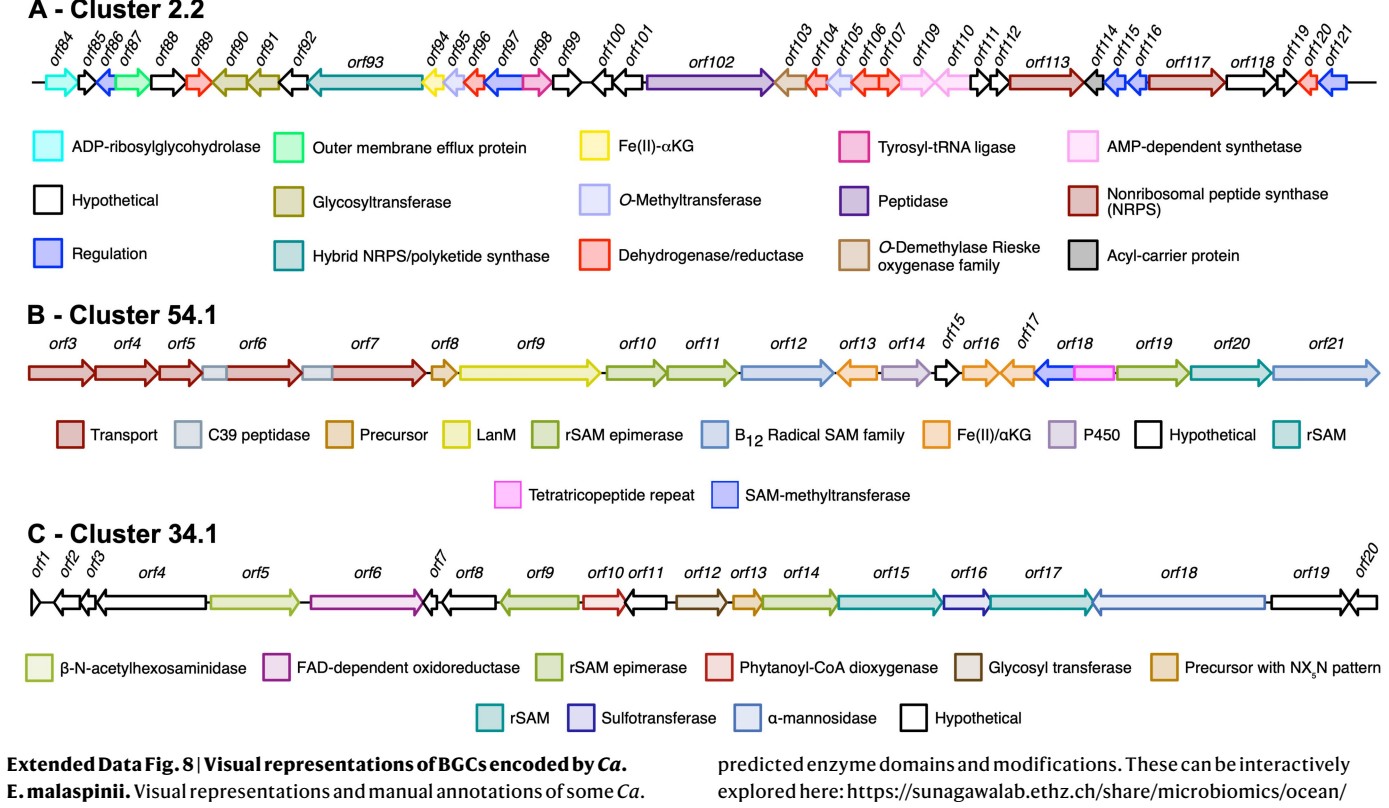

**A - Cluster 2.2**

ADP-ribosylglycohydrolase · Outer membrane efflux protein · Fe(II)-αKG · Tyrosyl-tRNA ligase · AMP-dependent synthetase · Hypothetical · Glycosyltransferase · *O*-Methyltransferase · Peptidase · Nonribosomal peptide synthase (NRPS) · Regulation · Hybrid NRPS/polyketide synthase · Dehydrogenase/reductase · *O*-Demethylase Rieske oxygenase family · Acyl-carrier protein

**B - Cluster 54.1**

Transport · C39 peptidase · Precursor · LanM · rSAM epimerase · B$_{12}$ Radical SAM family · Fe(II)/αKG · P450 · Hypothetical · rSAM · Tetratricopeptide repeat · SAM-methyltransferase

**C - Cluster 34.1**

β-N-acetylhexosaminidase · FAD-dependent oxidoreductase · rSAM epimerase · Phytanoyl-CoA dioxygenase · Glycosyl transferase · Precursor with NX$_s$N pattern · rSAM · Sulfotransferase · α-mannosidase · Hypothetical

**Extended Data Fig. 8 | Visual representations of BGCs encoded by *Ca.* E. malaspinii.** Visual representations and manual annotations of some *Ca.* Eudoremicorbium specific BGCs discussed in Supplementary Information, i.e. BGC 2.2 **(a)**, BGC 54.1 **(b)** and BGC 34.1 **(c)**. Colour-coding corresponds to predicted enzyme domains and modifications. These can be interactively explored here: https://sunagawalab.ethz.ch/share/microbiomics/ocean/db/1.0/marine_eremios/annotations/MALA_SAMN05422137_METAG_HLLJDLBE/antismash/MALA_SAMN05422137_METAG_HLLJDLBE-antismash/.

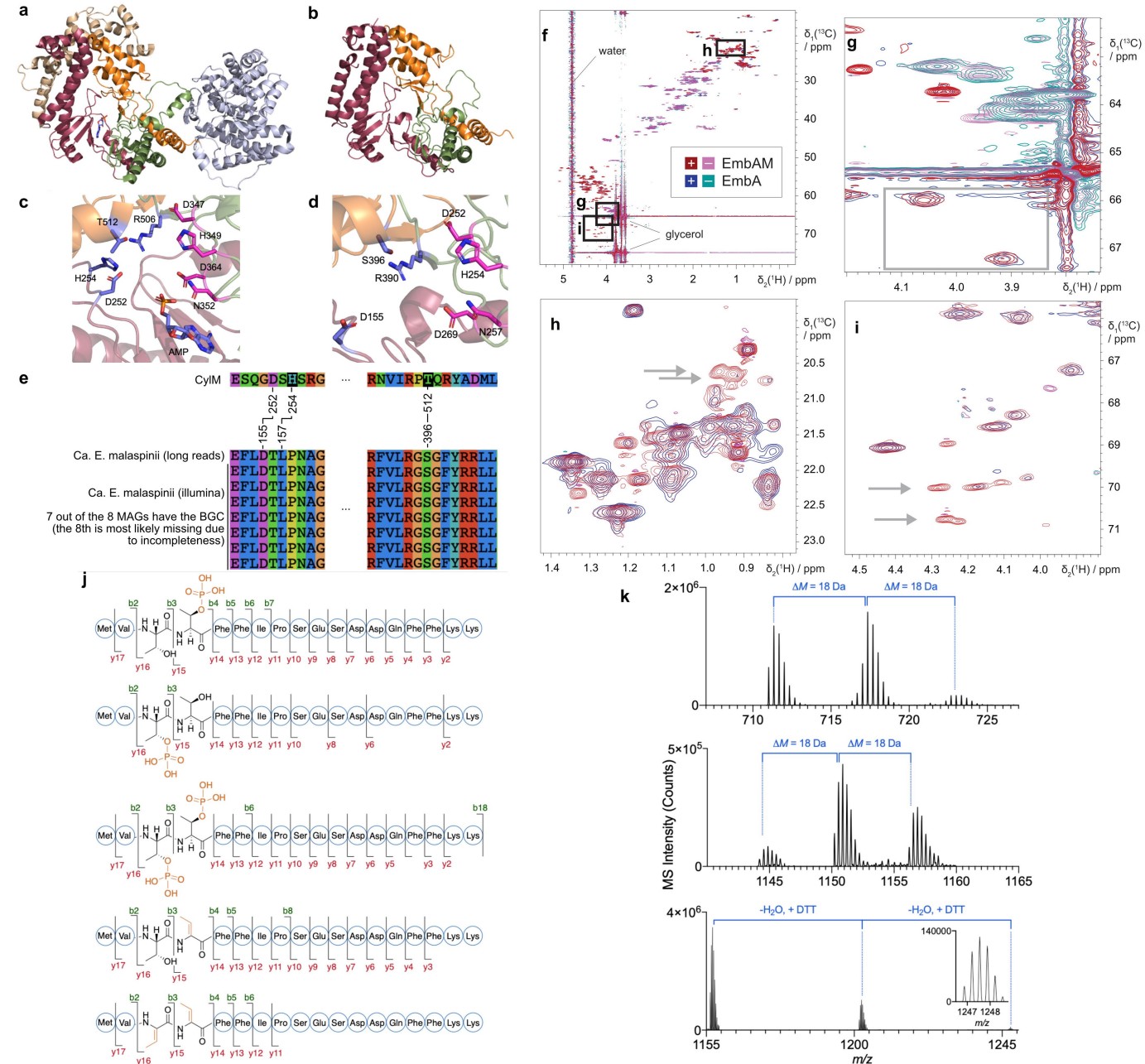

**Extended Data Fig. 9 | EmbM structural prediction and comparison to CylM (PDB: 5DZT); NMR and Mass spectrometry data for modified EmbA peptides. (a)** CylM crystal structure[43]. Coloured domains are involved in phosphorylation/dehydration and the domain in grey is responsible for cyclization. **(b)** EmbM structure prediction, highlighting similarities to CylM. **(c)** CylM active site. Residues in pink are proposed to be involved in phosphorylation and residues in purple are necessary for elimination. **(d)** Modelled active site of EmbM. **(e)** Multiple sequence alignment showing that mutated residues in the catalytic site are conserved across the independent *Ca*. E. malaspinii reconstructions. **(f)** Overlay of 2D [13C,1H] HSQC spectra of EmbA and modified EmbA (EmbAM). Multiplicity editing leads to positive signals for CH and CH3 groups (EmbA: blue, EmbAM: red) and negative signals for CH2 groups (EmbA: cyan, EmbAM: magenta). Regions of interest are identified with boxes and major buffer signals are labelled. **(g)** Serine Cβ

region. Serine Cβ moieties are identified by the negative sign of the signal (CH2-group), and the average chemical shift of 63.8 ppm. A change of the Cβ chemical shift of typically +3 ppm is expected upon a phosphorylation event[104], but there are no negative signals visible in the expected region in the EmbAM spectrum (grey box). **(h)** and **(i)**: threonine Cγ and Cβ regions, respectively, as identified by chemical shift and sign of signals. In the EmbAM spectra, additional signals are visible at expected chemical shifts for phosphorylated threonine residues, i. e. at a 13C chemical shift of 20.5 ppm for Cγ (grey arrows in **h**) and 70 ppm for Cβ (grey arrows in **i**). **(j)** HR-MS/MS fragmentation of EmbA core at different modification stages (cleaved with LahT150). **(k)** Mass spectrum of dehydrated EmbA species: unmodified, single- and double dehydrated EmbA core (top); unmodified, single- and double dehydrated EmbA cleaved with trypsin (middle); and unmodified, single- and double dehydrated, DTT adduct of EmbA cleaved with trypsin (bottom).

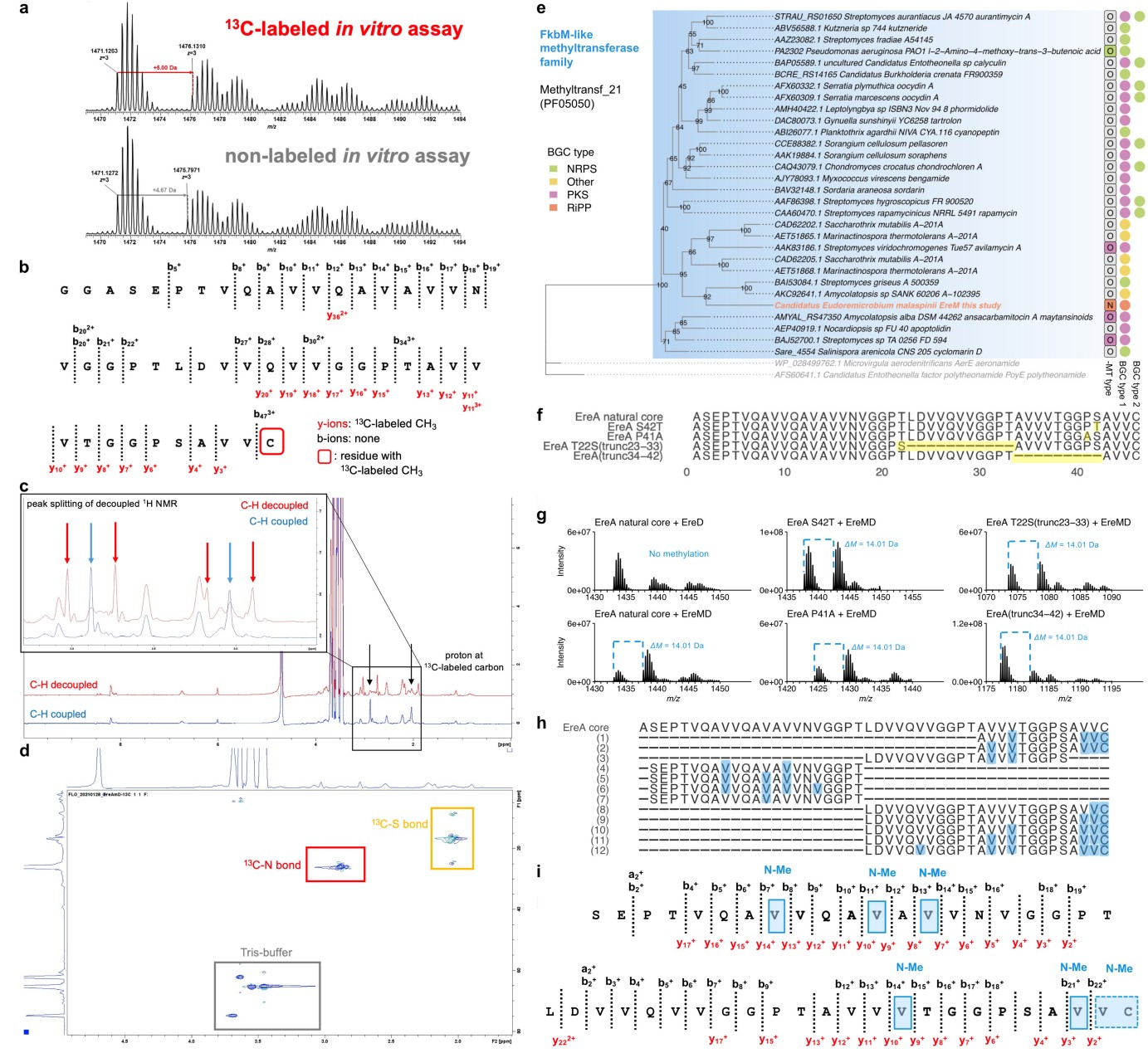

**Extended Data Fig. 10** | See next page for caption.

**Extended Data Fig. 10 | *In vitro* EreM $^{13}$C-labelling experiments, NMR and MS$^2$-fragmentation data; EreM phylogenetic tree; EreM synthetic core mass shifts and MS$^2$-fragmentation data. (a)** Mass spectra of the LahT-digested, single methylated Nhis-EreA from *in vitro* EreM assays with $^{13}$C-labelled SAM (top, red) and non-labelled SAM (bottom, grey). Top: Mass spectrum of the LahT-released 48 aa long EreA core with an N-terminal extension of two glycine residues ($m/z = 1471.1263$ Da) and the corresponding $^{13}$C-labelled methylated ($m/z = 1476.1310$ Da) core with an N-terminal extension of two leader-derived glycine residues. The mass shift of 5.00 Da ($z = 3$) is highlighted by a red arrow. Bottom: Mass spectrum of the LahT-released 48 aa long EreA core with an N-terminal extension of two glycine residues ($m/z = 1471.1272$ Da) and the corresponding methylated ($m/z = 1475.7971$ Da) core with an N-terminal extension of two glycine residues. The mass shift of 4.67 Da ($z = 3$) is highlighted by a grey arrow. **(b)** MS$^2$-fragmentation detected for the $^{13}$C-labelled core with an N-terminal extension of two glycine residues ($m/z = 1476.1310$ Da). All y-ions show masses corresponding to fragments with the addition of a $^{13}$C-labelled methyl group (red). All b-ions show masses corresponding to a fragment with no modification (black). The resulting fragmentation pattern suggests $^{13}$C-labelled methylation at the C-terminal cysteine residue (red box). MS$^2$-fragmentation data are available in Supplementary Table 5. **(c)** Overlay of a C-H decoupled (red) and standard (blue) proton NMR of an *in vitro* EreM assay with $^{13}$C-labelled SAM. The peak splitting of the singlets at 2.03 ppm and 2.88 ppm indicates the $^{13}$C-H bonds for these protons. **(d)** HSQC NMR of an *in vitro* EreM assay with $^{13}$C-labelled SAM. The spectrum shows two single signals at 2.03/17.3 ppm (yellow box) and 2.88/25.9 ppm (red box). Another four signals are detected downfield: 3.46/70.0 ppm, 3.55/70.0 ppm, 3.64/62.2 ppm and 3.69/74.6 ppm (grey box). Comparison with the literature suggest the presence of a $^{13}$C-S bond at 2.03/17.3 ppm (yellow box) from residual $^{13}$CH$_3$-L-methionine and of a $^{13}$C-N bond at 2.88/25.9 ppm (red box) from a methylated amide[105–109]. The remaining four signals are suggested to originate from the Tris-buffer of the reaction mixture (grey box). **(e)** Maximum-likelihood tree of FkbM-family methyltransferase (PF05050) Hidden Markov Model (HMM) hits within BGCs for natural products in the MIBiG 2.0 database (Supplementary Table 5). Outgroups involved in proteusin biosynthesis from a different methyltransferase protein family (PF05175) are shown in grey text. Branch support values are estimated using the 5,000 ultrafast bootstrap approximation[97] in IQ-TREE 2[96]. Letter in 'MT-type' column indicates documented *N*- or *O*-methyltransferase activity from publications based on genetic knockout or heterologous expression studies (coloured) or bioinformatic evidence, biosynthetic logic, and final natural product structure (grey). To date, EreM from this study is the only FkbM-family enzyme with reported *N*-methyltransferase activity in a characterized biosynthetic pathway. Coloured points in BGC type columns indicate the majority of FkbM-family enzymes are contained with PKS, NRPS, or Other (*e.g.*, nucleoside antibiotic) biosynthetic pathways. Thus EreM is also the only FkbM-family methyltransferase characterized in a RiPP cluster to date. **(f)** EreA core variants generated in this study. Mutation or truncation sites are highlighted in yellow. **(g)** Mass shifts of +14.01 Da corresponding to methylation of the EreA core and variants were observed expressed with EreM as compared to controls without EreM (data not shown for core variants, but results are in accordance with natural core control). All EreA variant + EreM co-productions were tested with and without EreD, but EreD co-productions are pictured. since epimerized (EreA + EreD) cores have better solubility and higher concentrations. **(h)** Proteinase K-generated fragments of the wild-type EreA core following co-productions with EreIMD reveal a mixture of variable methylation patterns. **(i)** MS$^2$-fragmentation of the wild-type EreA core after co-production with EreIMD. Mass shifts corresponding to up to 6 non-radical methylations (+84.09 Da) were observed and were localized to valine residues (highlighted in light blue, N-Me). Dashed lines around boxes indicate uncertainty regarding the position. MS$^2$-fragmentation data are in Supplementary Table 5.

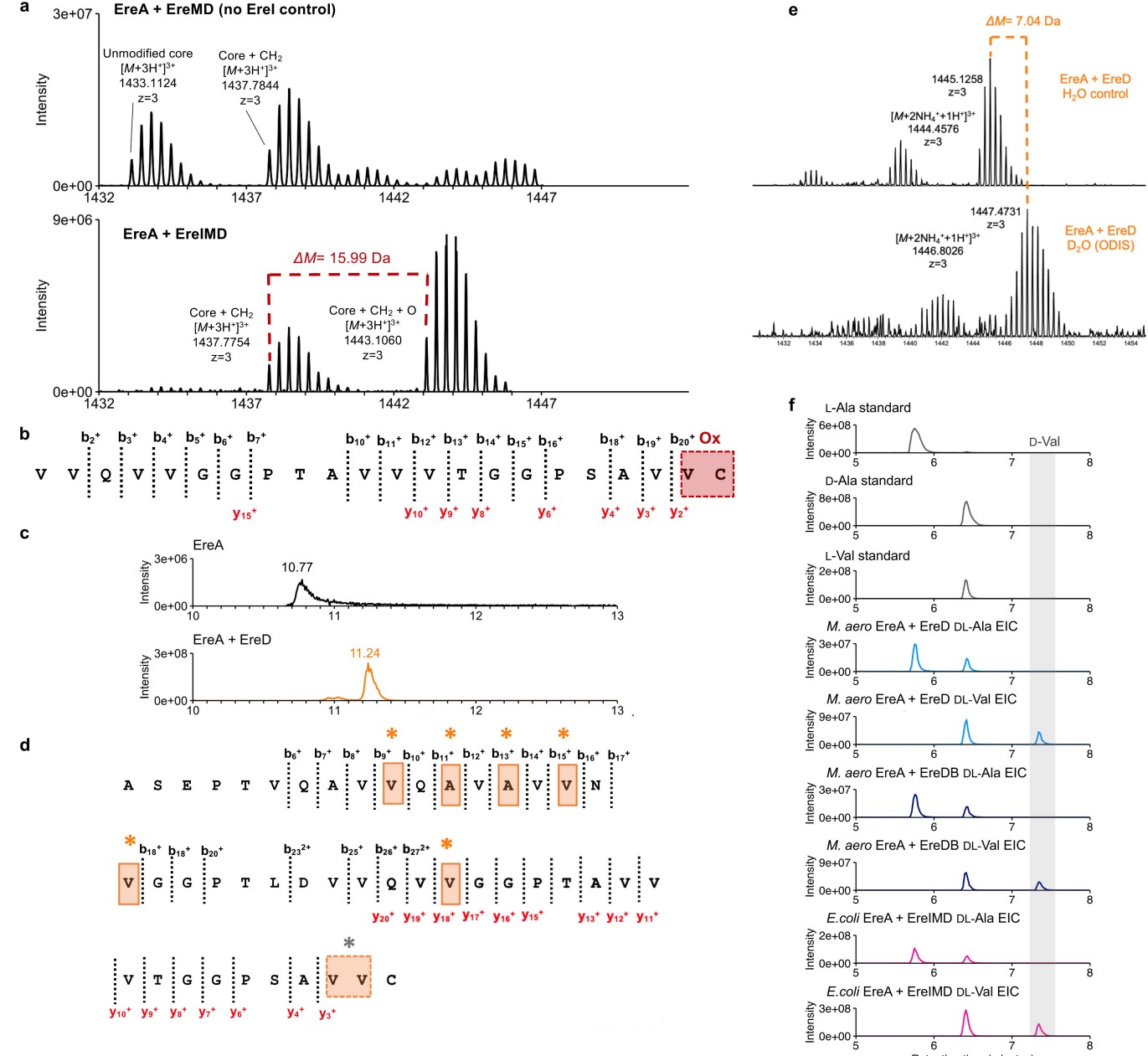

**Extended Data Fig. 11 | EreI mass shift and high resolution tandem mass spectroscopy (MS²); EreD retention time shift and ODIS and advanced Marfey's analysis for D-Val/D-Ala. (a)** Mass spectra and MS²-fragmentation of the LahT-digested Nhis-EreA modified by EreI, EreM, and EreD. Bottom: A mass shift of +15.99 Da corresponding to incorporation of one oxygen into the mono-methylated core (EreA + EreIMD, [M+3H⁺]³⁺ = 1443.1060 Da) was observed after co-expressions of *ereAIMD*. Top: No oxygen incorporation was observed in Nhis-EreA modified by EreM and EreD (EreA + EreMD) controls lacking the aspartinyl-asparaginyl β-hydroxylase protein family protein, EreI. Notably, the +15.99 Da modification was only observed on methylated, LahT-released EreA cores ([M+3H⁺]³⁺ = 1437.7754 Da) and not on the non-methylated core ([M+3H⁺]³⁺ = 1433.1124 Da) as observed by *in vivo ereAI* or *ereAID* co-expressions and *in vitro* assays with purified NHis-EreI and Nhis-EreA or Nhis-EreA modified by EreD. **(b)** MS²-fragmentation of [M+3H⁺]³⁺ = 1443.1060 Da. The data localize oxygen (Ox) incorporation to the C-terminus of the peptide but cannot distinguish between terminal cysteine (C46) or valine (V45). MS²-fragmentation data including calculated and observed masses for all b- and y- ions are available in Supplementary Table 5. **(c)** Extracted ion chromatograms (EICs) at 1433.1123 Da of LahT-digested precursors Nhis-EreA (black trace, top) and epimerized

Nhis-EreA from co-productions with radical SAM epimerase EreD (orange trace, bottom), which show a retention time shift of 0.47 min. No mass shift was observed, since L- to D- amino acid epimerization is a mass-neutral modification, requiring the use of the orthogonal D₂O induction systems (ODIS) to localize epimerization sites. **(d)** MS²-fragmentation of m/z = 1447.47308 Da patterns enabled localization of epimerized residues to V10, A12, A14, V16, V18, V29 (orange asterisks) and either V44 or V45 (grey asterisk). MS²-fragmentation data are available in Supplementary Table 5. **(e)** Expressions using ODIS results in a shift of +7.04 Da corresponding to the incorporation of 7 deuterium atoms. A mixture of products is observed due to slowdown of epimerization in the presence of deuterium. **(f)** EICs from advanced Marfey's analysis of epimerized and modified cores from two heterologous hosts: *E. coli* (pink) and *M. aerodenitrificans* (light and dark blue) consistently yielded D-Val and D-Ala (grey shading) as the only D-amino acids detected in EreA cores, as compared to D-Thr, D-Asp, and D-Ser standards (not shown), for which no corresponding peaks were detected. Both D-Val and D-Ala were measured in ratios of 1:3 to their L-amino acid counterparts. Based on EreA core amino acid composition, these ratios correspond to approximately 2 D-Ala and 5 D-Val per core consistent with ODIS results.

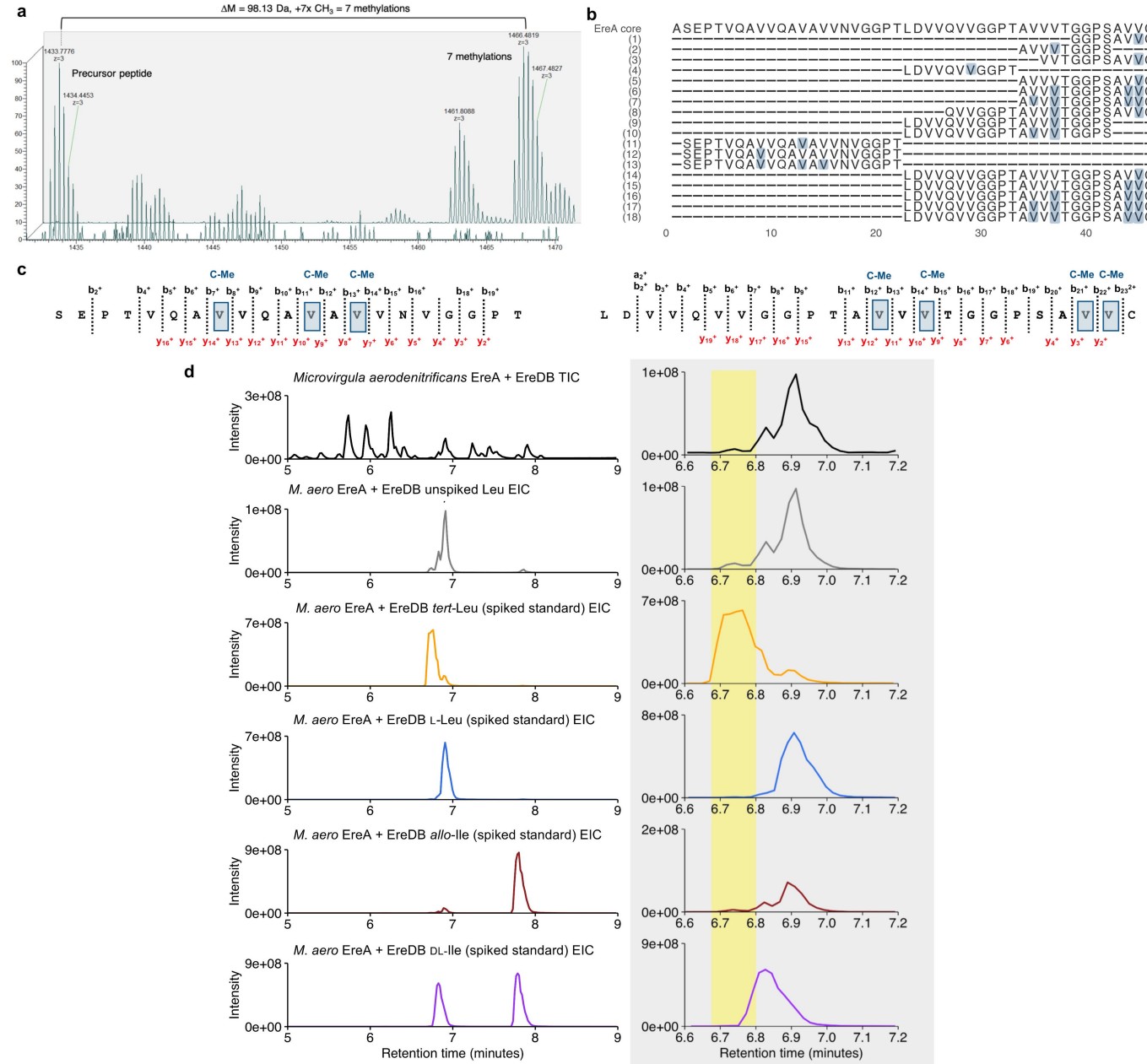

**Extended Data Fig. 12 | EreB mass shift, MS²-fragmentation data and advanced Marfey's analysis for *tert*-Leu. (a)** Co-production of Nhis-EreA with epimerase EreD and the B₁₂-dependent radical SAM *C*-methyltransferase EreB in *M. aerodenitrificans Δaer* with a knocked-out aeronamide BGC yielded a mixture of *C*-methylated products with mass shifts corresponding to up to 7 methylations (+98.13 Da). **(b)** Alignment of a mixture of differently-modified fragments detected by MaxQuant analysis of proteinase K digested Nhis-EreA following co-productions with EreDB. **(c)** Representative MS²-fragmentation of EreA core following co-production with EreDB at *m/z* = 1466.4819 Da. Observed

and calculated masses for b- and y-ions are in Supplementary Table 5. Modification sites (dark blue, C-Me) were localized to V9, V13, V15, V35, V37, V44, and V45. **(d)** Total ion chromatogram (TIC, black) and EICs from advanced Marfey's analysis of C-methylated core from co-productions of EreA + EreDB in *M. aerodenitrificans*. The unspiked sample (dark grey) is compared to identical samples that were spiked with synthetic standards: *tert*-Leu (orange), L-Leu (blue), *allo*-Ile (brown), and DL-Ile (purple). The grey box is an inset on a narrower retention time from the left panel highlighting a peak shoulder from the *M. aerodentrificans* EIC corresponding to *tert*-Leu (yellow shading).

# Reporting Summary

## Statistics

For all statistical analyses, confirm that the following items are present in the figure legend, table legend, main text, or Methods section.

| n/a | Confirmed | |
|---|---|---|
| ☐ | ☒ | The exact sample size (*n*) for each experimental group/condition, given as a discrete number and unit of measurement |
| ☐ | ☒ | A statement on whether measurements were taken from distinct samples or whether the same sample was measured repeatedly |
| ☐ | ☒ | The statistical test(s) used AND whether they are one- or two-sided *Only common tests should be described solely by name; describe more complex techniques in the Methods section.* |
| ☒ | ☐ | A description of all covariates tested |
| ☐ | ☒ | A description of any assumptions or corrections, such as tests of normality and adjustment for multiple comparisons |
| ☐ | ☒ | A full description of the statistical parameters including central tendency (e.g. means) or other basic estimates (e.g. regression coefficient) AND variation (e.g. standard deviation) or associated estimates of uncertainty (e.g. confidence intervals) |
| ☐ | ☒ | For null hypothesis testing, the test statistic (e.g. *F*, *t*, *r*) with confidence intervals, effect sizes, degrees of freedom and *P* value noted *Give P values as exact values whenever suitable.* |
| ☒ | ☐ | For Bayesian analysis, information on the choice of priors and Markov chain Monte Carlo settings |
| ☒ | ☐ | For hierarchical and complex designs, identification of the appropriate level for tests and full reporting of outcomes |
| ☐ | ☒ | Estimates of effect sizes (e.g. Cohen's *d*, Pearson's *r*), indicating how they were calculated |

*Our web collection on statistics for biologists contains articles on many of the points above.*

## Software and code

Policy information about availability of computer code

| Data collection | Data collected from ENA was downloaded from ENA using their API. All other publicly available data was downloaded directly from the sources specified in the data availability statement. |
|---|---|
| Data analysis | The data was analyzed using a pipeline developed as part of this study as well as with additional ad hoc python and R scripts and with the following softwares: BBMap v.38.71, metaSPAdes v3.11.1 and v3.12, BWA v0.7.17-r1188, MetaBAT2 v2.12.1, CheckM v1.0.13, Anvi'o v5.5.0, dRep v2.5.4, SpecI, GTDB-Tk v1.0.2, Prokka v1.14.5, fetchMGs v1.2, emapper v2.0.1, DIAMOND v0.9.30, antiSMASH v5.1.0 and v5.0.0, CD-HIT v4.8.1, mOTUs v2.5.1, BiG-SLICE v1.1, IQTREE v2.0.3, MUSCLE v3.8.1551, trimal v1.4.1, Traitar v1.1.2, TXSSCAN v1.0.2, FeatureCounts v2.0.1, PlsamidFinder v2.1, PlasFlow v1.1.0, cBar v1.2, VirSorter v1.0.5, DeepVirFinder v1.0, EukRep v0.6.6, ccontigs v1.0.0, STAG v0.7, GECCO v0.4.4, MMSEQS2 v13.45111, MAFFT v7.310, Python >= 3.6 with the packages pandas (v1.0.0-1.3.4), biopython (v1.73), umap-learn (v0.5.2), hdbscan (v0.8.28), scikit-learn (v1.0.2) and R (v4.0.0-v4.1.2) with the packages ggplot2 (v3.3.0-v3.3.5), tidyverse (v1.3.1), vegan (v2.5.7), ggtree (v3.3.0.901) and tidytree (v0.3.6), treeio (v1.19.1) and UpSetR (v1.4.0). The code used in this study is accessible at https://github.com/SushiLab/magpipe/ and archived at Zenodo (https://doi.org/10.5281/zenodo.6393817). |

For manuscripts utilizing custom algorithms or software that are central to the research but not yet described in published literature, software must be made available to editors and reviewers. We strongly encourage code deposition in a community repository (e.g. GitHub). See the Nature Portfolio guidelines for submitting code & software for further information.

## Data

Policy information about availability of data

All manuscripts must include a data availability statement. This statement should provide the following information, where applicable:

- Accession codes, unique identifiers, or web links for publicly available datasets
- A description of any restrictions on data availability
- For clinical datasets or third party data, please ensure that the statement adheres to our policy

The metagenomic and metatranscriptomic data used in this study was downloaded from the European Nucleotide Archive (ENA) and their accessions are summarized in Supplementary Table 1. Publicly available genomes were downloaded from https://doi.org/10.6084/m9.figshare.4902923 for manually curated MAGs from Tara Oceans, from ENA using the project accession PRJEB33281 for GORG and from https://mmp2.sfb.uit.no/databases/ for MarDB. The GEM MAGs were downloaded from https://portal.nersc.gov/GEM/. MAGs contained in the GTDB r89 were downloaded from https://data.gtdb.ecogenomic.org/releases/release89/. The MIBiG and BiG-FAM databases can be accessed at https://mibig.secondarymetabolites.org/ and https://bigfam.bioinformatics.nl/, respectively. The data produced in this study, including metagenomic assemblies, bins and MAGs have been deposited at the European Nucleotide Archive under the accession PRJEB45951 and individual accessions are summarized in Supplementary Table 1. Other supporting data has been deposited on Zenodo (https://doi.org/10.5281/zenodo.4474310), and the OMD can be interactively accessed at https://microbiomics.io/ocean/. Additional material generated in this study is available upon request.

# Field-specific reporting

Please select the one below that is the best fit for your research. If you are not sure, read the appropriate sections before making your selection.

☒ Life sciences      ☐ Behavioural & social sciences      ☐ Ecological, evolutionary & environmental sciences

For a reference copy of the document with all sections, see nature.com/documents/nr-reporting-summary-flat.pdf

# Life sciences study design

All studies must disclose on these points even when the disclosure is negative.

| | |
|---|---|
| Sample size | Samples sizes were defined by the availability of published data that were used to perform the analyses. |
| Data exclusions | No data were excluded from the analyses. |
| Replication | All post-translational modification of the peptides reported in this study were supported by replicated experiments, bio-activity assays included replicates (n >= 3) and all replicates were successful. |
| Randomization | For the different analyses conducted in this study, all samples were processed similarly and thus randomization was not necessary. |
| Blinding | For the different analyses conducted in this study, all samples were processed similarly and thus blinding was not necessary. |

# Reporting for specific materials, systems and methods

We require information from authors about some types of materials, experimental systems and methods used in many studies. Here, indicate whether each material, system or method listed is relevant to your study. If you are not sure if a list item applies to your research, read the appropriate section before selecting a response.

### Materials & experimental systems

| n/a | Involved in the study |
|---|---|
| ☒ | ☐ Antibodies |
| ☐ | ☒ Eukaryotic cell lines |
| ☒ | ☐ Palaeontology and archaeology |
| ☒ | ☐ Animals and other organisms |
| ☒ | ☐ Human research participants |
| ☒ | ☐ Clinical data |
| ☒ | ☐ Dual use research of concern |

### Methods

| n/a | Involved in the study |
|---|---|
| ☒ | ☐ ChIP-seq |
| ☒ | ☐ Flow cytometry |
| ☒ | ☐ MRI-based neuroimaging |

## Eukaryotic cell lines

Policy information about cell lines

| | |
|---|---|
| Cell line source(s) | HeLa cells. |

| Authentication | Not authenticated. |
| Mycoplasma contamination | Not tested. |
| Commonly misidentified lines<br>(See ICLAC register) | Not applicable. |

