## [Peer Review File · Nature]

Manuscript Title: Biosynthetic potential of the global ocean microbiome.

Reviewer Comments & Author Rebuttals

Reviewer Reports on the Initial Version:

Referees' comments:

Referee #1 (Remarks to the Author):

In this manuscript, Paoli et al describe a large resource for MAGs and BGCs from the open Ocean microbiome. This is leveraging previously acquired metagenomic and metatranscriptomic datasets, re-processed in this new analysis to produce MAGs and BGCs, as well as reference genomes and single-cell genomes from other studies. In my opinion, this resource will be useful to the scientific communities of marine microbiologists and natural product chemists and biochemists. The analysis employs various bioinformatics techniques and reflects a set of rigorous analyses, for which the authors are to be commended. Having said that, I do have several points of concerns that need to be addressed.

1. I find any discussion related to genome sizes of MAGs to be overstated. We really don't know how accurate the composed MAGs are, and extrapolating results based on estimated MAG genome sizes is somewhat misleading – from both ends: contamination and completeness. The cutoffs set for overall MAG completeness and contamination seem to be $\geq 50\%$ and $\leq 10\%$, which is a wide range (knowing that even these metrics are not necessarily accurate). One example of this is in the stated genome size difference between the clusters in Figure 2D and the unexpected large genomes in the <0.2 μm particle size samples – this could all be an artifact of imperfect binning.
2. I did not see any consideration of sequencing depth when accounting for diversity, abundance or even in the clustering. The authors are collecting samples processed by different groups and expeditions, and while correcting for batch effects (sample processing, collection method, DNA extraction methods, etc.) in such study is basically impossible, the authors should at least correct for sequencing depth and measure if it has an effect on the detection of MAGs and BGCs in a given sample (e.g., by correlating sequencing depth with number of detected BGCs per sample) and on the clustering outcome in Figure 2C. Also, it appears from the Methods that the authors downsampled when mapping reads, what is the rationale behind this?
3. Regarding the clustering in Figure 2C as well, how does that compare to a clustering analysis that is based on taxonomic profiles of the samples (mOTU profiles for example)? Do we see the same clusters or do GCFs provide a different view? Do GCFs give us any new information? Can we predict GCF profiles from taxonomic profiles of samples?
4. Can the authors estimate the completeness of the discovered BGCs? Perhaps by calculating the proportion of which that has housekeeping genes at the boundaries?
5. Since the authors mention that Ca. Eudoremicrobiaceae can be found more in eukaryote-enriched samples, why are they not considering a potential symbiotic state (with small eukaryotes, like diatoms or dinoflagellates) for species from this family?
6. Similar to point 2 above, I did not see consideration of sequencing depth when analyzing the metatranscriptomic data.
7. It would be super useful for the community if the metatranscriptomic analysis is expanded to all MAGs and BGCs and not only Ca. E. taraoceanii. If metatranscriptomic information is tied with the

genomes and BGCs on the web portal (and in supplementary information), it would certainly help interested scientists in using this resource to pick additional BGCs for characterization or genomes to study further.

8. Related to the previous points, what fraction of *Ca. E. taraoceanii* genes is typically covered by metatranscriptomic reads before performing the UMAP/DBSCAN? If the data here is sparse, meaning that only a small number of genes are found to be expressed under each condition, then the clustering can be less meaningful – similar to the issue encountered in eukaryotic single cell transcriptomic analyses. I do not think that the transcriptomic clustering in Figure 4D is providing any ecological insights (see point below),

9. In my opinion, the section named “putative ecology of *Ca. Eudoremicrobium* spp.” is the least supported section of the manuscript and the manuscript will be stronger without it. Inferring ecology simply by looking at genomes and transcriptomes and without any biological context is guessing at best – especially when the genome is also not 100% trusted (again, because of potential issues with binning). The statement about the two states of lifestyle is not supported, the statement about niche competition using natural products is not supported, etc. It is totally fine to call it what it is: a new species of marine bacteria, widespread in the Oceans (I suggest presenting a main figure piece that reflects this), many of its genes are actively transcribed (see point 7 above), and is biosynthetically rich. Beyond that is mostly mere speculation, especially when it is presented as “ecology”.

10. Related to the previous point, the expansion of the “ecology” section came at the expense of the natural product discovery/enzymology section – which has the only concrete experimental results in the paper, represents a tremendous effort and is beautifully and thoroughly explained in the supplementary text. In my opinion, this section should be expanded in the main text – honestly, it can make its own (great) paper. More discussion about the characterized BGCs and the clever methods used for characterization can definitely be promoted to the main text.

11. Line 338, how confident are the authors that these point mutations in the dehydration catalytic site are real and not an artifact of metagenomic assembly? Are they seen over and over in different samples?

Referee #2 (Remarks to the Author):

This study by Paoli et al uses a large-scale MAG based effort to describe the biosynthetic potential of the open ocean microbiome. There are two parts to this manuscript - 1. Construction of a comprehensive genome database termed as the Ocean Microbiomics Database that encompasses microbial genomes aggregated from a wide variety of source datasets, 2. Scanning the Ocean Microbiomics database for biosynthetic gene clusters (BGCs), profiling the activity of these identified clusters, and characterizing some of them using heterologous expression. Overall, there is a lot to like about this paper. The scope of the paper is significantly large, the identification of the BGCs and the expansion of clusters is also significant. This will be a good resource that can be used to survey for important natural products in the coming years. However, I also have some important criticisms of this work that may preclude its widespread use and adoption. Specifically, the two parts of the study (1) and (2) listed above seem to have been conducted with different levels of rigor. While (2) is solid and important work in my opinion, (1) seems exaggerated and I am unable to place its importance in context. While I do understand that the authors needed to place the BGCs in a global ocean context (hence the rationale for an Ocean Microbiomics Database), nevertheless there seem to be significant shortcomings here.

Overall, I think this is an exciting study that I am very happy to have read. With changes this would be an important contribution to the fields of microbial ecology and natural products going forward.

Major concerns

Repeatedly (including in the abstract) the authors mention that they reconstructed ~25000 genomes. This is not true as stated in the methods. Rather these genomes were aggregated from multiple studies. It is also unclear to me as to how much (if any overlap) exists between the other Tara Ocean MAG studies (such as with Delmont et al). Presumably, all of these studies are using the Tara ocean read datasets to assemble the genomes. While I do understand that this would be taken care of during dereplication – it is not clear if genomes from other datasets were also included in such analyses. As such, the total number of genomes used sounds large but I am not sure if it is relevant or accurate.

Along similar lines, it is mentioned that the genomes come from different size fractions including virus-enriched and also some eukaryote genomes. Why is this relevant? Were any BGCs identified from these fractions/genomes? Fig 2A and other parts of Fig 2 does show some of these details but the figure is complex to interpret and the colors associated with the size fractions are difficult to separate.

I have some concerns with regard to novelty of these BGCs. Have these new BGCs never been described before? The authors compare the BGCs to two recently developed databases mibig and bigfam – I think these analyses are great. But what about the context of BGCs in the open ocean when compared to other environments – it was my understanding that the authors also set out to do this? I also bring this up specifically because the authors mention *Eremiobacterota* as being BGC rich and only being previously found in terrestrial environments. Given the enormous types, range, and number of BGCs discovered, I think this could be useful analyses to add and provide context for BGCs in the oceans.

Data is open but practically unusable. I commend the authors for creating a website for OMD – the website interface is clean and easily interpretable. However, I what I find to be unacceptable is that only the assemblies have been deposited in NCBI/ENI. Even today, few groups have the resources to download and run analyses on large datasets such as this. The MAGs and BGCs will essentially be unsearchable and unusable at a broad scale if they are NOT deposited in these databases. I encourage the authors to deposit the MAGs and proteins in NCBI/ENI. Without this, I can imagine another group coming along in the coming years and describing novelty based on this data.

Methods

The methods for determination of BGCs, classification, and expression are well done. I have no concerns here.

The idea to not use individual assemblies in lieu of coassemblies was an excellent choice.

I mentioned having criticisms about the development of the MAG database above. Along those lines, the description of binning in the manuscript is shockingly elementary. The authors mention using abundance correlation to improve the recovery of MAGs. This has been known for the best part of 10 years and there are at the very least 10+ bidders and countless studies that use differential abundance coverage binning (See papers on Metabat, Metabat2, Maxbin, Concoct, Abawaca, Cocacola etc). These findings are not new or novel. While I appreciate using this form of binning, I am unsure if any other form of binning would even be acceptable in the field as things stand.

Binning – I am also unsure as to why only one bidder (Metabat2) was used in this study. Considering the authors have benchmarked their analyses (I commend them for this), I would have imagined that the use of multiple bidders followed by aggregation using tools such as (Dastool, Metawrap etc) would have performed better. Alternatively given that the approaches

described above are “more standard” in the field, I would have liked to see some justification as to why the authors preferred to do it their way. This would also justify as to why their approach is better than what the authors define as automatically curated MAGs (given that those MAGs were also identified using differential abundance binning – albeit with different binners and also include contamination analyses).

Minor concerns

Line 135: state this to be the OMD.

Referee #3 (Remarks to the Author):

Paoli and colleagues report an extensive data analysis of the global ocean microbiome that focuses on the biosynthetic potential of marine bacteria to produce bioactive compounds. The authors assembled >25k MAGs and identified nearly twice as many biosynthetic gene clusters (BGCs), many from new and unsuspected species. The massive scale of their work is reminiscent of several recent high-profile articles published in Nature 2018 on soil microbes (ref 18) and Nat Biotech 2021 on the Earth microbiome (ref 19) that also detailed the biosynthetic potential of newly identified bacterial groups. Like those recent reports, the authors identified new bacterial species from their respective environments that were biosynthetically “gifted” in terms of encoding diverse and plentiful BGCs.

The Paoli paper deviates from those other publications in one important way. While the others were primarily computational, Paoli additionally provides experimental data to substantiate their biosynthetic claims. The authors identified two BGCs encoding ribosomally-encoded and post-translationally modified peptides from a new BGC-rich actinobacterial lineage belonging to an uncultivated phylum. Through a series of rigorous in vivo and in vitro expression experiments and chemical analyses, they report the production of two complex peptide molecules that are new and unusual. Both molecules as reported in Fig 4 have structure features not well represented in the biosynthetic literature, although their structure features, namely amine methylation and phosphorylation, are ubiquitous in the natural product literature. The authors should be commended for their impressive biochemical work, yet they should tone down their claims that split hairs. The work is impressive by itself yet is distracted with unneeded posturing.

Overall, I was impressed with the sheer volume of data presented and the identification of a new biosynthetically rich marine lineage, *Ca. Eudoremicrobium* spp.

The paper is purely discovery based looking for something to discover without a clear hypothesis or motivation. Broadly speaking, the result is largely anticipated based on recent large-scale microbiome studies that new lineages will be discovered when you look hard enough in the global landscape, and that they will harbor new BGCs encoding new molecules that may support their specific lifestyles. The authors should more clearly clarify their motivations and how this report differs from other recent large scale microbiome studies.

Below are several items for the authors to additionally consider in a revision if encouraged.

1. Prior to this work, Tully et al reported in 2018 in Sci Data over 2.6k MAGs from the global ocean also using data collected by the Tara Oceans expedition that seemingly provided data for this paper’s analysis. The Tully paper is not cited nor discussed, yet there may appear to be redundancy. This should be clarified.

2. The authors suggest that the expression of BGCs by *Ca. E. taraoceanii* implies “a niche-partitioning role for the candidate natural products they encode”. That statement could be said

frankly of any bacterium, especially those that encode specialized metabolites. The challenge with the two candidate natural products reported in this manuscript is that the compounds may not be physiologically relevant due to other biosynthetic processes not captured by the heterologous expression experiments. Moreover, the structures are suggested based on MS data and biosynthetic precedence, yet rigorous, independent analyses based on NMR are missing.

3. The biological data showing that the diphosphorylated peptide is active against elastase but not other proteases may be an in vitro artefact and not biologically relevant as phosphorylated peptides will generally not be cell penetrant, perhaps explaining the lack of activity in a whole cell assay. This may suggest that the compound described may not be the biosynthetic product of the bacterium.

4. Is there a reason why the authors focused on the expression of the two specific RIPP BGCs, other than one of the corresponding authors being an expert in this BGC type? Please explain and justify. If for a "biotechnological interest" (line 318), then a reference beyond a self-reference is preferred that shows genuine interest.

5. The authors discuss that they queried a number of unique terpenoid BGCs, yet they report common hopanoids and carotenoids. Please expand.

Point-by-point response to reviewers

Reviewer #1: multi-omics, small molecule chemistry and biosynthesis

General

In this manuscript, Paoli et al describe a large resource for MAGs and BGCs from the open Ocean microbiome. This is leveraging previously acquired metagenomic and metatranscriptomic datasets, re-processed in this new analysis to produce MAGs and BGCs, as well as reference genomes and single-cell genomes from other studies. In my opinion, this resource will be useful to the scientific communities of marine microbiologists and natural product chemists and biochemists. The analysis employs various bioinformatics techniques and reflects a set of rigorous analyses, for which the authors are to be commended.

We would like to thank the reviewer for the positive assessment on the usefulness of the resource and rigor of performed analyses, which resonates well with the view from the other reviewers. We are very grateful for the constructive comments on several bioinformatic aspects and the suggestion to tone-down the putative ecology section to allow for bringing the biochemical analyses more to the foreground. The comments prompted us to perform additional bioinformatic analyses, generate new supporting genomic data, improve the functionality of the web-resource, and run confirmatory experiments to further corroborate the structure of the natural product we report. In the following, we provide detailed responses to each comment and feel that the actions we took to address them have significantly improved the manuscript.

Having said that, I do have several points of concerns that need to be addressed.

Comment #1

1. I find any discussion related to genome sizes of MAGs to be overstated. We really don't know how accurate the composed MAGs are, and extrapolating results based on estimated MAG genome sizes is somewhat misleading – from both ends: contamination and completeness. The cutoffs set for overall MAG completeness and contamination seem to be $\geq 50\%$ and $\leq 10\%$, which is a wide range (knowing that even these metrics are not necessarily accurate). One example of this is in the stated genome size difference between the clusters in Figure 2D and the unexpected large genomes in the <0.2 μm particle size samples – this could all be an artifact of imperfect binning.

We acknowledge the limitations that arise from working with MAGs, and that incompleteness and contamination may lead to uncertainties in predicted genome sizes. We would like to clarify that we took these potential uncertainties into account using a completeness-corrected estimator, which we found to perform well (Figure R1.1.1). However, considering (1) that communicating results based on such predicted genomes sizes would require a more in-depth discussion of the limitations and (2) that, in light of other comments from this and other reviewers, such a methodological discussion would distract from the main results of this manuscript, we decided to provide the relevant results as

supplementary Information along with new analyses to assess the level of uncertainty in the predicted genome sizes and a discussion on the limitation of this approach.

ACTIONS TAKEN:

- We moved the results based on genome size estimates from the main manuscript to the supplementary Information (*'Estimating genome sizes'*).
- We added a discussion on the limitations of this approach.
- We produced a Supplementary Figure (see below) to quantify the accuracy of genome size estimates based on MAGs (after accounting for completeness).

Figure R1.1.1. Included in the supplementary Information as Figure S11B. Comparing genome sizes from MAG-based predictions and reference genomes for 85 mOTUs (species-level) clusters with at least one reference genome. Genome sizes are estimated using MAGs of good quality and above only (completeness above 70%), a criterion that is met for >80% of the mOTUs clusters.

Comment #2

2. I did not see any consideration of sequencing depth when accounting for diversity, abundance or even in the clustering. The authors are collecting samples processed by different groups and expeditions, and while correcting for batch effects (sample processing, collection method, DNA extraction methods, etc.) in such study is basically impossible, the authors should at least correct for sequencing depth and measure if it has an effect on the detection of MAGs and BGCs in a given sample (e.g., by correlating sequencing depth with number of detected BGCs per sample) and on the clustering outcome in Figure 2C. Also, it appears from the Methods that the authors downsampled when mapping reads, what is the rationale behind this?

We value this thorough feedback and welcome the appreciation for the difficulty of working with a composite dataset.

First, we would like to clarify that sequencing depth is taken into account in our analyses of metagenomic and metatranscriptomic data through normalization by the abundance of universal single-copy marker genes (uscMGs). Specifically, the abundance profiles were normalised by mOTUs counts, which represent the number of uscMG copies in a sample and can therefore be used as a proxy for cell counts (Sunagawa et al. 2013; Milanese et al. 2019). These uscMGs-normalised profiles are thus sequencing depth normalised (normalised by the number of cells sequenced) (Figure R1.2.1), and also account for genome size differences (Beszteri et al. 2010). We, like others (Salazar et al. 2019; Nayfach et al. 2016) have previously applied this normalisation strategy to metagenomic data to facilitate comparative analyses, as well as to integrate metagenomic and metatranscriptomic data (explained in detail in Salazar et al., 2019).

Second, we agree with the reviewer that sequencing depth may impact the number of detected MAGs and BGCs in a sample. To address this concern, we normalized (rarefied) our data when comparing alpha-diversity metrics, for the only relevant analysis in our manuscript (i.e. the number of GCFs detected per sample in now figure S6).

Third, we would like to clarify that sequencing depth was indeed taken into account by the aforementioned uscMG-based normalisation for our distance-based analysis resulting in the clustering shown in Figure 2C (now Figure S5). In addition, we show in figure R1.2.1 that the three different clusters harbour very similar sequencing depths with medians ranging between 6,600-7,350 total mOTUs count (min 59, max 167,931) suggesting that sequencing depth differences are not impacting the clustering outcome.

We reformulated the way these parts are described in the methods to clarify this point.

Finally, regarding the downsampling of metagenomic reads before mapping, we would like to clarify that we only downsampled metagenomes to 5M reads to compare the mapping rates to different databases used as a reference in Figure 1D (now 1C). The rationale for this was to lower the computational burden for an almost identical outcome. This method was previously used in Pachiadaki et al. (2019) and we adopted it after conducting tests on our own (Figure R1.2.3). For all remaining analyses using the OMD as a reference database, sequencing reads were not downsampled, whereas the full profiles were normalized for sequencing-depth as described above.

ACTIONS TAKEN:

- We included an additional figure (see figure R1.2.1) to demonstrate that the total mOTUs accounts for the sequencing effort through a proxy of the number of bacterial and archaeal cells sequenced.
- We clarified the relevant sections of the methods ('Abundance and prevalence of GCFs and GCCs' and 'Transcriptomic profiling of *Ca. E. taraoceanii*') to explicitly state that the mOTUs count normalisation accounts for the sequencing effort.
- We moved some of the results (Figure 2C and D and the corresponding text) to the supplementaries (Figure S5) to allow for a more in depth discussion of their limitations. Including, for instance, analyses to support the clustering stability and robustness.

Figure R1.2.1. Included in the supplementary Information as Figure S11B. mOTUs counts as a good proxy for sequencing depth. We find a strong correlation in prokaryote-, particle-enriched and virus-depleted communities, while this correlation is more variable in virus-enriched communities. Indeed, deviations between the sequencing depth and the number of sequenced cells may occur due when the total sequenced DNA is expected to be dominated by e.g., viral DNA. This observation is actually in support of using the mOTUs count rather than sequencing depth when focusing on the bacteria and archaeal component of microbial communities, as we do here.

Figure R1.2.2. To investigate a potential of sequencing depth (as captured by the mOTUs count, a proxy for the number of cells sequenced) we looked at the distribution of mOTUs counts across the clusters. Despite variable distributions across the three clusters, we found the medians (6,600-7,350) and means (16,500-25,700) of total motus counts to be similar across clusters (min 59; max 167,931) suggesting that the clustering is not an artefact of uneven sequencing depths.

Effect of subsampling on mapping rates (%) to the OMD
Based on 1,028 metagenomic samples with sequencing depth >5M reads

Figure R1.2.3. Included in the supplementary Information as Figure S11A. Comparing mapping rates obtained from mapping subsampled readsets compared to those obtained from mapping the total number of reads shows that this procedure yields almost identical results at considerably less computational costs.

Comment #3

3. Regarding the clustering in Figure 2C as well, how does that compare to a clustering analysis that is based on taxonomic profiles of the samples (mOTU profiles for example)? Do we see the same clusters or do GCFs provide a different view? Do GCFs give us any new information? Can we predict GCF profiles from taxonomic profiles of samples?

We thank the reviewer for this valuable and constructive feedback.

We agree these are interesting questions and therefore investigated the relationship between taxonomic (mOTU) and GCFs profiles. Interestingly, we found the taxonomic and GCF-based distances to be only weakly correlated (Figure R1.3.1). We further clustered the taxonomic profiles using the same method as for the GCF profiles and found the resulting clustering to relate, but only partially, to the GCF-based clustering (V-measure ≈ 0.4). As the V-measure is similar to the F1-score, it also reflects the ability to predict one clustering based on the other, which appears to be low in this case. To contextualize this value, we estimate this difference to correspond to assigning about a third of the samples from the GCF-profiles randomly, suggesting that albeit related, the GCF-based clustering provides complementary and only partially predictable information to a taxonomic composition-based structuring of the ocean microbiome.

ACTIONS TAKEN:

- We added an analysis on the relationship between taxonomic and GCF-based profiles as a supplementary discussion point in 'Biosynthetic potential of the ocean microbiome'.

Figure R1.3.1. Correlating GCF-based and taxonomic distances shows a significant albeit low correlation, suggesting that the biosynthetic potential captures additional information compared to the taxonomic composition of a sample.

Comment #4

4. Can the authors estimate the completeness of the discovered BGCs? Perhaps by calculating the proportion of which that has housekeeping genes at the boundaries?

We thank the reviewer for mentioning the concern about incomplete BGCs in metagenomic samples, which are common and we agree should be discussed. Although the assessment of BGC completeness is a challenge in the field by itself (personal communication with Kai Blin, Marnix Medema and others during a recent workshop), we acknowledge that our original presentation lacked sufficient detail in communicating this challenge and how the presented data in our work might be impacted.

To improve clarity, we extended the description of methods that were taken to reduce the fraction of incomplete BGCs and explored the consequences of BGC incompleteness on the clustering into GCFs and GCCs. We also explored different completeness estimates showing that most BGC classes had similar lengths and a similar or higher number of genes compared to characterized pathways. Only NRPSs and PKSs were significantly shorter although the number of genes we found were higher than those present in the MIBiG database (Figure R1.4.1). We also investigated whether a machine-learning based approach (GECCO, Supplementary Information) could provide additional completeness information, and found that among the complete GECCO BGCs overlapping with antiSMASH predicted BGCs, 13% were flagged as incomplete by antiSMASH.

Furthermore, we (a) looked at antiSMASH predictions for BGC completeness and (b) evaluated the impact of incomplete BGCs on the number of GCFs and GCCs. Re (a), we would like to note that the criterion for antiSMASH to predict BGC completeness is solely based on the presence of flanking regions of a given size depending on the BGC class, which may be a reasonable criterion to identify complete BGCs for isolate genomes, but does not de facto imply incompleteness and may be excessive for metagenomic data. We nevertheless used these predictions and found that incomplete BGCs did not have a significantly lower dereplication rate (*i.e.*, resulting in a proportionally higher number of GCFs) compared to complete BGCs. The number of resulting GCCs after dereplication was almost identical.

ACTIONS TAKEN:

- We added a section '*Assessing the completeness of predicted BGCs*' in the Supplementary Information as well as figure S15 to reflect this point.
- We extended descriptions in the Methods to explain that we used a minimum contig length cutoff of 5 kbp for antiSMASH predictions to reduce the fraction of incomplete BGCs as suggested in previous work (Nayfach et al. 2020):
"BGCs were subsequently filtered, keeping only the ones encoded on scaffolds ≥ 5 kbp to reduce the risk of fragmentation, as done previously (Nayfach et al. 2020)"
- We rephrased the relevant text passage to clarify that the clustering step served, in addition to removing redundancy, as a measure to account for and to reduce the number of fragmented BGCs, along with the fractions of GCFs and GCCs with at least one complete member.

Original text:

To this end, we first used antiSMASH to predict a total of 39,055 BGCs on all MAGs, SAGs and REFs detected in the set of 1,038 ocean metagenomes (Methods). Owing to the inherent redundancy of BGCs (i.e., the same BGC can be encoded in several genomes), we further clustered them into 6,907 non-redundant gene cluster families (GCFs) and 151 gene cluster clans (GCCs) (Table S2, Methods).

Revised text:

To this end, we first used antiSMASH on all MAGs, SAGs and REFs detected in the set of 1,038 ocean metagenomes (Methods) to predict a total of 39,055 BGCs. We then clustered them into 6,907 non-redundant gene cluster families (GCFs) and 151 gene cluster clans (GCCs, Table S2, Methods) to account for inherent redundancy (i.e., the same BGC can be encoded in several genomes) and fragmentation of BGCs in metagenomic datasets. Incomplete BGCs did not significantly inflate, if at all (Supplementary Information), the number of GCFs and GCCs, which contained at least one complete member BGC in 44% and 86% of the cases, respectively.

Figure R1.4.1. Included in the supplementary Information as Figure S15. BGC length distributions across BGC classes are not significantly different between the set of BGCs studied in this work (antiSMASH) and the characterized BGCs in MIBiG with the exception of the polyketides and non-ribosomal peptide synthetases, which may be expected based on the particularly large clusters they can encompass (A) and the BGCs studied in this work (antiSMASH) to have a similar or higher number of genes than the characterized BGCs in MIBiG (B).

Comment #5

5. Since the authors mention that *Ca. Eudoremicrobiaceae* can be found more in eukaryote-enriched samples, why are they not considering a potential symbiotic state (with small eukaryotes, like diatoms or dinoflagellates) for species from this family?

We would like to thank the reviewer for pointing out this potential misunderstanding. We meant to include the possibility of a symbiotic lifestyle by using the expression 'particle-attached lifestyle' (considering organisms as particles). We have now rephrased the sentence in the main text. However, based on the large genome size (9.63 Mbp, see response to comment #9), it appears unlikely that *Ca. Eudoremicrobiaceae* are obligate symbionts. They could still be facultative/mutualistic symbionts; however, the gene-content analyses we performed (Supplementary Information, 'Genomic trait prediction in *Ca. Eudoremicrobiaceae* spp.')

 favors the likelihood of a predatory lifestyle as discussed in the main text and (now) Supplementary Information.

ACTIONS TAKEN:

- We rephrased the corresponding sentence in the main text to read:

"Furthermore, we found the relative abundances of Ca. Eudoremicrobiaceae spp. and their BGCs expression levels to be highest in eukaryote-enriched fractions (Figure 3C, S7), suggesting possible interactions with particulate matter including planktonic organisms. This observation, and the homology of some Ca. Eudoremicrobium BGCs to known pathways producing cytotoxic natural products could suggest a predatory behavior (Supplementary Information), akin to other specialized metabolites-producing predators such as Myxococcus⁴¹."

Comment #6

6. Similar to point 2 above, I did not see consideration of sequencing depth when analyzing the metatranscriptomic data.

As explained in our response to comment #2, all profiles were marker gene normalised rather than sequencing depth normalised to prevent genome size biases while still accounting for the sequencing depth of a sample.

ACTIONS TAKEN:

- We clarified in the methods that the marker gene normalisation also accounts for the sequencing effort.

Comment #7

7. It would be super useful for the community if the metatranscriptomic analysis is expanded to all MAGs and BGCs and not only *Ca. E. taraoceanii*. If metatranscriptomic information is tied with the genomes and BGCs on the web portal (and in supplementary Information), it would certainly help interested scientists in using this resource to pick additional BGCs for characterization or genomes to study further.

We appreciate this feedback and in an effort to make this resource as useful for the community as possible, we completely overhauled the companion website for the OMD (microbiomics.io/ocean/) and included several new features.

ACTIONS TAKEN:

- We developed a whole new web framework that now enables users to explore any genome or any BGC present in the database (Figure R1.7.1). We designed genome and BGC index pages, which provide, as requested, an overview of both metagenomic and metatranscriptomic abundances for each genome or BGC (Figure R1.7.2). This implementation allows users to rapidly draw ecological insights from our work. For example, the BGC highlighted in Figure R1.7.2 is a T3PKS enriched and preferentially transcribed in polar regions.
- We implemented a search function, enabling the user to perform fast, graph-based nucleotide searches on the complete genome collection (Figure R1.7.3).
- We added pre-computed GCF metagenomic and metatranscriptomic profiles as downloadable files.

Microbiomics

Ocean Home Genome Collection Genome Search Biosynthetic Gene Clusters Map Supporting data

Browse the genome collection

README

Genome Collection Page 1 of 6730

Display 5 records per page

Select all Deselect all Clear Column Reordering Export selection Toggle columns Search:

SELECT	GENOME	# BIOSYNTHETIC REGIONS	# BIOSYNTHETIC PRODUCTS	RIPPS	RIPPS-PROTEUSINS	NRPS	PKSI	OTHER PKS	SACCHARIDES	TERPENES
[ ]	BATS_SAMN07137064_METAG_ADEDPHOH	0	0	0	0	0	0	0	0	0
[ ]	BATS_SAMN07137064_METAG_BDCGNAHA	1	1	0	0	0	0	0	0	1
[ ]	BATS_SAMN07137064_METAG_DMDDONGP	1	1	0	0	0	0	0	0	0
[ ]	BATS_SAMN07137064_METAG_SFIEENIK	0	0	0	0	0	0	0	0	0
[ ]	BATS_SAMN07137064_METAG_EHLJPLHG	0	0	0	0	0	0	0	0	0

Showing 1 to 5 of 33,647 entries (33,647 filtered from 34,815 total records)

Filters Active - 1

First Previous 1 2 3 4 5 ... 6730 Next Last

Collapse All Show All Clear All

# Biosynt...	# Biosynt...	RIPPs	RIPPs:Pro...	NRPS	PKSI	Other PK
0	14823	0	14823	0	31798	32372
1	9891	1	9780	1	1251	1008
10	23	10	51	2	15	80
11	16	11	34	3	8	29
12	15	12	19	4	1	5
13	9	13	18	5	5	5

Figure R1.7.1. An overview of the new website design showing the genome collection table.

Figure R1.7.2. The new version of the website includes index pages for both genomes and BGCs. These index pages display their respective annotations as well as the distribution of metagenomic and metatranscriptomic abundances across geography, depth and size fraction. Here, we show an example of a T3PKS BGC showing a bipolar distribution with enrichment and higher transcription levels in polar regions.

Microbiomics
Tools
Databases
About
Contact

Ocean Home Genome Collection Genome Search Biosynthetic Gene Clusters Map Supporting data

Search in the genome collection

TCATATATTTTGAATCATCAGAGAAAGAAATTAACCTACTTACCAATACAGATTTTCCGAAACTATACAAAT
TTTAAACAAATAGTAAACACATGATTTTC

k-mer count cutoff: 50%

50.0%

Search

#	SCAFFOLD	GENOME	K-MER MATCHES
0	GORG_AG910L17_SAGS-scaffold_1	GORG_SAMEA6072986_SAGS_AG910L17	78

Genome Collection Page 1 of 1

Display 5 records per page Search:

SELECT	SCAFFOLD	K-MER MATCHES	GENOME	# BIOSYNTHETIC REGIONS
[ ]	GORG_AG910L17_SAGS-scaffold_1	78	GORG_SAMEA6072986_SAGS_AG910L17	0

Showing 1 to 1 of 1 entries (1 filtered from 34,815 total records) First Previous 1 Next Last

Copyright © 2021 | All rights reserved by Sunagawa Lab

Figure R1.7.3. To help users search for a specific genome, BGC or gene represented in the OMD, we implemented a fast, graph-based, kmer search across all genomes.

Comment #8

8. Related to the previous points, what fraction of *Ca. E. taraoceanii* genes is typically covered by metatranscriptomic reads before performing the UMAP/DBSCAN? If the data here is sparse, meaning that only a small number of genes are found to be expressed under each condition, then the clustering can be less meaningful – similar to the issue encountered in eukaryotic single cell transcriptomic analyses. I do not think that the transcriptomic clustering in Figure 4D is providing any ecological insights (see point below),

We appreciate the concern and note that we ensured through metagenomic analyses that the genes were present in the profiled metatranscriptomic samples to make sure differences were not driven by gene content variation (Figure S7). To ensure that the clustering was not driven by detection biases, we only included samples in which a minimum proportion of universal single-copy marker genes (6 out of 10) was detected and inspected the total number of genes detected in each metatranscriptomic sample, available in Table S4. However, to address this concern, we have added two new panels (Figure R1.8.1) to the relevant figure (Figure S7), displaying both the distribution of genes according to the number of samples they are detected in and the number of genes detected per metatranscriptomic samples. Notably, the minimal number of genes detected in the included samples is >1,350 and ~1,000 genes are detected across 90% of the samples, highlighting that the clustering is not driven by excessive data sparsity.

ACTIONS TAKEN:

- We added two panels to figure S7 displaying the distribution of gene detection across samples and the minimum number of genes detected in each metatranscriptomic sample.
- We specified the universal single-copy marker gene detection threshold in the corresponding figure.

Figure R1.8.1. Included in the supplementary Information as Figure S7. The 28 metatranscriptomic samples used for the *Ca. E. taraoceanii* expression analyses were selected based on the detection of at least 6 out of 10 universal single-copy marker genes. (A) distribution of the number of genes depending on the number of samples they were detected in. (B) Number of genes detected across the different metatranscriptomic samples.

Comment #9

9. In my opinion, the section named “putative ecology of *Ca. Eudoremicrobium* spp.” is the least supported section of the manuscript and the manuscript will be stronger without it. Inferring ecology simply by looking at genomes and transcriptomes and without any biological context is guessing at best – especially when the genome is also not 100% trusted (again, because of potential issues with binning). The statement about the two states of lifestyle is not supported, the statement about niche competition using natural products is not supported, etc. It is totally fine to call it what it is: a new species of marine bacteria, widespread in the Oceans (I suggest presenting a main figure piece that reflects this), many of its genes are actively transcribed (see point 7 above), and is biosynthetically rich. Beyond that is mostly mere speculation, especially when it is presented as “ecology”.

We appreciate the reviewer's concern and accordingly reduced the entire section into a minimal description of *Ca. Eudoremicrobium* spp.'s genomic and transcriptomic abundance distribution and the short discussion of its gene content relating it to other predatory bacteria (see response to comment #5), while refraining from the use of (putative) ecology in the main text. The original material has been reworked into a supplementary discussion on the observed patterns of BGC expression and gene content in *Ca. Eudoremicrobium taraoceanii*.

Regarding the uncertainty in the trustworthiness of the binned genomes, we included new results reporting a chromosome-level assembly of the genome of *Ca. Eudoremicrobium malaspinii*, which was recovered through ultra low-input PacBio metagenomic sequencing to address this point as well as comment #11.

ACTIONS TAKEN:

- We replaced the ‘Putative ecology’ section with a much reduced and rephrased passage to complement the description of the distribution and gene expression/content of *Ca. Eudoremicrobiaceae* spp.
- We subjected small amounts of leftover DNA to ultra low-input PacBio metagenomic sequencing to reconstruct the genome of *Ca. Eudoremicrobium malaspinii*, and incorporated the results into the main text. The corresponding passage reads:

*“The short-read metagenomic reconstruction of *Ca. E. malaspinii* draft genomes were corroborated by ultra-low input, long-read metagenomic sequencing and targeted assembly for one sample (Methods) into a single 9.63 Mbp linear chromosome with a 75 kbp repeat as the only remaining ambiguity.”*

- We have added a panel to (now) figure 3 to display the prevalence and abundance of *Ca. Eudoremicrobiaceae* spp.

Comment #10

10. Related to the previous point, the expansion of the “ecology” section came at the expense of the natural product discovery/enzymology section – which has the only concrete experimental results in the paper, represents a tremendous effort and is beautifully and thoroughly explained in the supplementary text. In my opinion, this section should be

expanded in the main text –honestly, it can make its own (great) paper. More discussion about the characterized BGCs and the clever methods used for characterization can definitely be promoted to the main text.

We very much appreciate the positive feedback on the experimental part of the work, which is also shared by other reviewers. Following the reviewer's advice, we toned down the purely computational analyses on the putative ecology of *Ca. Eudoremicrobiaceae* to expand this section on the biochemical characterisation of the selected BGCs.

ACTIONS TAKEN:

- We have split figure 4 into two to leave a full figure dedicated to the BGC biochemistry, allowing for the display of the full chemical structures of the products as well as references to some of the methods used (e.g. inset on the NMR data for the unusual amide *N*-methylation in the proteusin natural product).
- We extended the biochemistry section to include details on the novelty of the proteusin, additional context on the identification of a new amide *N*-methyltransferase, and mention the conservation of the mutation in the dehydration catalytic site (see below).
- We have included new NMR results for the phosphorylated peptide to further support the modifications observed using MS/MS and additional mentions to the methods used.

Comment #11

11. Line 338, how confident are the authors that these point mutations in the dehydration catalytic site are real and not an artifact of metagenomic assembly? Are they seen over and over in different samples?

We thank the reviewer for this very good point. We indeed made sure that this observation was not a result of an assembly error. This was reported in the original version already (Supplementary Text stating: "This is supported by the consistency of the mutations across all MAGs containing this BGC."). However, we agree this could have been better communicated. We added this information to the main text and show a multiple sequence alignment as a panel in Figure S8.

ACTIONS TAKEN:

- We rephrased the main text to reflect that observation by stating:
"[...] the identification of conserved mutated key residues in the dehydration catalytic site of the maturase, which was consistently found in all reconstructed *Ca. E. malaspinii* genomes (Figure S8)."

Figure R1.11.1. Included in the supplementary Information as Figure S8E. Multiple sequence alignment showing the residues in the dehydration catalytic site. Mutated residues are matched based on the structural modelling comparisons of EreM and CylM.

Reviewer #2: meta-omic analyses, microbial ecology in marine environment

General

This study by Paoli et al uses a large-scale MAG based effort to describe the biosynthetic potential of the open ocean microbiome. There are two parts to this manuscript - 1. Construction of a comprehensive genome database termed as the Ocean Microbiomics Database that encompasses microbial genomes aggregated from a wide variety of source datasets, 2. Scanning the Ocean Microbiomics database for biosynthetic gene clusters (BGCs), profiling the activity of these identified clusters, and characterizing some of them using heterologous expression. Overall, there is a lot to like about this paper. The scope of the paper is significantly large, the identification of the BGCs and the expansion of clusters is also significant. This will be a good resource that can be used to survey for important natural products in the coming years. However, I also have some important criticisms of this work that may preclude its widespread use and adoption. Specifically, the two parts of the study (1) and (2) listed above seem to have been conducted with different levels of rigor. While (2) is solid and important work in my opinion, (1) seems exaggerated and I am unable to place its importance in context. While I do understand that the authors needed to place the BGCs in a global ocean context (hence the rationale for an Ocean Microbiomics Database), nevertheless there seem to be significant shortcomings here.

Overall, I think this is an exciting study that I am very happy to have read. With changes this would be an important contribution to the fields of microbial ecology and natural products going forward.

We thank the reviewer for the overall positive feedback, including that this work would be a valuable contribution to the fields of microbial ecology and natural products, which is well aligned with the views expressed by Reviewer #1. We also acknowledge the reservations expressed and hope that the detailed point-by-point response below along with the revised manuscript will address the concerns.

Major concerns

Comment #1

Repeatedly (including in the abstract) the authors mention that they reconstructed ~25000 genomes. This is not true as stated in the methods. Rather these genomes were aggregated from multiple studies. It is also unclear to me as to how much (if any overlap) exists between the other Tara Ocean MAG studies (such as with Delmont et al). Presumably, all of these studies are using the Tara ocean read datasets to assemble the genomes. While I do understand that this would be taken care of during dereplication – it is not clear if genomes from other datasets were also included in such analyses. As such, the total number of genomes used sounds large but I am not sure if it is relevant or accurate.

We appreciate the reviewer's comment, which suggests a lack of clarity regarding the reported numbers of reconstructed MAGs and the overlap of these MAGs with data released by prior studies. To clarify, the reported numbers and statement are both correct, that is, we did reconstruct ~25,000 genomes *de novo* from publicly available metagenomes, some of

which (288 out of 1,038) were previously used for reconstructing MAGs in prior studies (Parks et al. 2017, Tully et al. 2018, Delmont et al. 2018). The reason for re-processing the fraction of overlapping data was that none of these studies reconstructed MAGs using both single-sample assemblies and co-abundance correlation for quality improvement. The reviewer sharing this view is reassuring for the additional investment we made into establishing a consistent set of MAGs according to high quality standards.

We then complemented these 25k MAGs with ca. 10k additional (externally assembled) genomes mainly from collections of publicly available SAGs and reference genomes, in addition to manually curated MAGs (Delmont et al. 2018) for benchmarking purposes (see below). All 35k genomes were, as the reviewer mentioned, subsequently dereplicated to allow for subsequent analyses.

The overlap with data from prior studies is illustrated in Figure 1E-F, showing that this is restricted to <20% of the 5,000 'species' represented by the 25,000 new MAGs. We additionally show in Figure S12 comparisons and quality benchmarks with the previous MAG reconstruction efforts mentioned above (Parks et al. 2017, Tully et al. 2018, Delmont et al. 2018).

ACTIONS TAKEN:

- We have modified Figure 1, notably panel B and E to improve the presentation of what are newly reconstructed genomes in this work vs. those included as they were available from external (publicly-available) sources.

Comment #2

Along similar lines, it is mentioned that the genomes come from different size fractions including virus-enriched and also some eukaryote genomes. Why is this relevant? Were any BGCs identified from these fractions/genomes? Fig 2A and other parts of Fig 2 does show some of these details but the figure is complex to interpret and the colors associated with the size fractions are difficult to separate.

We would like to clarify that we first included virus- and eukaryote- enriched fractions as they have been shown to contain prokaryotic genomes, and indeed, we recovered a total of 3,414 genomes from these metagenomes. Then, as displayed in Figures S2D and S4D (now Figures S5C and S6A), we found the genomes recovered from these fractions to be associated with high abundances of BGCs, including novel ones. To address other editorial and reviewers' comments, we have now moved results associated with parts of Figure 2 to Figure S5, which allowed us to provide a clearer representation of the different size fractions in the relevant panel, as suggested here. Furthermore, we added a supplementary discussion point to specifically report on the distribution of BGCs and genomes recovered across the different size fractions, and we clarified our mention of eukaryotic MAGs in the Supplementary Information.

ACTIONS TAKEN:

- We have clarified the relevant panel of Figure 2 (now figure S5C) by expliciting the different size-fractions in addition to the color coding..

- We have added a supplementary discussion point (*'Recovery of BGCs across different size fractions'*) to report that:

"Out of the 39,055 BGCs analysed in this work, 28,109 were predicted in the newly reconstructed MAGs. Among these, 8,619 (31%) were predicted in genomes from virus-enriched fractions, 16,350 (58%) in genomes from prokaryote-enriched fractions, 781 (3%) in genomes from particle-enriched fractions and 2,359 (8%) in genomes from virus-depleted fractions."

and further discuss these numbers in the context of the fraction of genomes reconstructed per size-fraction.

- We removed the sentence *'This notably allowed for the identification of 148 Eukaryotic MAGs by Anvi'o'* in the methods, which was a potential source of confusion, to mention that *'Excluding eukaryotes (148 MAGs), each genome was functionally annotated [...]'*.

Comment #3

I have some concerns with regard to novelty of these BGCs. Have these new BGCs never been described before? The authors compare the BGCs to two recently developed databases mibig and bigfam – I think these analyses are great. But what about the context of BGCs in the open ocean when compared to other environments – it was my understanding that the authors also set out to do this? I also bring this up specifically because the authors mention Eremiobacterota as being BGC rich and only being previously found in terrestrial environments. Given the enormous types, range, and number of BGCs discovered, I think this could be useful analyses to add and provide context for BGCs in the oceans.

We would like to clarify that MIBiG and BiGFAM contain characterized and predicted BGCs from any deposited genome irrespective of the sampled environment. As such, when comparing the set of BGCs identified in this study to these databases, we ensure that any BGC identified as novel has not previously been predicted in microbial genomes recovered from the ocean or elsewhere (as represented by the NCBI RefSeq database).

With regards to contextualising the ocean BGCs compared to other ecosystems, we do mention some of the differences, e.g. as we identify other ecosystems to be enriched in PKS and NRPS compared to the ocean, which is richer in RiPPs and Terpenes. However, a more focused comparative analysis (which would require disentangling sampling and reference biases) would certainly be interesting, but deviate from the core results of this work and thus out of scope of the presented work.

ACTIONS TAKEN:

- We clarified in the main text that we compared the BGCs predicted in our work to BGCs *'from any of the ~190,000 genomes deposited in the NCBI RefSeq database'*, and thus included all environments represented in the NCBI genomic database, to evaluate their novelty.

Comment #4

Data is open but practically unusable. I commend the authors for creating a website for OMD – the website interface is clean and easily interpretable. However, what I find to be unacceptable is that only the assemblies have been deposited in NCBI/ENI. Even today, few groups have the resources to download and run analyses on large datasets such as this. The MAGs and BGCs will essentially be unsearchable and unusable at a broad scale if they are NOT deposited in these databases. I encourage the authors to deposit the MAGs and proteins in NCBI/ENI. Without this, I can imagine another group coming along in the coming years and describing novelty based on this data.

We would like to thank the reviewer for bringing up this important point. To provide context, we initially refrained from doing so as the quality of public repositories has been negatively impacted by the deposition of MAGs in the past (Shaiber and Eren 2019; Orakov et al. 2021; Arkhipova 2020). However, after further inspection, we feel ENA provides an appropriate solution and we have now submitted all bins (including all reconstructed MAGs) at ENA using the ‘binned metagenomic assembly’ option (they are scheduled by ENA for release on the 5th of January 2022).

In addition, we implemented a search function on the companion web database to enable users to search the complete genome collection for a specific genome, BGC or gene.

ACTIONS TAKEN:

- The binning results have been deposited at ENA as ‘binned metagenomic assemblies’ under the project PRJEB45951.
- We have also submitted to ENA the new long-read-based chromosome-level assembly of *Ca. E. malaspinii*.
- We implemented a search function, enabling the user to perform fast, graph-based nucleotide searches on the complete genome collection (Figure R1.7.3).

Methods

Comment #5

The methods for determination of BGCs, classification, and expression are well done. I have no concerns here.

We would like to thank the reviewer for this positive assessment.

Comment #6

The idea to not use individual assemblies in lieu of coassemblies was an excellent choice.

We would like to thank the reviewer for this positive assessment.

Comment #7

I mentioned having criticisms about the development of the MAG database above. Along those lines, the description of binning in the manuscript is shockingly elementary. The authors mention using abundance correlation to improve the recovery of MAGs. This has

been known for the best part of 10 years and there are at the very least 10+ binners and countless studies that use differential abundance coverage binning (See papers on Metabat, Metabat2, Maxbin, Concoct, Abawaca, Cocacola etc). These findings are not new or novel. While I appreciate using this form of binning, I am unsure if any other form of binning would even be acceptable in the field as things stand.

We appreciate the external view provided here and would like to (1) clarify that the high-level description of the binning step in the main text was tailored for a broad audience leaving the details for methods and Supplementary Information, and (2) acknowledge that the binning strategy used here (single-sample assembly and abundance correlation binning) is not new or novel, yet this approach is not as widely used in the field as we believe it should, hence the emphasis. Specifically, none of the major studies releasing global ocean MAGs have used a combination of single-sample assembly and abundance correlation binning:

- Parks et al. 2017: The authors reconstructed 1.4k MAGs from the global ocean using single sample assemblies but without abundance correlation binning.
- Tully, Graham, and Heidelberg 2018: The authors reconstructed 2.6k MAGs from the global ocean using single-assembly, followed by regional co-assembly of the resulting contigs, and included abundance correlation for binning.
- Delmont et al. 2018: The authors reconstructed <1k manually curated MAGs from the global ocean using regional co-assembly and included abundance correlation for binning.
- Nayfach et al. 2020: The authors reconstructed 5.9k MAGs from the global ocean using single-sample assembly but without abundance correlation.

We further note that the three major Human Gut Microbiome MAG papers used single-sample assemblies, but did not use abundance correlation binning.

As we fully agree with the reviewer that binning with abundance correlation should be the only form of binning acceptable in the field and, while we did not intend to claim this as a novelty from our side, we attempted to highlight the approach in association with a large-scale benchmark demonstrating the improved quality of the MAGs binned with abundance correlation.

In response to this comment, we have rephrased relevant sentences to make sure the phrasing conveys this message rather than claiming novelty on this approach.

ACTIONS TAKEN:

- We have added additional details to the methods on the choice of the software used and set up a repository with the code for the MAG reconstruction pipeline to provide the highest possible level of details: <https://github.com/SushiLab/magpipe>.
- We have rephrased the main text to ensure that abundance correlation is not perceived as a novelty of this work:

“We found this computationally intensive, yet important step²⁴, which was omitted in several large-scale MAG reconstruction efforts^{16,19,25}, to significantly improve both the number (mean 2.7 times) and quality score (mean +20%) of genomes reconstructed from the ocean metagenomes studied here (Figure S3, Supplementary Information).”

- We have added descriptions of the previous ocean MAG datasets (Parks et al. 2017, Delmont et al. 2018, Tully et al. 2018, Nayfach et al. 2020) and highlighted the change of scale in this work in the relevant supplementary discussion point 'Reconstructed MAGs have improved qualities compared to previous efforts'.

Comment #8

Binning – I am also unsure as to why only one binner (Metabat2) was used in this study. Considering the authors have benchmarked their analyses (I commend them for this), I would have imagined that the use of multiple binners followed by aggregation using tools such as (Dastool, Metawrap etc) would have performed better. Alternatively given that the approaches described above are “more standard” in the field, I would have liked to see some justification as to why the authors preferred to do it their way. This would also justify as to why their approach is better than what the authors define as automatically curated MAGs (given that those MAGs were also identified using differential abundance binning – albeit with different binners and also include contamination analyses).

We fully understand the point raised by the reviewer and acknowledge that ensemble binning methods (aggregating the results of several independent methods) have been shown to produce better results than single binners. However, considering the size of the data processed, we had to strike a balance between the best possible methods and the computational load. Specifically, ensemble binning approaches multiply the computational load by integrating multiple binners, some of which did not scale well with both the size of the assemblies and the number of samples used for abundance correlation. For instance, we found in our own benchmarks that vamb took close to ten times longer than MetaBAT2 (data not shown), a value now corroborated by the latest CAMI challenge (F. Meyer et al. 2021). Such benchmarks also found most binners to take ten times longer than MetaBAT2 on a test marine dataset and, in particular, MaxBin2 was 50 times slower. As such, we used a performant single binner approach that was shown to perform best in independent comparisons (Fernando Meyer et al. 2018).

ACTIONS TAKEN:

- We have clarified the choice of the binner in the methods.

Minor concerns

Line 135: state this to be the OMD.

We thank the reviewer for this suggestion. However, we would like to note that the OMD contains more than the genomes referred to in (formerly) line 135 and includes their annotations, in particular BGCs, the metagenomic and metatranscriptomic distributions of both genomes and BGCs as well as additional unbinned genomic fragments, including plasmids and phages, that may not be captured by the MAGs. As such, we feel it would be misleading to define the OMD at an earlier stage.

Reviewer #3: marine bacterial biosynthesis

General

Paoli and colleagues report an extensive data analysis of the global ocean microbiome that focuses on the biosynthetic potential of marine bacteria to produce bioactive compounds. The authors assembled >25k MAGs and identified nearly twice as many biosynthetic gene clusters (BGCs), many from new and unsuspected species. The massive scale of their work is reminiscent of several recent high-profile articles published in Nature 2018 on soil microbes (ref 18) and Nat Biotech 2021 on the Earth microbiome (ref 19) that also detailed the biosynthetic potential of newly identified bacterial groups. Like those recent reports, the authors identified new bacterial species from their respective environments that were biosynthetically “gifted” in terms of encoding diverse and plentiful BGCs.

The Paoli paper deviates from those other publications in one important way. While the others were primarily computational, Paoli additionally provides experimental data to substantiate their biosynthetic claims. The authors identified two BGCs encoding ribosomally-encoded and post-translationally modified peptides from a new BGC-rich actinobacterial lineage belonging to an uncultivated phylum. Through a series of rigorous in vivo and in vitro expression experiments and chemical analyses, they report the production of two complex peptide molecules that are new and unusual. Both molecules as reported in Fig 4 have structure features not well represented in the biosynthetic literature, although their structure features, namely amine methylation and phosphorylation, are ubiquitous in the natural product literature. The authors should be commended for their impressive biochemical work, **yet they should tone down their claims that split hairs. The work is impressive by itself yet is distracted with unneeded posturing.**

Overall, I was impressed with the sheer volume of data presented and the identification of a new biosynthetically rich marine lineage, Ca. Eudoremicrobium spp.

The paper is purely discovery based looking for something to discover without a clear hypothesis or motivation. Broadly speaking, the result is largely anticipated based on recent large-scale microbiome studies that new lineages will be discovered when you look hard enough in the global landscape, and that they will harbor new BGCs encoding new molecules that may support their specific lifestyles. **The authors should more clearly clarify their motivations and how this report differs from other recent large scale microbiome studies.**

We thank the reviewer for the positive feedback recognizing the added value of combining the computational predictions with biochemical validations. We are also grateful for all the constructive comments and acknowledge the expressed reservations. Based on these remarks, we have reframed the manuscript to better emphasize the biochemical part of the work, clarifying that establishing the database was a necessary step to facilitate the exploration of the diversity and novelty of the ocean microbiome’s biosynthetic potential, and this way, also differs from other work in its entirety. Regarding the biochemical work, we have generated new supportive data and generally toned down the wording, also by providing

better context when communicating the novelty. We hope that the following point-by-point response and actions taken will address any remaining concerns.

Below are several items for the authors to additionally consider in a revision if encouraged.

Comment #1

1. Prior to this work, Tully et al reported in 2018 in *Sci Data* over 2.6k MAGs from the global ocean also using data collected by the Tara Oceans expedition that seemingly provided data for this paper's analysis. The Tully paper is not cited nor discussed, yet there may appear to be redundancy. This should be clarified.

We acknowledge that the paper by Tully et al. reconstructed MAGs from a subset (234 out of 1,038) of the metagenomes used here to reconstruct new genomes. We would like to clarify that we did not use the MAGs reconstructed by Tully et al., but rather reconstructed MAGs *de novo* to build the database used in this work. However, we would like to note that we cite and discuss Tully et al. (along with other previously established MAG datasets: Parks et al. 2017, Tully et al. 2018, Delmont et al. 2018), in the Supplementary Information to evaluate the overlap and compare the quality of MAGs reported in different studies to the newly reconstructed ones in this work (Figure S12). Furthermore, as the MAGs generated by Tully et al. are included in the GTDB database, any redundancy of the newly reconstructed MAGs is reflected in our analysis behind Figure 1D.

ACTIONS TAKEN:

- We updated Figure 1 to clarify differences between newly reconstructed MAGs and external, publicly available genomes as well as their redundancy.
- We clarified the distinctions with previous ocean MAG datasets (Parks et al. 2017, Delmont et al. 2018, Tully et al. 2018, Nayfach et al. 2020) and the change in scale in our work (see response to Reviewer #2, comment #7) in the supplementary discussion point 'Reconstructed MAGs have improved qualities compared to previous efforts'.

Comment #2

2. The authors suggest that the expression of BGCs by *Ca. E. taraoceanii* implies "a niche-partitioning role for the candidate natural products they encode". That statement could be said frankly of any bacterium, especially those that encode specialized metabolites. The challenge with the two candidate natural products reported in this manuscript is that the compounds may not be physiologically relevant due to other biosynthetic processes not captured by the heterologous expression experiments. Moreover, the structures are suggested based on MS data and biosynthetic precedence, yet rigorous, independent analyses based on NMR are missing.

We acknowledge that (1) a niche-partitioning role could be expected for most natural products. In fact, we interpret the differential expression of BGCs rather as evidence for their existence and functionality of these BGCs (i.e., rather than being computational artifacts). Nevertheless, replaced the reference to a broad 'niche-partitioning role' by a more specific

interpretation in the context of a putative 'secondary metabolite-driven predatory behavior' in the corresponding discussion point (now Supplementary Information).

We would also like to clarify that (2) heterologous experiments indeed may not lead to the final natural products due to unreproducible biosynthetic processes specific to the original host. For the first biosynthetic cluster (proteusin), we thus focused the analyses on the characterization of the different maturase enzymes, and compiled the results to propose a structure for the final natural product. For the second biosynthetic cluster (phosphorylated peptide), we collected results along several, independent lines of evidence suggesting that the characterised product is indeed the actual natural product, including additional NMR experiments. Regarding this latter point, we thank the reviewer for the implied suggestion (in the general comment above) to better highlight the challenge of the biochemical work (performing NMR analyses on core peptides of 46 and 18 amino acids, respectively, with moderate production yields in heterologous hosts systems).

Overall, we would like to note that the most ambiguous and relevant modification (amide-N-methylation) observed in the proteusin cluster was confirmed using HSQC NMR. We have now highlighted these results by adding an inset of the NMR data in Figure 4. In addition, we have conducted further NMR experiments on the phosphorylated peptide to confirm the two phosphorylations that were previously suggested solely based on MS/MS data.

ACTIONS TAKEN:

- We have adjusted the mention of a niche-partitioning role for BGC products in the corresponding supplementary discussion point.
- We have adjusted the main text to better reflect the limits of heterologous expression experiments. Namely, we (1) in the case of the phosphorylated peptide, replaced '*we identified the final product to be*' by the more accurate '*several lines of evidence supported it to be the final product*', and (2) in the case of the proteusin pathway we explicitly mention that our results lead to a '*proposed structure for the product*'.
- We have produced NMR data for the phosphorylated peptide to confirm MS/MS results (Figure S9).
- We have highlighted NMR results that were confirming the backbone amide-N-methylation by adding an inset in Figure 4.

Comment #3

3. The biological data showing that the diphosphorylated peptide is active against elastase but not other proteases may be an in vitro artefact and not biologically relevant as phosphorylated peptides will generally not be cell penetrant, perhaps explaining the lack of activity in a whole cell assay. This may suggest that the compound described may not be the biosynthetic product of the bacterium.

We (1) acknowledge that due to some intrinsic limitations of heterologous expression, we cannot rule out that the product observed is a late intermediate (see above). However, we have accumulated several independent lines of evidence suggesting that the proposed structure represents the final product (heterologous expression in multiple hosts, mutation in

the dehydration catalytic site supported by multiple independent assemblies and bioactivity). We have made adjustments to the relevant section in the main text to account for that point.

We (2) would like to note: that there is precedence for cell-penetrating phosphorylated natural products (e.g., calyculins are inhibitors of intracellular protein phosphatases), that the product could also target secreted proteases or even proteinases of the producer itself, that the natural product may also be delivered through contact-dependent strategies, or that there may be synergistic effects with other cell-disruptant natural products (e.g., polymyxin, polytheonamides). Together, this suggests that our observations are not incompatible with the idea of the proposed product having biological relevance. Irrespective of biological relevance however, the detection of elastase inhibitory activity by a new phosphorylated peptide may by itself be of biotechnological interest.

ACTIONS TAKEN:

- We have better contextualized our proposition of the identified phosphorylated product as the final natural product and added a disclaimer to clarify that the observed bioactivity may not reflect the biological relevance of the final product. The corresponding text reads: “[...], we found several lines of evidence supported it to be the final product: the absence of dehydration in two different heterologous hosts as well as in vitro assays, the identification of mutated key residues in the dehydration catalytic site of the maturase, which were consistently found in all reconstructed *Ca. E. malaspinii* genomes (Figure S8, Supplementary Information), and finally, the bioactivity of the phosphorylated product rather than the chemically synthesised, dehydrated form (Figure 4D). Indeed, we found it to display low-micromolar protease inhibitory activity against neutrophil elastase within a concentration range (IC_{50} 14.3 μ M) comparable to other relevant natural products⁴⁴, although the ecological role of this unusual natural product remains to be elucidated”

Comment #4

4. Is there a reason why the authors focused on the expression of the two specific RiPP BGCs, other than one of the corresponding authors being an expert in this BGC type? Please explain and justify. If for a “biotechnological interest” (line 318), then a reference beyond a self-reference is preferred that shows genuine interest.

We thank the reviewer for the opportunity to clarify this point. Both RiPP clusters were selected based on the fact that they were predicted to be novel by computational methods and found to encode unusual maturases upon manual inspection.

ACTIONS TAKEN:

- We have added an additional sentence to motivate the selection of RiPPs based on a well-cited review: “Among the different BGC classes, RiPP pathways are known to encode for a wealth of chemical and functional diversity thanks to the various modifications installed post translationally on a core peptide by maturase enzymes⁴²” and better justify the selection of the two specific clusters based on their

computationally predicted novelty: “We thus selected two *Ca. Eudoremicrobium* RiPP BGCs (Figure 3B and 4A-E) that were predicted to produce novel metabolites based on their dissimilarity to any known BGC (\bar{d} MIBiG and \bar{d} RefSeq above 0.2)”.

Comment #5

5. The authors discuss that they queried a number of unique terpenoid BGCs, yet they report common hopanoids and carotenoids. Please expand.

We would like to clarify that in the analysis behind Figure 2, we uncovered a set of ~40k BGCs that were clustered into ~7k GCFs. Among these, we found numerous terpene BGCs with potential for novelty, some that may represent new subclasses of compounds (distances up to 0.89 from MIBiG or 0.63 to BiGFAM/RefSeq) while others may be new products within more well known subclasses (e.g. hopanoids or carotenoids). To better represent this putative terpene diversity and novelty, we have now computed the phylogenetic tree of all identified terpene synthases in our database as displayed in figure R3.5.1.

In Figure 4 (now Figure 3), we attempted to describe *Eudoremicrobium* spp.’s BGCs with as much specificity as possible, including through manual inspection and annotation. We thus annotated the terpenoid clusters as carotenoid or hopanoid based on the biosynthetic domains identified by antiSMASH (phytoene synthase and terpene cyclase, respectively). However, we would like to clarify that these three clusters are distant from characterized representatives (distance to MIBiG > 0.4, supported by their placement in the tree) and that two of them are predicted to be novel compared to what has previously been sequenced (distance > 0.2 to BiGFAM/RefSeq).

ACTIONS TAKEN:

- We have now included a phylogeny of terpene biosynthetic core genes, as defined by antiSMASH (Figure S6B) to provide an overview of the terpenoid diversity in our database
- We have added a reference to this new figure panel in the main text when we mention ‘*promising sources of new terpenes (Figure S6 A-B)*’.

Figure R3.5.1 (Included in supplementaries as SFigure 6B). A phylogenetic tree of all terpene biosynthetic core genes (as defined by antiSMASH) identified in the OMD, in the context of the 195 MIBiG terpene biosynthetic core genes, provides an overview of the terpenoid diversity and novelty. The three terpenoid clusters identified in *Ca. Eudoremicrobium* spp. and described in (now) figure 3 are highlighted. Briefly, the 31,398 terpene biosynthetic core genes identified across all predicted BGCs were filtered (length ≥ 120 aa, removing $< 2\%$ of the sequences), dereplicated (using MMSEQS2 clustering, 60% identity) into 2,904 protein sequences and aligned with the 195 MIBiG proteins using MAFFT. The resulting alignment was trimmed with trimal to remove positions with more than 50% gaps and used to build the tree (using fasttree). The inner annotation layers indicate whether a gene is coming from a MIBiG cluster and if this one was annotated as a carotenoid or hopene cluster. The outer layers correspond to the biosynthetic core gene domain according to antiSMASH categories. Plants were used to root the tree.

References

- Arkipova, Irina R. 2020. "Metagenome Proteins and Database Contamination." *mSphere* 5 (6). <https://doi.org/10.1128/mSphere.00854-20>.
- Beszteri, Bánk, Ben Temperton, Stephan Frickenhaus, and Stephen J. Giovannoni. 2010. "Average Genome Size: A Potential Source of Bias in Comparative Metagenomics." *The ISME Journal* 4 (8): 1075–77.
- Delmont, Tom O., Christopher Quince, Alon Shaiber, Özcan C. Esen, Sonny Tm Lee, Michael S. Rappé, Sandra L. McLellan, Sebastian Lücker, and A. Murat Eren. 2018. "Nitrogen-Fixing Populations of Planctomycetes and Proteobacteria Are Abundant in Surface Ocean Metagenomes." *Nature Microbiology* 3 (7): 804–13.
- Meyer, Fernando, Peter Hofmann, Peter Belmann, Ruben Garrido-Oter, Adrian Fritz, Alexander Sczyrba, and Alice C. McHardy. 2018. "AMBER: Assessment of Metagenome BinnERs." *GigaScience* 7 (6). <https://doi.org/10.1093/gigascience/giy069>.
- Meyer, F., A. Fritz, Z-L Deng, D. Koslicki, A. Gurevich, G. Robertson, M. Alser, et al. 2021. "Critical Assessment of Metagenome Interpretation - the Second Round of Challenges." *bioRxiv*. <https://doi.org/10.1101/2021.07.12.451567>.
- Milanese, Alessio, Daniel R. Mende, Lucas Paoli, Guillem Salazar, Hans-Joachim Ruscheweyh, Miguelangel Cuenca, Pascal Hingamp, et al. 2019. "Microbial Abundance, Activity and Population Genomic Profiling with mOTUs2." *Nature Communications* 10 (1): 1014.
- Nayfach, Stephen, Beltran Rodriguez-Mueller, Nandita Garud, and Katherine S. Pollard. 2016. "An Integrated Metagenomics Pipeline for Strain Profiling Reveals Novel Patterns of Bacterial Transmission and Biogeography." *Genome Research* 26 (11): 1612–25.
- Nayfach, Stephen, Simon Roux, Rekha Seshadri, Daniel Udvary, Neha Varghese, Frederik Schulz, Dongying Wu, et al. 2020. "A Genomic Catalog of Earth's Microbiomes." *Nature Biotechnology*, November. <https://doi.org/10.1038/s41587-020-0718-6>.
- Orakov, Askarbek, Anthony Fullam, Luis Pedro Coelho, Supriya Khedkar, Damian Szklarczyk, Daniel R. Mende, Thomas S. B. Schmidt, and Peer Bork. 2021. "GUNC: Detection of Chimerism and Contamination in Prokaryotic Genomes." *Genome Biology* 22 (1): 178.
- Pachiadaki, Maria G., Julia M. Brown, Joseph Brown, Oliver Bezuidt, Paul M. Berube, Steven J. Biller, Nicole J. Poulton, et al. 2019. "Charting the Complexity of the Marine Microbiome through Single-Cell Genomics." *Cell* 179 (7): 1623–35.e11.
- Parks, Donovan H., Christian Rinke, Maria Chuvochina, Pierre-Alain Chaumeil, Ben J. Woodcroft, Paul N. Evans, Philip Hugenholtz, and Gene W. Tyson. 2017. "Recovery of Nearly 8,000 Metagenome-Assembled Genomes Substantially Expands the Tree of Life." *Nature Microbiology* 2 (11): 1533–42.
- Salazar, Guillem, Lucas Paoli, Adriana Alberti, Jaime Huerta-Cepas, Hans-Joachim Ruscheweyh, Miguelangel Cuenca, Christopher M. Field, et al. 2019. "Gene Expression Changes and Community Turnover Differentially Shape the Global Ocean Metatranscriptome." *Cell* 179 (5): 1068–83.e21.
- Shaiber, Alon, and A. Murat Eren. 2019. "Composite Metagenome-Assembled Genomes Reduce the Quality of Public Genome Repositories." *mBio*. <https://doi.org/10.1128/mBio.00725-19>.

Sunagawa, Shinichi, Daniel R. Mende, Georg Zeller, Fernando Izquierdo-Carrasco, Simon A. Berger, Jens Roat Kultima, Luis Pedro Coelho, et al. 2013. "Metagenomic Species Profiling Using Universal Phylogenetic Marker Genes." *Nature Methods* 10 (12): 1196–99.

Tully, Benjamin J., Elaina D. Graham, and John F. Heidelberg. 2018. "The Reconstruction of 2,631 Draft Metagenome-Assembled Genomes from the Global Oceans." *Scientific Data* 5 (January): 170203.

Reviewer Reports on the First Revision:

Referees' comments:

Referee #1 (Remarks to the Author):

The authors have appropriately responded to all my comments and concerns, I believe the new manuscript has been greatly improved and is now ready for publication (congratulations for the chromosome-level assembly of the new bacterium and the new and improved portal!).

Referee #2 (Remarks to the Author):

I have now had the opportunity to revisit the manuscript by Paoli et al. I really commend the authors for making extensive revisions and addressing my comments. The text and the manuscript are more streamlined, and the additions and clarifications certainly help. Overall, I still believe that this will be an important contribution to the fields of microbial ecology and natural products.

I only have one major outstanding concern. Specifically, the authors should make it easier for people to access their data - I still cannot find their MAGs online. The response mentions that the MAGs were released on Jan 5 2022 but they do not seem to be linked to their bioproject in an easy to comprehend manner (Nor on ENA or their website). Could the authors include a supplementary table that links MAGs, accession numbers, taxonomy, and any other relevant details? This specific supplementary table should also be listed in the data availability section. This is very important in the context of important claims such as expansion of Ocean MAGs by 6x. This would also be extremely useful as I strongly believe that the majority of data usage will be through ENA rather than their website (I do like the website though!).

Referee #3 (Remarks to the Author):

The authors have done an excellent and thorough job in revising their manuscript. I am very satisfied with their responses and congratulate them on a very fine piece of work.

Author Rebuttals to First Revision:

Point-by-point response to reviewers (after revisions)

Referee #1 (Remarks to the Author):

The authors have appropriately responded to all my comments and concerns, I believe the new manuscript has been greatly improved and is now ready for publication (congratulations for the chromosome-level assembly of the new bacterium and the new and improved portal!).

We would like to thank the reviewer for the positive remarks and the recommendation for the manuscript to be published.

Referee #2 (Remarks to the Author):

I have now had the opportunity to revisit the manuscript by Paoli et al. I really commend the authors for making extensive revisions and addressing my comments. The text and the manuscript are more streamlined, and the additions and clarifications certainly help. Overall, I still believe that this will be an important contribution to the fields of microbial ecology and natural products.

We would like to thank the reviewer again for the constructive comments on the submitted version and the positive feedback on the revisions we made.

I only have one major outstanding concern. Specifically, the authors should make it easier for people to access their data - I still cannot find their MAGs online. The response mentions that the MAGs were released on Jan 5 2022 but they do not seem to be linked to their bioproject in an easy to comprehend manner (Nor on ENA or their website). Could the authors include a supplementary table that links MAGs, accession numbers, taxonomy, and any other relevant details? This specific supplementary table should also be listed in the data availability section. This is very important in the context of important claims such as expansion of Ocean MAGs by 6x. This would also be extremely useful as I strongly believe that the majority of data usage will be through ENA rather than their website (I do like the website though!).

We would like to clarify that the assemblies, bins and MAGs were available at ENA under the project accession specified in the Data Availability Statement (PRJEB45951), see <https://www.ebi.ac.uk/ena/browser/view/PRJEB45951>. However, would like to thank the reviewer for the suggestion to provide the individual accession numbers as a Supplementary Table.

ACTIONS TAKEN:

- We have included additional sheets to Supplementary Table S1 with accession numbers for the metagenomic assemblies and for the bins MAGs generated in this

study, and a summary table of all the genomes contained in the OMD. We refer to this table in the Data Availability Statement.

Referee #3 (Remarks to the Author):

The authors have done an excellent and thorough job in revising their manuscript. I am very satisfied with their responses and congratulate them on a very fine piece of work.

We would like to thank the reviewer again for the constructive comments on the submitted version and the positive feedback on the revisions we made.